# Probabilistic Retrofitting of Learned Simulators

Cristiana Diaconu [1 2]   Miles Cranmer [1 2]   Richard E. Turner [2 3]   Tanya Marwah [† 1]   Payel Mukhopadhyay [† 1 2]

## Abstract

Dominant approaches for modelling Partial Differential Equations (PDEs) rely on deterministic predictions, yet many physical systems of interest are inherently chaotic and uncertain. While training probabilistic models from scratch is possible, it is computationally expensive and fails to leverage the significant resources already invested in high-performing deterministic backbones. In this work, we adopt a training-efficient strategy to transform pre-trained deterministic models into probabilistic ones via retrofitting with a proper scoring rule: the Continuous Ranked Probability Score (CRPS). Crucially, this approach is architecture-agnostic: it applies the same adaptation mechanism across distinct model backbones with minimal code modifications. The method proves highly effective across different scales of pre-training: for models trained on single dynamical systems, we achieve 20–54% reductions in rollout CRPS and up to 30% improvements in variance-normalised RMSE (VRMSE) relative to compute-matched deterministic fine-tuning. We further validate our approach on a PDE foundation model, trained on multiple systems and retrofitted on the dataset of interest, to show that our probabilistic adaptation yields an improvement of up to 40% in CRPS and up to 15% in VRMSE compared to deterministic fine-tuning. Validated across diverse architectures and dynamics, our results show that probabilistic PDE modelling need not require retraining from scratch, but can be unlocked from existing deterministic backbones with modest additional training cost.[1]

[†]Equal advising, listed alphabetically [1]The Polymathic AI Collboration [2]University of Cambridge [3]Alan Turing Institute. Correspondence to: Cristiana Diaconu <cdd43@cam.ac.uk>, Payel Mukhopadhyay <pm858@cam.ac.uk>.

*Proceedings of the 43rd International Conference on Machine Learning*, Seoul, South Korea. PMLR 306, 2026. Copyright 2026 by the author(s).

[1]Code is provided at https://github.com/cddcam/lola_crps.

## 1. Introduction

The field of deep learning for Partial Differential Equation (PDE) modelling has seen explosive growth in recent years (Raissi et al., 2019; Li et al., 2021; Karniadakis et al., 2021). Accurately capturing these dynamics presents unique challenges arising from their diverse and complex physical nature. Many real-world systems are inherently chaotic, corrupted by observational noise, or driven by unobserved quantities acting on unresolved scales. These characteristics determine critical modelling decisions which can differ substantially from traditional numerical solvers, ranging from the choice of spatiotemporal resolution and the extent of historical context (Ruiz et al., 2025), to the trade-off between pixel- and latent-space representations (Alkin et al., 2024; Rozet et al., 2025). Despite this diversity in architectural design and modelling paradigm, the vast majority of current approaches converge on a deterministic training framework, optimising models to predict future states by minimising a mean squared or absolute error (MSE / MAE) loss.

While this deterministic paradigm is effective in stable, fully observed regimes, it is fundamentally ill-suited for the complex scenarios described above. This limitation is particularly acute for neural network surrogates, which typically operate at coarser spatiotemporal resolutions than classical numerical solvers, thereby exacerbating the impact of unresolved scales. In systems characterised by chaos and partial observability, a single deterministic prediction cannot capture the true distribution of plausible future states, often leading to physically inconsistent rollouts that fail to account for inherent uncertainty (Price et al., 2025; Alet et al., 2025; Ruiz et al., 2025). To address this, recent research (Lippe et al., 2024; Rozet & Louppe, 2023; Shysheya et al., 2024; Rozet et al., 2025) has begun to explore probabilistic frameworks, such as diffusion and flow-based generative models (Ho et al., 2020; Song et al., 2021; Lipman et al., 2023). Although these methods often yield superior performance and more stable rollouts, they impose a heavy computational burden: they require specialised, resource-intensive training regimes and often suffer from slow, iterative inference processes that can be prohibitive for real-time applications.

In this work, we propose a unified strategy to bridge this gap. We adopt a training-efficient framework for transforming pre-trained deterministic checkpoints into probabilis-

tic models via retrofitting with a proper scoring rule—the Continuous Ranked Probability Score (CRPS). Crucially, since the CRPS objective is closely related to the MAE and MSE losses used in pre-training, we hypothesise that the pre-trained features provide a stronger initialisation for this objective than for alternatives like diffusion- or flow-based models, where the shift to a denoising task creates a larger misalignment. Leveraging this compatibility, our approach (illustrated in Figure 1) modifies the deterministic architecture with a noise injection mechanism based on conditional layer normalisation (Peebles & Xie, 2023), enabling the model to represent stochasticity and generate ensembles without substantially altering the core backbone.

We validate this approach across a spectrum of pre-training regimes, ranging from single-system models to multi-system foundation models. This capability is particularly timely given that the field is currently undergoing a paradigm shift towards foundation models for dynamical systems, mirroring the trajectory of AI for weather prediction (Nguyen et al., 2023; Bodnar et al., 2025) or computational fluid dynamics (Ashton et al., 2025). As models scale rapidly in parameter count and training data to achieve generalisation, training high-fidelity probabilistic foundation models from scratch is becoming economically and environmentally unsustainable. By enabling the effective repurposing of the robust *deterministic* backbones that the community has already invested significant resources into, our method offers a viable alternative to the prohibitive cost of retraining. We demonstrate that this strategy is not only computationally efficient but also empirically effective, significantly outperforming continued deterministic fine-tuning. Our contributions are:

- **A Unified, Architecture-Agnostic Recipe**: Adapting a successful strategy from weather forecasting, we demonstrate its transferability to diverse PDE dynamics. We show that the same "recipe" proves effective across three distinct architectures (pixel- and latent-space) and five physical systems, requiring minimal modifications to the deterministic backbone.
- **Efficiency and Performance**: We demonstrate that our method achieves substantial gains over deterministic fine-tuning. Beyond the expected improvements in probabilistic scoring (reductions of up to $20 - 54\%$ in rollout CRPS), we additionally show reductions of up to $30\%$ in variance-normalised root mean square error (VRMSE). Crucially, we show that this strategy is effective even for foundation models trained on multiple systems, where, when performing retrofitting on the dataset of interest, it yields up to $40\%$ CRPS and $15\%$ VRMSE improvements over continued single-system deterministic training.
- **Inference-Time Flexibility**: We show that our retrofitted models exhibit test-time scaling properties, where predictive skill monotonically improves with the number of ensemble members generated, allowing users to trade off

compute for accuracy dynamically without retraining.
- **Ablations and Insights**: We provide extensive ablations on the learning dynamics of CRPS retrofitting, offering practical recommendations for hyperparameter tuning to ensure stable probabilistic adaptation.

## 2. Background

**Problem Formulation** We focus on modelling the evolution of continuous-time dynamical systems governed by PDEs. Let the state of the system at time $t$ be denoted by $\mathbf{u}(t) \in \mathcal{X}$, where $\mathcal{X}$ represents a function space defined over a spatial domain $\Omega \subseteq \mathbb{R}^D$. In practice, we do not observe the system continuously; instead, we have access to a dataset of trajectory snapshots generated by numerical solvers, discretised both in space (we assume a regular grid) and in time (with a fixed timestep $\Delta t$). We denote the discretised state vector at time step $t$ as $\mathbf{x}_t \in \mathbb{R}^d$. The objective is to learn an autoregressive model $\mathcal{M}_\theta$ that approximates the evolution operator of the system. Given a history of $k$ previous states $\mathbf{h}_t = (\mathbf{x}_{t-(k-1)}, \ldots, \mathbf{x}_t)$, the task is to predict the state at the next time step: $\mathbf{x}_{t+1} \approx \mathcal{M}_\theta(\mathbf{h}_t)$. Commonly, to improve stability and leverage the smoothness of physical dynamics, models are parameterised to predict the temporal residual (time derivative approximation) rather than the absolute state: $\Delta\hat{\mathbf{x}}_{t+1} = \hat{\mathbf{x}}_{t+1} - \mathbf{x}_t = \mathcal{M}_\theta(\mathbf{h}_t)$.

**Deterministic Modelling** In the deterministic setting, the model $\mathcal{M}_\theta$ outputs a single point estimate for the future state (or the residual). The parameters $\theta$ are optimised by minimising a distance metric between the predicted state and the ground truth, typically the MSE or MAE. While effective for stable, fully observed systems, and in short-term forecasting, this approach fails to capture the inherent uncertainty of chaotic systems, often leading to blurry predictions at longer horizons as the model converges to the mean of the possible future states (Price et al., 2025; Couairon et al., 2024).

**Probabilistic Modelling via CRPS** To capture the stochastic nature of the dynamics and quantify uncertainty, we transition from point estimates to probabilistic predictions. We optimise the Continuous Ranked Probability Score (CRPS) (Gneiting & Raftery, 2007), a strictly proper scoring rule applied to the marginal distributions of the output. Unlike likelihood-based approaches that require explicit density estimation, the CRPS can be computed directly from samples using its kernel representation (Baringhaus & Franz, 2004; Waghmare & Ziegel, 2025). For an observation $y$ and an ensemble of $M$ generated predictions $\{x_j\}_{j=1}^M$, the empirical CRPS is given by CRPS$(\{x_j\}_{j=1}^M, y) = \frac{1}{M}\sum_{j=1}^M |x_j - y| - \frac{1}{2M^2}\sum_{j=1}^M\sum_{k=1}^M |x_j - x_k|$. The first term encourages individual ensemble members to be accurate (minimising absolute error), while the second term

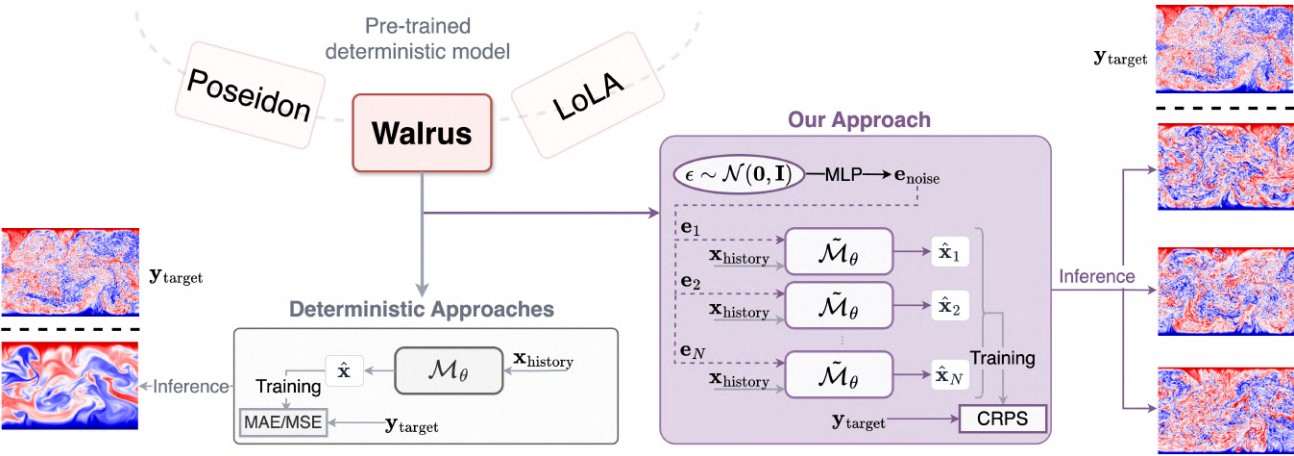

*Figure 1.* Comparison between deterministic and probabilistic retrofitting. **Left (Deterministic)**: Existing models ($\mathcal{M}_\theta$) map input history $\mathbf{x}_{\text{history}}$ to a single output using MAE/MSE loss, often yielding smoothed, average predictions. **Right (Our Approach)**: We introduce a stochastic branch where noise $\epsilon$ is projected via an MLP to modulate the backbone $\tilde{\mathcal{M}}_\theta$. The model is retrofitted using the CRPS loss. At inference, sampling multiple $\epsilon$ vectors generates an ensemble of sharp, diverse predictions for a single input history.

encourages diversity (dispersing ensemble members), preventing mode collapse. However, the standard empirical estimator is known to be biased for finite $M$, so we employ the unbiased CRPS estimator ('fair' CRPS) following (Alet et al., 2025):

$$
\begin{aligned}
\text{fCRPS}(\{x_j\}_{j=1}^M, y) = \frac{1}{M}\sum_{j=1}^{M}|x_j - y| \quad (1)\\
- \frac{1}{2M(M-1)}\sum_{j=1}^{M}\sum_{k=1}^{M}|x_j - x_k|.
\end{aligned}
$$

We apply this loss independently to each component of the state vector $\mathbf{x}_t$ and average the results.

## 3. Related Work

**Deterministic PDE Surrogates and Foundation Models**   While early data-driven approaches focused on modelling specific physical systems using bespoke architectures (Gupta & Brandstetter, 2022; Brandstetter et al., 2023; Kovachki et al., 2023; McCabe et al., 2024; Morel et al., 2025), the field has recently converged towards general-purpose foundation models with large-scale models such as Walrus (1.3B) (McCabe et al., 2025), Poseidon (628M) (Herde et al., 2024), DPOT (1.2B) (Hao et al., 2024), and PhysicsX (4.5B) (Nguyen et al., 2025). However, all these approaches share a common limitation: the majority are optimised with a deterministic loss, minimising reconstruction error. Consequently, they fail to capture the intrinsic stochasticity of complex dynamical systems. Given the significant cost of pre-training, discarding these high-capacity backbones is unsustainable. Instead, we should leverage these learned representations to enable probabilis-

tic forecasting without the burden of training from scratch.

**Probabilistic PDE Modelling**   Parallel to deterministic advances, there is growing interest in probabilistic approaches, particularly for chaotic systems where capturing the full distribution of possible future scenarios is critical. To date, PDE probabilistic modelling has largely been explored in single-system settings (Kohl et al., 2024; Lippe et al., 2024; Rozet & Louppe, 2023; Shysheya et al., 2024; Cachay et al., 2023; Rozet et al., 2025). Diffusion models have emerged as the dominant technique, demonstrating superior performance over deterministic baselines in both pixel space (Lippe et al., 2024; Shysheya et al., 2024) and latent space (Rozet et al., 2025). However, such approaches present significant challenges for the foundation model era: they are computationally intensive to train, slow at inference, and lack a straightforward mechanism for retrofitting deterministic backbones. Unlike CRPS-based retrofitting, which aligns naturally with deterministic objectives, converting a deterministic model into a diffusion model typically requires learning a fundamentally different denoising task.

Recently, the domain of weather forecasting has demonstrated that large-scale probabilistic training is possible with diffusion or flow-based models such as GenCast (Price et al., 2025) and ArchesWeather (Couairon et al., 2024). However, these achievements necessitated training massive generative models from scratch, incurring substantial computational costs. We argue that this cost would be even more prohibitive for general PDE foundation models. Unlike weather forecasting, which operates on a fixed spherical geometry and a single set of atmospheric equations, general PDE models must generalise across varying geometries, diverse boundary conditions, and distinct physical coef-

ficients. This presents a dilemma for practitioners: one often has access to a high-quality deterministic checkpoint—either a specialised single-system model or a multi-system foundation model—but lacks the resources to train a probabilistic diffusion model from scratch. This motivates our central question: *Can we leverage the representations of an existing deterministic checkpoint to achieve probabilistic capabilities with little additional cost?* We propose a training-efficient retrofitting procedure, applicable to both single-system and foundation models.

**Scoring Rules and CRPS**  Our approach draws inspiration from weather forecasting, where multiple works propose optimising proper scoring rules. The Continuous Ranked Probability Score (CRPS) is widely used; AIFS-CRPS (Lang et al., 2024b) was among the first to propose training with a modified CRPS loss, followed by FGN (Alet et al., 2025) which is currently the state-of-the-art in weather modelling, and FourCastNet3 (Bonev et al., 2025). Other works, such as Swift (Stock et al., 2025), combine consistency models (Song et al., 2023) with CRPS retrofitting.

While CRPS is powerful, it only evaluates univariate marginals and, thus, does not guarantee spatial consistency. To address this, some weather models incorporate spectral loss terms (Nordhagen et al., 2025; Bonev et al., 2025). In the context of general PDE modelling, Bülte et al. (2025) utilise the Energy Score (a multivariate generalisation of CRPS). However, their method introduces stochasticity via either stochastic dropout in the Fourier domain or by restricting predictions to closed-form Gaussian distributions, which limits the expressivity of the predicted distribution.

In contrast to these works, we propose a general framework to inject stochasticity into deterministic architectures via learnable noise modulation. Our method avoids restrictive Gaussian assumptions and allows for the efficient *conversion* of a deterministic backbone—whether a single-system model or a foundation model—into a probabilistic emulator, significantly improving predictive skill with relatively low computational overhead.

## 4. Methodology: Retrofitting a deterministic model into a probabilistic one

In this work we follow Alet et al. (2025) and adopt a *general* strategy to retrofit a deterministic model into a probabilistic one. This strategy requires two key considerations: (1) the noise injection strategy to generate an ensemble, and (2) the probabilistic loss used during retrofitting which is, as outlined in Section 2, the fair CRPS (Equation 1).

### 4.1. Noise Injection

We instantiate a global noise vector $\epsilon \sim \mathcal{N}(\mathbf{0}, \mathbf{I}_{d_{\text{noise}}})$ and pass it through a Multi-Layer Perceptron (MLP) to obtain noise embeddings (one set per ensemble member). We posit, following Alet et al. (2025), that injecting a low-dimensional global noise source—mapped to modulation parameters shared across all spatial dimensions—encourages the generation of spatially coherent fields, even when the model is trained on a univariate marginal loss. These embeddings are then fed into the architecture's conditional normalisation layers (AdaLN) (Peebles & Xie, 2023) to modulate the hidden representations. If the architecture has an encoder-processor-decoder structure, we inject it into all processor layers, whereas if it is an encoder-decoder architecture we restrict the noise injection to the encoder layers. The modulation mechanism, also described in Algorithm 1, proceeds as follows:

1. The noise embeddings are projected to obtain a set of modulating parameters via conditional normalisation.
2. Two of these parameters, $\gamma$ and $\beta$, modulate the normalised inputs via scaling and shifting. We apply this post-normalisation to ensure architectural stability.[2]
3. The modulated input is passed into the main feature transformation (e.g., attention in transformer architectures).
4. The output of this transformation is gated by the parameter $\alpha$ before being added to the residual connection.
5. Optionally, a fourth parameter, $\delta$, modulates the integration of skip connections.

These architectural modifications are complemented by specific learning dynamics designed to efficiently leverage the deterministic checkpoints. Key strategies include: 1) initialising the backbone parameters from the pre-trained deterministic model; 2) employing differential learning rates, with smaller values for the backbone to preserve learned features and larger values for the newly-introduced noise branch; and 3) carefully initialising the conditional normalisation layers to small values. This ensures the model starts close to its deterministic behavior, allowing it to smoothly learn how to integrate stochasticity during retrofitting.

### 4.2. Single-system Models

We first apply our approach to single-system models—architectures trained and tested on specific PDE dynamics, as opposed to multi-physics foundation models. To demonstrate generality, we evaluate our method across three distinct architectures representing diverse modelling paradigms. We summarise these models in Table 1 and provide full specifications in Appendix B.2.

---

[2]For architectures that perform the transformation (i.e., attention) prior to normalisation (post-norm), an additional normalisation layer is introduced before the transformation.

**Algorithm 1** Stochastic Modulation via Conditional Layer Normalisation

---

**Require:** Input $\mathbf{x} \in \mathbb{R}^d$, Skip connection $\mathbf{x}_{\text{skip}}$ (optional)
Dimensions $d_{\text{noise}}$, $d_{\text{emb}}$, $d_{\text{hidden}}$; Ensemble members $M$
**Initialisation:**
1: Noise projection: MLP : $\mathbb{R}^{d_{\text{noise}}} \to \mathbb{R}^{d_{\text{emb}}}$
2: Modulation head AdaLN : Linear$(d_{\text{emb}}, d_{\text{hidden}}) \to$ SiLU $\to$ Linear$(d_{\text{hidden}}, 4 \cdot d_{\text{hidden}})$
**Forward Pass:**
3: **1. Generate Noise Embeddings**
4: $\boldsymbol{\epsilon} \sim \mathcal{N}(\mathbf{0}, \mathbf{I}) \in \mathbb{R}^{M \times d_{\text{noise}}}$
5: $\mathbf{e}_{\boldsymbol{\epsilon}} \leftarrow \text{MLP}(\boldsymbol{\epsilon})$

6: **2. Obtain Modulation Parameters**
7: $(\boldsymbol{\gamma}, \boldsymbol{\beta}, \boldsymbol{\alpha}, [\boldsymbol{\delta}]) \leftarrow \text{AdaLN}(\mathbf{e}_{\boldsymbol{\epsilon}})$

8: **3. Pre-Transformation Modulation**
9: $\mathbf{x}_{\text{norm}} \leftarrow \text{Norm}(\mathbf{x})$         ▷ e.g., LayerNorm
10: $\tilde{\mathbf{x}} \leftarrow (1 + \boldsymbol{\gamma}) \odot \mathbf{x}_{\text{norm}} + \boldsymbol{\beta}$

11: **4. Main Transformation Operation**
12: $\mathbf{y} \leftarrow \text{Transform}(\tilde{\mathbf{x}})$         ▷ e.g., Self-Attention

13: **5. Residual & Skip Modulation**
14: $\mathbf{x}_{\text{out}} \leftarrow \mathbf{x} + (1 + \boldsymbol{\alpha}) \odot \mathbf{y}$
15: **if** using skip connections **then**
16: $\quad \mathbf{x}_{\text{out}} \leftarrow (\mathbf{x}_{\text{out}} + \boldsymbol{\delta} \odot \mathbf{x}_{\text{skip}})/\text{rsqrt}(1 + \boldsymbol{\delta}^2)$
17: **end if**
18: **return** $\mathbf{x}_{\text{out}}$

---

- The **Walrus** family (McCabe et al., 2025) : a transformer-based architecture developed primarily for fluid-like continuum dynamics, which forecasts the residual state at $t + 1$ ($\Delta\hat{\mathbf{x}}_{t+1}$) based on 6 previous time steps $\mathbf{x}_{t-5:t}$. We train a deterministic base model for each single-system with an MAE loss, following McCabe et al. (2025).
- The **Lola** family (Rozet et al., 2025): a latent-space architecture that jointly models 5 latent states at a time, and conditions on the PDE parameters $\zeta$ following the DiT conditioning approach (Peebles & Xie, 2023). At inference time, it jointly generates the next 4 states $\mathbf{x}_{t+1:t+4}$ conditioned on the current state $\mathbf{x}_t$. We use pre-trained checkpoints from Rozet et al. (2025) where available, and train an autoencoder and a deterministic neural solver with an MSE loss where checkpoints are not available. For the autoencoder, we use the configuration with 16 channels from Rozet et al. (2025).
- The **Poseidon** family (Herde et al., 2024): a multiscale operator transformer-based architecture that outputs $\mathbf{x}_{t+1}$ conditioned on the current state $\mathbf{x}_t$. We use the pre-trained `POSEIDON-L` checkpoint (where pre-training only includes one of the PDEs we test on), followed by fine-

*Table 1.* Base deterministic checkpoints in single-system models. $N_{\text{det}}$ denotes the number of samples used for training.

| Model | Representation | History | $\zeta$ cond. | $N_{\text{det}}$ | #Params |
|---|---|---|---|---|---|
| Walrus | Physical | 6 | ✗ | 3.2M | 277M |
| Lola | Latent | 1–4[3] | ✓ | 67.1M | 222M |
| Poseidon | Physical | 1 | ✗ | 6.4M | 629M |

tuning on each single-system to obtain the deterministic baseline. Given that Poseidon only operates on square states, we downsample all datasets to $128 \times 128$.

### 4.3. Foundation Model

We extend our validation to foundation models by pre-training a 640M-parameter multi-system backbone from the Walrus family (*HalfWalrus*) on $5.12$ million samples, adhering to the architecture and curriculum of McCabe et al. (2025). For each downstream system (within-distribution), we fine-tune/retrofit this checkpoint for an equivalent compute budget using either the original reconstruction loss (MAE) or our proposed CRPS objective. This allows us to assess whether probabilistic adaptation offers tangible gains over simply extending the deterministic training.

### 4.4. Datasets

Across all our experiments, we validate our method on five 2D physics simulations from The Well (Ohana et al., 2025), and respect the train/validation/test splits provided there. For the single-system experiments, we use Rayleigh-Bénard (RB), Euler Multi-Quadrants[4], and Shear Flow[5]. These are chosen to challenge the probabilistic modelling capabilities of the method with diverse uncertainty sources: chaotic thermal convection (RB), sharp shock waves that deterministic methods tend to blur (Euler), and mechanical instabilities (Shear Flow). For the multi-system foundation model, we evaluate on RB, Turbulent Radiative Layer (TRL), and Viscoelastic Instability (VI). In this case, we picked a subset of the datasets included in the foundation model's pre-training corpus, as specified in McCabe et al. (2025). TRL introduces high sensitivity to initial random noise seeds, while VI presents a strictly multimodal challenge where distinct attractors (two chaotic, one steady, and transitions) coexist for identical parameters; accordingly, we evaluate the two chaotic regimes and the steady/transition states separately. Full specifications are detailed in Appendix C.

---

[3]Lola is trained by modelling joint distributions over 5-step trajectories with randomised conditioning. This allows for flexible inference, allowing to condition on any history length between 1 and 4 steps at query time.

[4]For Walrus and Poseidon we only use the subset with periodic boundary conditions.

[5]We evaluate up to 100 time steps.

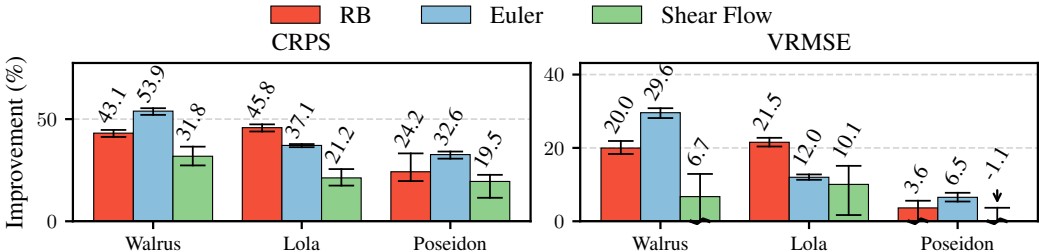

*Figure 2.* Improvement of CRPS retrofitting over deterministic fine-tuning for the single-system models: (Left) CRPS; (Right) VRMSE. The confidence intervals are obtained through 100 bootstrapping iterations and use a confidence level of 68%.

### 4.5. Evaluation Metrics

For chaotic systems, exact long-term prediction is fundamentally limited. Therefore, a robust surrogate must not only minimise trajectory error but also capture the system's statistical properties. To assess both capabilities, we evaluate the deterministic accuracy of the ensemble mean and the probabilistic fidelity of the full ensemble. For the former, following Ohana et al. (2025), we report the rollout variance-normalised root mean squared error (VRMSE), while for the latter we report the rollout fCRPS. For the probabilistic method, we use a 16-member ensemble; specifically, the VRMSE is computed between the ground truth and the ensemble mean. We use the following definition for the VRMSE: $\text{VRMSE}(u, v) = \sqrt{\frac{\langle (u-v)^2 \rangle}{\langle (u-\langle u \rangle)^2 \rangle + \epsilon}}$, where $\langle \cdot \rangle$ denotes the spatial mean operator and $\epsilon = 10^{-6}$ is added for numerical stability. In the appendix we also report the spread-skill ratio, as described in Appendix D.

For all rollout metrics, we compute the channel-wise and temporal averages for each trajectory. We report the aggregate median with 95% (tables) and 68% (plots) confidence intervals obtained via bootstrap resampling (100 iterations with replacement). Additional details about the evaluation procedure can be found in Appendix D.2.

### 5. Results

We aim to answer the following questions: 1. Does CRPS retrofitting lead to metric improvement in comparison to deterministic fine-tuning for all the datasets and models considered?, 2. What distinct qualitative behaviours emerge from CRPS retrofitting across different flow regimes and model architectures?, and 3. How do performance metrics vary with the ensemble size?. Finally, we perform ablations to shed light on the important hyperparameters that need careful tuning when using our approach. Results cover both single-system and the multi-system foundation model (training workflow illustrated in Figure 7), with computational costs summarised in Table 7. We note that Walrus, Lola, and Poseidon refer strictly to the architectural families; all model instances used here were trained independently (Section 4.2

and Appendix B.2).

### 5.1. Single-system models

**Performance compared to deterministic fine-tuning** As shown in Figure 2, our method leads to substantial improvements in rollout CRPS across all datasets and models (20–54%) compared to deterministic fine-tuning, evaluated over 200 steps for RB and 100 steps for Euler and Shear Flow. While a reduction in one-step CRPS is expected, translating this into long-horizon stability is non-trivial due to error accumulation in the autoregressive loop; yet, our 16-member ensemble effectively mitigates this drift. We also observe broad improvements in VRMSE, particularly for the models within the Walrus and Lola families, with reductions of up to 21.5% on RB, 29.6% on Euler, and 10.1% on Shear Flow. Crucially, these quantitative gains translate into striking qualitative differences. As illustrated in the RB rollout in Figure 3, the deterministic baseline from the Walrus family collapses into an over-smoothed prediction by $t = 65$. In contrast, the CRPS-tuned ensemble preserves the intricate, high-frequency turbulent structures inherent to the chaotic regime. This confirms that given sufficient model capacity, the CRPS objective not only improves calibration but actively prevents the smoothening typical of deterministic PDE modeling at long rollout times in uncertain regimes.

**Dataset Characteristics and Baseline Strength** While rollout CRPS gains remain robust ($> 20\%$) across all architectures and datasets, VRMSE improvements are more nuanced for Poseidon and for Walrus on Shear Flow, where confidence intervals often overlap with zero. We attribute this to the convergence state of the specific deterministic backbones. As shown in Table 9, the deterministic performance for all Poseidon models, as well as Walrus on Shear Flow, continues to improve with extended training. This contrasts with the RB and Euler regimes for Walrus and Lola, as well as Lola on Shear Flow, where the deterministic baselines effectively plateau. In cases where the backbone is still improving or the error floor is already low (e.g., Walrus on Shear Flow starts at VRMSE $\approx 0.07$), the margin for purely deterministic VRMSE gains shrinks. Crucially,

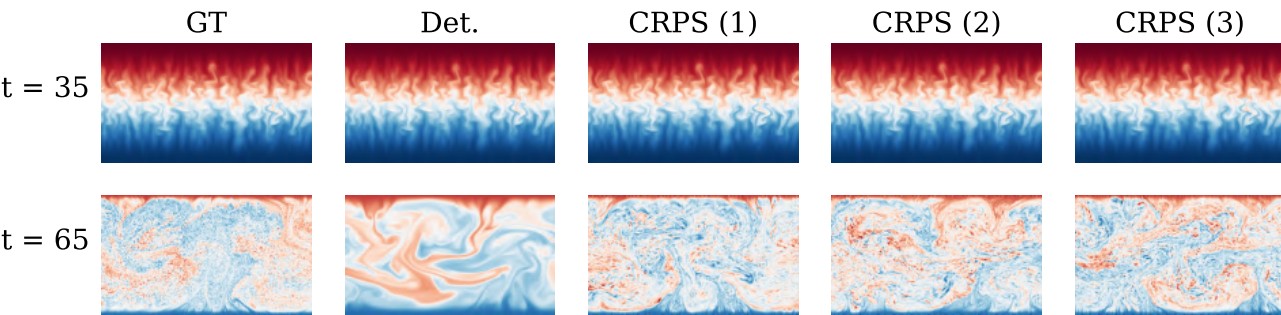

Figure 3. Comparison of Ground Truth (GT), Deterministic-tuned baseline, and three independent samples from the CRPS-tuned ensemble (Walrus architecture) on the RB dataset. While the deterministic model blurs significantly at later time steps due to uncertainty averaging, the CRPS samples maintain fine-scale details throughout the rollout.

however, our method still successfully lowers the rollout CRPS in these scenarios, demonstrating that probabilistic retrofitting adds value even when the deterministic backbone is already highly competitive or under-converged.

**Architecture-Specific Constraints** Beyond dataset characteristics, the observed performance is also shaped by distinct architectural limitations. During both training and testing, Poseidon always conditions on a history of one time step, which limits its ability to capture highly complex systems compared to models with longer memory (Ruiz et al., 2025). As shown in Figure 12 (bottom), this forces the deterministic baseline into severe over-smoothing; in contrast, the CRPS ensemble preserves physical structure but compensates for the temporal ambiguity through over-dispersion. While this behaviour penalises VRMSE, it is correctly captured by the CRPS metric. Conversely, Lola is bounded by its projection into a compressed latent space, potentially bottlenecking the resolution of high-frequency features. Remarkably, despite this compression, Lola achieves consistent improvements in both VRMSE and CRPS across all datasets. This confirms that probabilistic retrofitting remains effective even within the constraints of latent-space models—a paradigm particularly attractive for its reduced training requirements.

Overall, we observe that CRPS retrofitting is particularly effective when applied to (near-)converged deterministic models, offering a way to break through performance plateaus. While it universally improves probabilistic calibration, its ability to surpass deterministic baselines in VRMSE is strongest when the underlying deterministic training has already been maximised. Full performance metrics are detailed in Table 9. For a deeper investigation, Appendix E.1 provides more detailed results (including rollout energy score metrics, showing that univariate gains translate into multivariate ones too), expanded visualisations, and a discussion of the structural changes induced by the CRPS objective.

**Improvement with additional ensemble members** An additional benefit of the proposed approach is the flexibility to balance predictive performance against computational cost at inference time. Figure 4 quantifies the gain from probabilistic aggregation. By reporting the VRMSE relative to a single ensemble member, we highlight a consistent trend: increasing the ensemble size lowers the rollout errors across all datasets in Walrus. The same holds for the other models, as shown in Figure 19. The extent of these gains are model- and dataset-dependent, but they offer a controllable knob between test-time performance and compute budget.

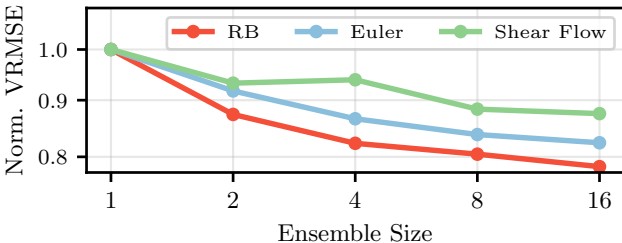

Figure 4. Error scaling. Log-log plot of rollout VRMSE (normalised by the single-member baseline) vs. ensemble size ($M$) for the Walrus model.

### 5.2. Foundation Model - HalfWalrus

We extend our empirical investigation to a foundation model setting. The procedure consists of two stages: 1) Pretraining—we train a Walrus-based backbone (HalfWalrus) on a diverse mixture of PDEs; followed by 2) Adaptation—we then specialise this model for individual downstream datasets (drawn from the pre-training distribution) using either CRPS retrofitting, or deterministic fine-tuning.

**Performance versus deterministic fine-tuning** As illustrated in Figure 5, retrofitting with the CRPS objective yields substantial improvements over the deterministic baseline for the foundation model. We observe gains of 34–40% in CRPS and $> 13\%$ in VRMSE for both RB and TRL,

demonstrating that probabilistic retrofitting improves both calibration and accuracy. For the viscoelastic dataset, we analyse performance across three distinct regimes: two chaotic phases—Elasto-Inertial Turbulence (EIT) and the Chaotic Arrowhead Regime (CAR)—as well as the non-chaotic Steady Arrowhead Regime (SAR) and transition phases. In the chaotic phases (EIT and CAR), the CRPS model significantly outperforms the deterministic baseline on both metrics. In the transitional regime, the CRPS model only improves the rollout CRPS and degrades the rollout VRMSE. We note, however, that the SAR and transitional regimes are limited to short horizons ($T = 20$) compared to the chaotic regimes ($T = 60$); this shorter window likely constrains the accumulation of error, potentially masking the long-term benefits of the probabilistic approach.

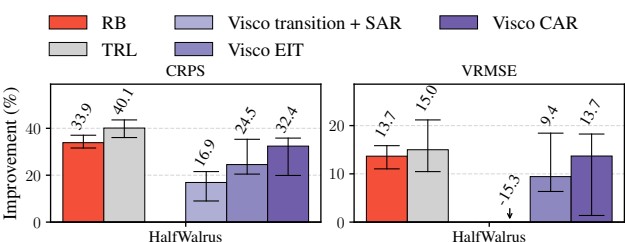

*Figure 5.* Improvement of CRPS retrofitting over deterministic fine-tuning for HalfWalrus: (Left) CRPS; (Right) VRMSE.

Qualitative analysis of rollouts (Appendix E.2) demonstrates that CRPS retrofitting significantly improves physical fidelity in chaotic regimes. Across RB, TRL, and viscoelastic turbulence (EIT/CAR), the deterministic baseline frequently suffers from over-smoothing and temporal misalignment—most notably the unphysical flow acceleration observed in RB. In contrast, CRPS-tuned models preserve sharp, high-frequency structures and accurate temporal evolution. Crucially, in stochastic limits where the deterministic prediction collapses to a blurred mean (e.g., TRL at $t > 60$), the CRPS ensemble maintains better physical plausibility by capturing the variance of valid outcomes. In stable regimes (SAR and transition), both methods visually perform equally well.

**Improvement with additional ensemble members** Similarly to the single-system models, the rollout VRMSE improves with the number of ensemble members (Figure 6).

### 5.3. Ablation Studies

We conduct ablation studies in Appendix E.1.4 (single-system models) and Appendix E.2.4 (foundation models) to assess sensitivity to key hyperparameters, including learning rate, retrofitting duration, training ensemble size, noise embedding dimensionality, and noise injection density. Our results highlight the overall robustness of the method.

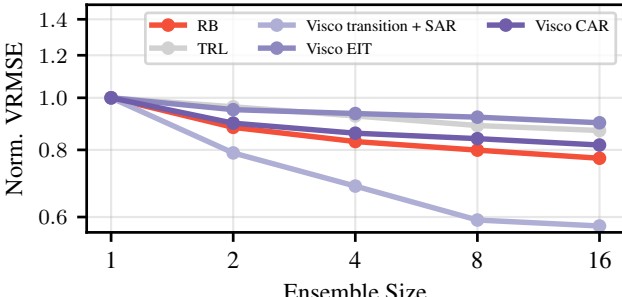

*Figure 6.* Error scaling. Log-log plot of rollout VRMSE (normalised by the single-member baseline) vs. ensemble size ($M$) for the HalfWalrus model.

**Hyperparameter Stability:** Performance is generally robust to learning rate choices across both settings, provided the values are selected within the appropriate regime for the specific architecture. Similarly, reducing the size of the training ensemble yields minimal performance degradation, indicating that the method remains effective even under tighter computational constraints.

**Architectural Capacity:** While higher noise embedding dimensions and dense noise injection (in all layers) generally yield the most consistent results—particularly on complex datasets—sparser configurations (lower dimensionality or half-layer injection) offer competitive, parameter-efficient alternatives with only marginal, system-dependent reductions in performance.

**Training Duration:** Extending retrofitting generally yields diminishing returns. We anticipate that future gains in long-horizon stability are more likely to come from dedicated rollout fine-tuning rather than prolonged retrofitting.

**Comparison to alternative probabilistic baselines:** Finally, we compare the CRPS retrofitting procedure with two alternative probabilistic methods. Appendix E.3.1 shows a comparison to deep ensembles (fine-tuning the original deterministic checkpoint four separate times with different initialisations), confirming the performance gains are largely driven by the CRPS objective itself, not by just the presence of an ensemble. In addition, Appendix E.3.2 compares our method against diffusion-based retrofitting in Lola, showing that CRPS retrofitting leverages the pre-trained deterministic representations more efficiently than diffusion alternatives.

## 6. Conclusion

This work presents a unified strategy for retrofitting deterministic backbones into probabilistic emulators. By adapting a noise injection mechanism and optimising the CRPS, our approach proves effective across diverse modelling paradigms and dynamical systems—it yields consistent metric improvements with minimal code modifications and re-

duced computational overhead compared to training from scratch. However, our analysis reveals that the retrofitted model's behaviour is dependent on the inductive biases of the underlying backbone. We observed that architectural constraints—such as limited temporal dependencies (Poseidon) or compressed latent representations (Lola)—can bias the ensemble towards over- or under-dispersion, respectively. Furthermore, because our method relies on the backbone's learned features, it can inherit and occasionally amplify existing instabilities, such as artifacts arising from error accumulation in long rollouts. Addressing these limitations offers exciting future research avenues. Potential solutions include incorporating multi-step rollout fine-tuning (Lang et al., 2024a; Alet et al., 2025) or employing auxiliary spectral losses (such as spectral CRPS (Nordhagen et al., 2025)) to enforce physical consistency. Ultimately, this framework provides a practical path to leverage the resources invested in deterministic pre-training, unlocking the benefits of probabilistic modelling without discarding existing checkpoints.

## Acknowledgements

We would like to acknowledge the support of the Simons Foundation and of Schmidt Sciences. This work was supported in part by the AI2050 program at Schmidt Sciences (Grant G-25-70028). Payel Mukhopadhyay thanks the Infosys-Cambridge AI centre for support. The compute for this project involved multiple sources, and the authors would like to thank the Scientific Computing Core, a division of the Flatiron Institute, a division of the Simons Foundation for extensive computational support. Additionally, Cristiana Diaconu is supported by the Cambridge Trust Scholarship. Miles Cranmer is grateful for support from the Schmidt Sciences AI2050 Early Career Fellowship and the Isaac Newton Trust. Richard E. Turner is supported by Google, Amazon, ARM, Improbable and the EPSRC Probabilistic AI Hub (EP/Y028783/1). We also thank Michael McCabe for useful comments and support with the Walrus codebase, François Rozet for help with the Lola codebase, Geraud Krawezik for help with compute resources, and the Polymathic AI team for valuable discussions and feedback.

## Impact Statement

This paper presents work whose goal is to advance the field of machine learning. There are many potential societal consequences of our work, none of which we feel must be specifically highlighted here.

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

# A. Implementation Details

This section provides implementation details for the Walrus, Lola, and Poseidon model families. We focus explicitly on the modifications made to the backbone architectures to integrate the CRPS retrofitting procedure. For the foundational model specifications, we refer the reader to the original publications: Walrus (McCabe et al., 2025), Lola (Rozet et al., 2025), and Poseidon (Herde et al., 2024). While all implementations adhere to the general stochastic modulation strategy outlined in Algorithm 1, we provide the model-specific adaptations below for completeness.

**Noise embeddings** For all architectures, we begin by obtaining the noise embeddings $\mathbf{e}_\epsilon$. We sample a global noise vector $\boldsymbol{\epsilon} \sim \mathcal{N}(\mathbf{0}, \mathbf{I}_{d_{\text{noise}}})$ and transform it via a two-layer MLP with an expansion factor of 4. The embedding process is defined by the following composition of operations:

$$
\begin{aligned}
\mathbf{h} &= \text{SiLU}\left(\mathbf{W}_1\boldsymbol{\epsilon} + \mathbf{b}_1\right) & \triangleright \quad \text{Expand: } \mathbb{R}^{d_{\text{noise}}} \to \mathbb{R}^{4d_{\text{noise}}} \\
\mathbf{e}_\epsilon &= \text{LayerNorm}\left(\mathbf{W}_2\mathbf{h} + \mathbf{b}_2\right) & \triangleright \quad \text{Project: } \mathbb{R}^{4d_{\text{noise}}} \to \mathbb{R}^{d_{\text{noise}}}
\end{aligned}
\tag{2}
$$

where $\mathbf{W}_1 \in \mathbb{R}^{4d_{\text{noise}} \times d_{\text{noise}}}$, $\mathbf{W}_2 \in \mathbb{R}^{d_{\text{noise}} \times 4d_{\text{noise}}}$ are learnable weights.

The rationale for injecting a single global noise vector into all conditional normalisation layers follows the hypothesis proposed by Alet et al. (2025). They speculate that the combination of a low-dimensional noise source and global injection—where parameters are shared across the spatial dimensions—acts as an inductive bias. This setup effectively constrains the model to generate globally coherent spatial variability, even when the training objective (CRPS) only explicitly optimises univariate marginal distributions.

**Initialisation strategies** Another common architectural choice is the careful initialisation of the conditional normalisation layers. To ensure a smooth transition from the deterministic backbone, we carefully initialise the final linear projection of the modulation MLP. By scaling these weights by a factor of $10^{-2}$, we ensure that the modulation parameters $(\boldsymbol{\gamma}, \boldsymbol{\beta}, \boldsymbol{\alpha}, (\boldsymbol{\delta}))$ are initially close to zero. This effectively starts the training process with the noise injection turned off (acting as an approximate identity function), allowing the model to gradually learn to incorporate stochasticity without destroying the pre-trained representations.

## A.1. CRPS Retrofitting in Walrus

The Walrus architecture employs a Space-Time Split Transformer design, consisting of sequential axial time attention and spatial attention layers. We inject stochastic noise into every transformer block, targeting both the axial time attention module and the spatial attention module. The specific architectural modifications for the axial time attention block are detailed in Algorithm 2, while the changes to the spatial attention block are presented in Algorithm 3. Similarly to McCabe et al. (2025) we employ asymmetric normalisation for model inputs and outputs.

## A.2. CRPS Retrofitting in Lola

The architecture of the Lola model family is based on the Vision Transformer (ViT) framework (Dosovitskiy et al., 2020). We inject stochastic noise into every ViT block within the network. A distinguishing feature of Lola, compared to the other architectures in this study, is its existing conditioning on PDE parameters via conditional layer normalisation. We preserve this mechanism entirely and introduce an auxiliary AdaLN layer specifically for the noise injection. The modulation parameters (scale $\boldsymbol{\gamma}$, shift $\boldsymbol{\beta}$, gates $\boldsymbol{\alpha}, \boldsymbol{\delta}$) from the noise head are added element-wise to those from the PDE parameter head before being applied to the normalised tokens. We follow the normalisation procedure from the codebase associated with Rozet et al. (2025), using statistics computed based on the training dataset.

## A.3. CRPS Retrofitting in Poseidon

The Poseidon architecture is a hierarchical, multiscale vision transformer designed with explicit lead-time conditioning. While the model is conditioned on variable lead times during pre-training, we restrict it to next-step prediction during fine-tuning (both for the deterministic baseline and the CRPS retrofitting). The backbone consists of multiple SwinV2 transformer blocks (Liu et al., 2022) organised within a U-Net style encoder-decoder framework (Cao et al., 2021). We inject stochastic noise exclusively into the encoder blocks. Furthermore, because the original SwinV2 blocks employ a post-normalisation scheme (applying normalisation after the main transformation), we introduce an additional pre-normalisation

---

**Algorithm 2** Walrus Axial Time Attention Block with CRPS Retrofitting

---

**Require:** Batch size $B$, Ensemble members $M$, Input tensor $\mathbf{x} \in \mathbb{R}^{T \times (B \times M) \times C \times H \times W \times d_{\text{hidden}}}$, Noise embeddings $\mathbf{e}_\epsilon \in \mathbb{R}^{(B \times M) \times d_{\text{noise}}}$

**Initialisation:**

    Modulation head AdaLN : $\text{Linear}(d_{\text{emb}}, d_{\text{hidden}}) \rightarrow \text{SiLU} \rightarrow \text{Linear}(d_{\text{hidden}}, 3 \cdot d_{\text{hidden}})$

**Forward Pass:**

  $(\boldsymbol{\gamma}, \boldsymbol{\beta}, \boldsymbol{\alpha}) \leftarrow \text{AdaLN}(\mathbf{e}_\epsilon)$                                                           ▷ Obtain modulation parameters

  $\mathbf{x}_{\text{norm}} \leftarrow \text{RMSGroupNorm}(\mathbf{x})$

  $\tilde{\mathbf{x}} \leftarrow (1 + \boldsymbol{\gamma}) \odot \mathbf{x}_{\text{norm}} + \boldsymbol{\beta}$                                        ▷ Pre-transformation modulation

  # Main transformation operation (axial time attention with relative bias)

  $\mathbf{Q}, \mathbf{K}, \mathbf{V} \leftarrow \text{LinearProjections}(\tilde{\mathbf{x}})$

  $\mathbf{Q}, \mathbf{K} \leftarrow \text{LayerNorm}(\mathbf{Q}), \text{LayerNorm}(\mathbf{K})$

  $\mathbf{y} \leftarrow \text{Softmax}\left(\frac{\mathbf{Q}\mathbf{K}^\top}{\sqrt{d_{\text{hidden}}}} + \mathbf{B}_{\text{rel}}\right)\mathbf{V}$

  $\mathbf{y} \leftarrow \text{OutputHead}(\mathbf{y})$

  $\mathbf{x}_{\text{out}} \leftarrow \mathbf{x} + \text{DropPath}((1 + \boldsymbol{\alpha}) \odot \mathbf{y})$                              ▷ Residual and skip modulation

  **return** $\mathbf{x}_{\text{out}}$

---

layer before the transformation step to ensure training stability during the CRPS retrofitting procedure. The modified block structure is detailed in Algorithm 5. We apply the asymmetric normalisation strategy used for the Walrus family of models.

# B. Architecture and Training Details

We begin by presenting the architectural details for the models used, followed by the training protocol.

## B.1. Architectural Details

We generally followed the architectural design from the original papers, but provide full architectural details for each model for completeness. For the Walrus family (Table 2) we show the configurations for two models: a single-system one, and a multi-system one (HalfWalrus) that was original pre-trained on a multi-system dataset and then fine-tuned on single systems within that dataset. We give the details for the Lola and Poseidon models in Table 3.

---

**Algorithm 3** Walrus Axial Spatial Attention Block with CRPS Retrofitting

---

**Require:** Batch size $B$, Ensemble members $M$, Input tensor $\mathbf{x} \in \mathbb{R}^{T \times (B \times M) \times C \times H \times W \times d_{\text{hidden}}}$, Noise embeddings $\mathbf{e}_\epsilon \in \mathbb{R}^{(B \times M) \times d_{\text{noise}}}$

**Initialisation:**
    Modulation head AdaLN : Linear($d_{\text{emb}}, d_{\text{hidden}}$) $\to$ SiLU $\to$ Linear($d_{\text{hidden}}, 3 \cdot d_{\text{hidden}}$)

**Forward Pass:**
    $(\boldsymbol{\gamma}, \boldsymbol{\beta}, \boldsymbol{\alpha}) \leftarrow \text{AdaLN}(\mathbf{e}_\epsilon)$          ▷ Obtain modulation parameters
    $\mathbf{x}_{\text{norm}} \leftarrow \text{RMSGroupNorm}(\mathbf{x})$
    $\tilde{\mathbf{x}} \leftarrow (1 + \boldsymbol{\gamma}) \odot \mathbf{x}_{\text{norm}} + \boldsymbol{\beta}$          ▷ Pre-transformation modulation

    # Main transformation operation (fused spatial attention & FFN)
    $\mathbf{Q}, \mathbf{K}, \mathbf{V}, \mathbf{x}_{\text{ff}} \leftarrow \text{FusedProjection}(\tilde{\mathbf{x}})$
    $\mathbf{Q}, \mathbf{K} \leftarrow \text{LayerNorm}(\mathbf{Q}), \text{LayerNorm}(\mathbf{K})$
    Apply rotary positional embeddings to $\mathbf{Q}, \mathbf{K}$
    $\mathbf{y}_{\text{att}} \leftarrow \text{OutputHead}\left(\text{Softmax}\left(\frac{\mathbf{Q}\mathbf{K}^\top}{\sqrt{d_{\text{hidden}}}}\right)\mathbf{V}\right)$
    $\mathbf{y}_{\text{ff}} \leftarrow \text{Linear}(\text{SwiGLU}(\mathbf{x}_{\text{ff}}))$
    $\mathbf{y} \leftarrow \mathbf{y}_{\text{att}} + \mathbf{y}_{\text{ff}}$
    $\mathbf{x}_{\text{out}} \leftarrow \mathbf{x} + \text{DropPath}((1 + \boldsymbol{\alpha}) \odot \mathbf{y})$          ▷ Residual and skip modulation
    **return** $\mathbf{x}_{\text{out}}$

---

---

**Algorithm 4** Lola ViT Block with CRPS Retrofitting

---

**Require:** Batch size $B$, Ensemble members $M$, Input tokens $\mathbf{x} \in \mathbb{R}^{(B \times M) \times L \times C}$, PDE Parameters Modulation $\mathbf{z} \in \mathbb{R}^{(B \times M) \times d_{\text{mod}}}$, Coordinates $\mathbf{p} \in \mathbb{R}^{(B \times M) \times L \times N}$, Skip $\mathbf{x}_{\text{skip}}$ (optional), Noise embedding $\mathbf{e}_\epsilon \in \mathbb{R}^{(B \times M) \times d_{\text{noise}}}$

**Initialisation:**
    PDE Parameter Modulation Head AdaLN : Linear($d_{\text{emb}}, d_{\text{emb}}$) $\to$ SiLU $\to$ Linear($d_{\text{emb}}, 4 \cdot C$)
    Noise Modulation Head AdaLN$_\epsilon$ : Linear($d_{\text{emb}}, d_{\text{emb}}$) $\to$ SiLU $\to$ Linear($d_{\text{emb}}, 4 \cdot C$)

**Forward Pass:**
    $(\boldsymbol{\gamma}, \boldsymbol{\beta}, \boldsymbol{\alpha}, \boldsymbol{\delta}) \leftarrow \text{AdaLN}(\mathbf{z})$          ▷ Standard modulation parameters
    $(\boldsymbol{\gamma}_\epsilon, \boldsymbol{\beta}_\epsilon, \boldsymbol{\alpha}_\epsilon, \boldsymbol{\delta}_\epsilon) \leftarrow \text{AdaLN}_\epsilon(\mathbf{e}_\epsilon)$          ▷ Obtain noise modulation parameters
    $\boldsymbol{\gamma} \leftarrow \boldsymbol{\gamma} + \boldsymbol{\gamma}_\epsilon; \quad \boldsymbol{\beta} \leftarrow \boldsymbol{\beta} + \boldsymbol{\beta}_\epsilon; \quad \boldsymbol{\alpha} \leftarrow \boldsymbol{\alpha} + \boldsymbol{\alpha}_\epsilon; \quad \boldsymbol{\delta} \leftarrow \boldsymbol{\delta} + \boldsymbol{\delta}_\epsilon$

    $\mathbf{x}_{\text{norm}} \leftarrow \text{LayerNorm}(\mathbf{x})$
    $\tilde{\mathbf{x}} \leftarrow (1 + \boldsymbol{\gamma}) \odot \mathbf{x}_{\text{norm}} + \boldsymbol{\beta}$

    # Main transformation operation (attention & FFN)
    $\mathbf{Q}, \mathbf{K}, \mathbf{V} \leftarrow \text{LinearProjections}(\tilde{\mathbf{x}})$
    $\mathbf{Q}, \mathbf{K} \leftarrow \text{RMSNorm}(\mathbf{Q}), \text{RMSNorm}(\mathbf{K})$
    Calculate rotary frequencies $\boldsymbol{\Theta}$ from coordinates $\mathbf{p}$
    Apply rotary positional embeddings $\boldsymbol{\Theta}$ to $\mathbf{Q}, \mathbf{K}$
    $\mathbf{y}_{\text{att}} \leftarrow \text{Softmax}\left(\frac{\mathbf{Q}\mathbf{K}^\top}{\sqrt{d_{\text{hidden}}}}\right)\mathbf{V}$
    $\mathbf{y}_{\text{mid}} \leftarrow \tilde{\mathbf{x}} + \text{OutputHead}(\mathbf{y}_{\text{att}})$
    $\mathbf{y} \leftarrow \text{FFN}(\mathbf{y}_{\text{mid}})$

    $\mathbf{x} \leftarrow (\mathbf{x} + \boldsymbol{\alpha} \odot \mathbf{y}) \odot \text{rsqrt}(1 + \boldsymbol{\alpha}^2)$          ▷ Gated residual connection
    **if** skip connection exists **then**
        $\mathbf{x} \leftarrow (\mathbf{x} + \boldsymbol{\delta} \odot \mathbf{x}_{\text{skip}}) \odot \text{rsqrt}(1 + \boldsymbol{\delta}^2)$
    **end if**
    **return** $\mathbf{x}$

---

---

**Algorithm 5** Poseidon ScOT Layer with CRPS Retrofitting

---

**Require:** Batch size $B$, Ensemble members $M$, Input tensor $\mathbf{x} \in \mathbb{R}^{(B \times M) \times L \times C}$, Time embedding $\mathbf{e}_t$, Noise embedding
  $\mathbf{e}_\epsilon \in \mathbb{R}^{(B \times M) \times d_{\text{noise}}}$
**Initialisation:**
  Modulation head AdaLN : Linear$(d_{\text{emb}}, C) \to$ SiLU $\to$ Linear$(C, 3 \cdot C)$
**Forward Pass:**
  $(\boldsymbol{\gamma}, \boldsymbol{\beta}, \boldsymbol{\alpha}) \leftarrow \text{AdaLN}(\mathbf{e}_\epsilon)$                         ▷ Obtain modulation parameters

  $\tilde{\mathbf{x}} \leftarrow \text{LayerNorm}(\mathbf{x})$                         ▷ Pre-attention normalisation
  $\tilde{\mathbf{x}} \leftarrow (1 + \boldsymbol{\gamma}) \odot \tilde{\mathbf{x}} + \boldsymbol{\beta}$                         ▷ Pre-attention modulation

  # Main transformation operation (Swin attention)
  **if** shift_size $> 0$ **then** Cyclic Shift $\tilde{\mathbf{x}}$ **end if**
  $\mathbf{y} \leftarrow \text{WindowPartition}(\tilde{\mathbf{x}})$
  $\mathbf{y} \leftarrow \text{SwinV2Attention}(\mathbf{y})$
  $\mathbf{y} \leftarrow \text{WindowReverse}(\mathbf{y})$
  **if** shift_size $> 0$ **then** Reverse Cyclic Shift $\mathbf{y}$ **end if**
  $\mathbf{y} \leftarrow \text{ConditionalLayerNorm}(\mathbf{y}, \mathbf{e}_t)$                         ▷ Time conditioning (Post-Norm)
  $\mathbf{x} \leftarrow \mathbf{x} + \text{DropPath}(\mathbf{y})$

  $\mathbf{v}_{\text{mid}} \leftarrow \text{IntermediateProjection}(\mathbf{x})$
  $\mathbf{v} \leftarrow \text{OutputProjection}(\mathbf{v}_{\text{mid}})$
  $\mathbf{v} \leftarrow \text{ConditionalLayerNorm}(\mathbf{v}, \mathbf{e}_t)$
  $\mathbf{v} \leftarrow \mathbf{v} \odot (1 + \boldsymbol{\alpha})$                         ▷ Post-FFN modulation gate
  $\mathbf{x}_{\text{out}} \leftarrow \mathbf{x} + \text{DropPath}(\mathbf{v})$
  **return** $\mathbf{x}_{\text{out}}$

---

*Table 2.* Hyperparameters used for the models within the Walrus family of models: single-system and multi-system (HalfWalrus).

| | Single-system | Multi-system |
|---|---|---|
| | **Walrus** | **HalfWalrus** |
| Architecture | Space-time Split Transformer | Space-time Split Transformer |
| Backbone Parameters | $2.77 \times 10^8$ | $6.42 \times 10^8$ |
| Noise Branch Parameters | $0.57 \times 10^8$ | $2.15 \times 10^8$ |
| Noise Embedding dimension | 32 | 32 |
| Time History (2D) | 6 | 6 |
| Base Token Shape (2D) | $32^2$ | $32^2$ |
| Projection dimension | 48 | 48 |
| Encoder dimension | 352 | 352 |
| Hidden dimension | 1088 | 1088 |
| MLP dimension | 4352 | 4352 |
| Space block | Parallel Attention | Parallel Attention |
| Space positional embedding | AxialRoPE | AxialRoPE |
| Time block | Causal Attention | Causal Attention |
| Time positional embedding | LearnedRPE | LearnedRPE |
| Attention heads | 16 | 16 |
| Activation | SwiGLU | SwiGLU |
| Normalization | RMSGroupNorm | RMSGroupNorm |
| Normalization Groups | 16 | 16 |
| Blocks | 8 | 30 |
| Drop Path | 0.05 | 0.05 |

*Table 3.* Hyperparameters for the Lola and Poseidon model families. For Lola, the patch size is $1 \times 1 \times 1$ as the backbone operates in a latent space downsampled by a factor of 32. We use $C_{\text{latent}} = 16$ and a history of $n = 4$ states. For Poseidon, the architecture follows a hierarchical U-Net structure with SwinV2 blocks. We use the configuration corresponding to POSEIDON-L. Because Poseidon operates on square states, we downsample the datasets to $128 \times 128$ before inputting them into the model. $H$ and $W$ denote the height and width of the PDE state.

**Lola Configuration**

| Parameter | Value |
|---|---|
| Architecture | ViT Transformer |
| Backbone parameters | $2.22 \times 10^8$ |
| Noise branch params | $0.18 \times 10^8$ |
| Noise embed. dim. | 32 |
| Input shape | $C_{\text{latent}} \times (n-1) \times \frac{H}{32} \times \frac{W}{32}$ |
| Patch size | $1 \times 1 \times 1$ |
| Tokens | $(n+1) \times \frac{H}{32} \times \frac{W}{32}$ |
| Embedding size | 1024 |
| Blocks | 16 |
| Positional embedding | Absolute + RoPE |
| Activation | SiLU |
| Normalisation | LayerNorm |
| Drop Path | 0.05 |

**Poseidon Configuration**

| Parameter | Value |
|---|---|
| Architecture | ScOT (SwinV2 U-Net) |
| Backbone parameters | $6.29 \times 10^8$ |
| Noise branch params | $1.52 \times 10^8$ |
| Noise embed. dim. | 32 |
| Patch size | 4 |
| Window size | 16 |
| Number of levels | 4 |
| Embed./latent dim. | 192 |
| Blocks per level | 8 |
| Attn. heads per level | [3, 6, 12, 24] |
| ConvNeXt blocks/level | 2 |
| Activation | GeLU |
| Normalisation | LayerNorm |

## B.2. Training Details

Figure 7 provides an overview of the training workflow for the single- and multi-system experiments.

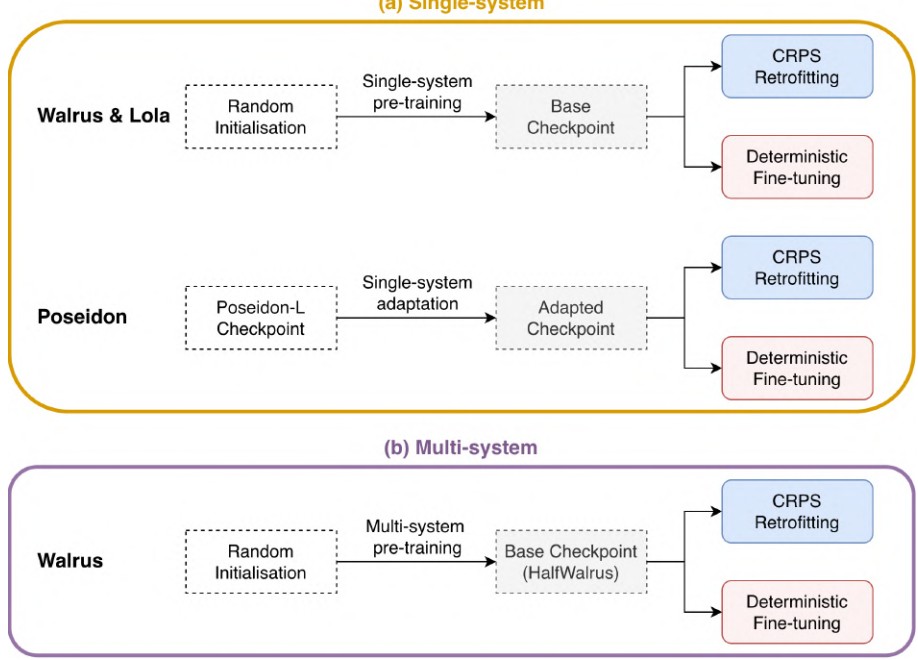

*Figure 7.* Training workflows for (a) single-system and (b) multi-system experiments. (a) In the single-system setting (top), we first establish a deterministic base checkpoint specific to the target dataset. This is achieved either by training from scratch (Walrus and Lola) or by adapting a foundation model to a target system that has not necessarily been seen during pre-training (Poseidon). From this checkpoint, we branch into two parallel paths to compare CRPS retrofitting against a continued deterministic fine-tuning baseline. (b) In the multi-system setting (bottom), we pre-train a foundation model on a mixture of systems, following the HalfWalrus protocol (McCabe et al., 2025). This pre-trained model serves as the shared base checkpoint, upon which we perform the same comparison between CRPS retrofitting and deterministic fine-tuning.

### B.2.1. SINGLE-SYSTEM MODELS

As illustrated in Figure 7, for the Walrus and Lola families, we start by training a deterministic model from scratch on each target dataset. Upon establishing this base model, we then train two models using the configurations detailed in Table 5: 1) a CRPS-retrofitted model, which incorporates noise injection and optimises the probabilistic objective, and 2) a continued deterministic fine-tuning baseline to serve as a direct benchmark.

For the Poseidon family, we initialise our experiments using the pre-trained POSEIDON-L foundation checkpoint. As the foundation model's training corpus did not encompass all the specific systems under test, we first employ an intermediate deterministic adaptation stage, where we fine-tune on single systems. From this system-adapted checkpoint, we replicate the strategy employed for Walrus and Lola: training separate CRPS-retrofitted and deterministic baseline variants.

For Walrus and Poseidon, because the model is operating on the same effective resolution, the training time was the same amongst all datasets. In contrast, the training time was system-dependent for Lola. In the tables we report the training configurations that resulted in the best-performing models.

**Base deterministic single-system checkpoint** For Walrus, training each base checkpoint required approximately 6.5 days on a single H100 GPU. For Lola, we leveraged existing checkpoints for the Rayleigh-Bénard (RB) and Euler systems. As reported by Rozet et al. (2025), training the autoencoders took 1 day (RB) and 2 days (Euler) on 8 H100 GPUs, while the latent-space emulators required 2 days (RB) and 5 days (Euler) on the same hardware. We additionally trained a new autoencoder and emulator for the Shear Flow dataset, which took approximately 1.5 and 3 days, respectively, on 8 H100 GPUs. Finally, for Poseidon, we initialised from the POSEIDON-L foundation checkpoint (trained for 165 hours on 8 NVIDIA GeForce RTX 4090 GPUs (Herde et al., 2024)) and performed single-system adaptation for approximately 1 day

on a single H100 GPU.

*Table 4.* Hyperparameters for the base single-system checkpoints. RB = Rayleigh-Bénard, E = Euler, and SF = Shear Flow.

| | Walrus | Lola | Poseidon |
|---|---|---|---|
| Optimiser | AdamW | AdamW | AdamW |
| Learning rate | 1e-3 | 1e-4 | 1e-4 |
| Weight decay | 1e-4 | 0.0 | 1e-4 |
| Warm-up Epochs | 10 | 0 | 10 |
| Cool-down Epochs | 10 | 0 | 10 |
| Scheduler | InvSqrt | Cosine | InvSqrt |
| Gradient norm clipping | 10.0 | 1.0 | 5.0 |
| Batch size | 8 | 256 (RB, E), 64 (SF) | 64 |
| GradAcc Steps | 1 | 1 | 1 |
| Steps per epoch | 2000 | 64 (RB, E), 256 (SF) | 500 |
| Epochs | 200 | 4096 | 200 |
| GPUs | 1 | 8 | 1 |
| Total unique samples | 3.2M | 67.1M | 6.4M |

**CRPS/Deterministic Retrofitted Checkpoint** All retrofitting experiments were conducted on a single H100 GPU. For Walrus, both CRPS-based and deterministic retrofitting required less than 2 days per system. For Lola, fine-tuning times varied by dataset: approximately 0.5 days for Rayleigh-Bénard (RB), less than 1 day for Shear Flow (SF), and roughly 1.5 days for Euler. For Poseidon, fine-tuning was completed in approximately 0.5 days. Detailed hyperparameters are provided in Table 5. Note that for CRPS retrofitting, the effective batch size is set to $1/M$ of the deterministic baseline's batch size. This adjustment compensates for the input batch being replicated $M$ times to generate the ensemble members. We use a default ensemble size of $M = 4$, though other values are explored in our ablation studies.

*Table 5.* Hyperparameters for the retrofitting procedure of single-system models. RB = Rayleigh-Bénard, E = Euler, and SF = Shear Flow.

| | Walrus | Lola | Poseidon |
|---|---|---|---|
| Optimiser | AdamW | AdamW | AdamW |
| Deterministic FT learning rate | 5e-5 | 3e-5 | 5e-5 |
| CRPS backbone / noise learning rate | 5e-5 / 1e-4 (RB) | 1e-4 / 3e-4 (RB) | 5e-5 / 1e-4 (RB) |
| | 5e-5 / 1e-4 (E) | 3e-5 / 1e-4 (E) | 5e-5 / 5e-4 (E) |
| | 1e-4 / 5e-4 (SF) | 1e-4 / 3e-3 (SF) | 5e-6 / 5e-5 (SF) |
| Weight decay | 1e-4 | 0.0 | 1e-4 |
| Warm-up Epochs | 5 | 0 (Det.) / 10 (CRPS) | 10 |
| Cool-down Epochs | 5 | 0 | 10 |
| Scheduler | InvSqrt | Cosine | InvSqrt |
| Gradient norm clipping | 10.0 | 1.0 | 5.0 |
| Effective batch size | 8 (Det.) / 2 (CRPS) | 128 / 32 (Det. / CRPS) | 64 (Det.) / 16 (CRPS) |
| GradAcc Steps | 1 | 1 (RB), 2 (SF), 4 (E) | 1 |
| Steps per epoch | 2000 | 512 | 500 |
| Epochs | 50 | 100 | 200 |
| GPUs | 1 | 1 | 1 |
| Total unique samples | 800k (Det.) / 200k (CRPS) | 6.55M (Det.) / 1.64M (CRPS) | 3.2M (Det.) / 800k (CRPS) |

### B.2.2. MULTI-SYSTEM MODELS

For the multi-system (HalfWalrus) experiments, we adhere to the workflow depicted in Figure 7: we first pre-train a foundation model on a heterogeneous dataset mixture to establish a base deterministic checkpoint. From this common initialisation, we then compare CRPS retrofitting against a deterministic fine-tuning baseline. Pre-training the foundation model required approximately 5 days on 8 H100 GPUs. Subsequent retrofitting steps were conducted on a single H100 GPU and completed in approximately 1.5 days. For the CRPS retrofitting phase in this setting, we utilised an ensemble size of $M = 2$.

*Table 6.* Hyperparameters for the multi-system (HalfWalrus) experiments. **Left:** Hyperparameters for training the base deterministic foundation model from scratch. **Right:** Hyperparameters for the subsequent fine-tuning (both deterministic and CRPS) on specific systems.

**Base Foundation Model**

| Parameter | HalfWalrus Base |
|---|---|
| Optimiser | AdamW |
| Learning rate | 1e-4 |
| Weight decay | 1e-4 |
| Warm-up Epochs | 10 |
| Cool-down Epochs | 10 |
| Scheduler | InvSqrt |
| Grad. norm clipping | 10.0 |
| Effective batch size | 8 |
| GradAcc Steps | 4 |
| Steps per epoch | 2000 |
| Epochs | 160 |
| GPUs | 8 |
| Total unique samples | 5.12M |

**Fine-Tuning / Retrofitting (Det. & CRPS)**

| Parameter | HalfWalrus FT |
|---|---|
| Optimiser | AdamW |
| Det. FT Learning Rate | 1e-5 (RB) |
| | 1e-5 (TRL) |
| | 5e-4 (VI) |
| CRPS backbone / noise LR | 1e-5 / 1e-4 (RB) |
| | 1e-4 / 1e-3 (TRL) |
| | 1e-4 / 5e-4 (VI) |
| Weight decay | 1e-4 |
| Warm-up Epochs | 5 |
| Cool-down Epochs | 5 |
| Scheduler | InvSqrt |
| Grad. norm clipping | 10.0 |
| Eff. batch size | 2 (Det.) / 1 (CRPS) |
| GradAcc Steps | 1 |
| Steps per epoch | 2000 |
| Epochs | 50 |
| GPUs | 1 |
| Total unique samples | 200k / 100k |

### B.2.3. SUMMARY OF COMPUTATIONAL OVERHEAD FOR CRPS RETROFITTING

We also provide a summary of the approximate computational overhead of the CRPS retrofitting procedure in Table 7. While the costs reported here reflect the default configuration used for the majority of results in this paper, our ablation studies suggest that this additional parameter overhead could be halved with relatively little performance loss, though the exact trade-off remains system-dependent.

*Table 7.* Computational cost comparison. CRPS Retrofitted values indicate the **additional** percentage cost relative to the deterministic baseline.

| Model | Method | Params | Wall-clock (H100 days) |
|---|---|---|---|
| **Walrus** | Det. Baseline | $2.77 \times 10^8$ | 6.5 |
| | CRPS Retrofitted | $+20\%$ | $+25\%$ |
| **Lola** | Det. Baseline | $2.22 \times 10^8$ | 16 (RB) / 40 (Euler) / 24 (Shear Flow) |
| | CRPS Retrofitted | $+3\%$ | $+3\%$ |
| **Poseidon** | Det. Baseline | $6.29 \times 10^8$ | 56[*] |
| | CRPS Retrofitted | $+24\%$ | $+1\%$ |
| **HalfWalrus** | Det. Baseline | $6.42 \times 10^8$ | 40 |
| | CRPS Retrofitted | $+34\%$ | $+4\%$ |

[*] Includes Pre-training (POSEIDON-L) + Deterministic Adaptation.

# C. Data

All datasets are sourced from The Well (Ohana et al., 2025), and we adhere to their canonical training, validation, and test splits. We evaluate Euler, Rayleigh-Bénard (RB), and Shear Flow for the single-system experiments, while the multi-system models are evaluated on RB, Turbulent Radiative Layer (TRL 2D), and Viscoelastic. We provide more details in Table 8, and refer to the original Ohana et al. (2025) paper for additional information.

These selections were guided by three primary factors:

1. Computational Efficiency: For the single-system setting, we prioritised datasets with available pre-trained checkpoints (specifically Euler and RB for the Lola family) to avoid the significant computational overhead of training new autoencoders and emulators from scratch.

2. Foundation Model Alignment: For the multi-system setting, our selection was constrained to the specific datasets included in the HalfWalrus foundation model's pre-training corpus (a subset comprising Maze, Discontinuous, Gray-Scott, RB, Shear Flow, TRL 2D, Staircase, and Viscoelastic).

3. Modelling Suitability: We specifically chose systems exhibiting complex or chaotic dynamics where probabilistic treatment offers the most significant theoretical advantage over deterministic methods.

*Table 8.* Dataset specifications for Euler Multi-Quadrants, Rayleigh-Bénard, Shear Flow, TRL (2D), and Viscoelastic simulations.

| | Euler Multi-Quadrants | Rayleigh-Bénard | Shear Flow | TRL (2D) | Viscoelastic |
|---|---|---|---|---|---|
| Software | Clawpack | Dedalus | Dedalus | Athena++ | Dedalus |
| Size | 5243 GB | 342 GB | 547 GB | 6.9 GB | 66 GB |
| Fields | energy, density, pressure, momentum | buoyancy, pressure, velocity | tracer, pressure, velocity | density, pressure, velocity | pressure, velocity, conformation tensor |
| Channels $C_{\text{pixel}}$ | 5 | 4 | 4 | 4 | 7 |
| Resolution | $512 \times 512$ | $512 \times 128$ | $256 \times 512$ | $384 \times 128$ | $512 \times 512$ |
| Trajectories | 10000 | 1750 | 1120 | 90 | 260 |
| Time steps $L$ | 100 | 200 | 200 | 101 | 20 - 60 |

We also provide a short description and motivation for probabilistic modelling for each of the five systems considered.

## C.1. Single-system Models

We use three different systems: **Rayleigh-Bénard** for chaotic thermal buoyancy, **Euler Multi-Quadrants** for compressible shocks and discontinuities, and **Shear Flow** for mechanical instability. This diversity ensures the proposed CRPS loss is robust across different sources of uncertainty—whether arising from shock wave locations, thermal fluctuations, or vortex shedding.

**Rayleigh-Bénard (RB)**   RB simulates a horizontal fluid layer heated from below, governed by coupled fluid and thermodynamic equations. The resulting temperature gradient creates buoyancy forces that drive the formation of convective "Bénard cells", characterised by rising warm plumes and sinking cool fluid. Their position is highly sensitive to small variations in the initial conditioning, motivating a probabilistic treatment. The dataset captures these chaotic vortices and boundary layers using 2 scalar (buoyancy, pressure) and one vector field (velocity) on a $512 \times 128$ grid. Each trajectory has 200 timesteps.

**Euler**[6]   This dataset models compressible, non-viscous gas dynamics governed by the Euler equations. Initialised with piecewise constant values in four quadrants, the system evolves into sharp interactions of shock waves and contact discontinuities—a regime where deterministic averaging could lead to blurring of fine features. The state is represented by 3 scalar fields (density, energy, and pressure), and one vector field (momentum) on a $512 \times 512$ grid. Each trajectory has 100 timesteps.

**Shear Flow**   This dataset explores 2D periodic shear flow, a non-linear phenomenon central to fluid mechanics where adjacent layers slide past each other. These flows become unstable at large Reynolds numbers, making their prediction

---

[6]For Walrus and Poseidon we only use the subset with periodic boundary conditions

under varying Reynolds and Schmidt numbers essential for applications in aerodynamics, automotive engineering, and biomedicine. Each state contains 2 scalar fields (tracer and pressure), and one vector field (velocity) at a resolution of $256 \times 512$. Each trajectory has 200 timesteps, and we evaluate up to 100.

### C.2. Multi-system Foundation Model

For the foundation model we picked three datasets from the pre-training HalfWalrus datasets, as specified in McCabe et al. (2025). Besides RB, we also chose **2D Turbulent Radiative Layer (TRL)** and **Viscoelastic Instability (VI)**.

**Turbulent Radiative Layer**  This dataset models the interface between moving hot and cold gas phases, common in interstellar environments. The dynamics are driven by Kelvin-Helmholtz instabilities seeded by initial random noise, which triggers turbulent mixing and rapid radiative cooling of the intermediate gas. The sensitivity to the initial seed motivates a probabilistic treatment. Each state is composed of 2 scalar (density and pressure) and one vector field (velocity) on a $384 \times 128$ grid. Each trajectory has 101 timesteps.

**Viscoelastic Instability (VI)**  This dataset captures elasto-inertial turbulence in dilute polymer solutions, exhibiting multistability: four distinct flow attractors (Laminar, Steady Arrowhead, EIT, Chaotic Arrowhead) coexist for identical parameters, determined solely by initial conditions. It also includes "edge states" on the boundaries between these regimes. Probabilistic modeling is essential here to represent the multimodal potential of these coexistent states. A state is composed of pressure (scalar field), velocity (vector field), and positive conformation tensor (with three tensor, and one scalar field). The dataset contains a mix of trajectories with 20 and 60 timesteps.

## D. Evaluation

### D.1. Metrics

We use the rollout VRMSE and fCRPS as the main evaluation metrics, with equations provided in the main manuscript. We additionally provide spread-to-skill ratios in Appendix E, and use the definition from Rozet et al. (2025). Say we have access to an ensemble of $M$ trajectories $v_k$. We define the skill as the RMSE of the ensemble mean:

$$\text{Skill} = \sqrt{\left\langle \left( u - \frac{1}{M} \sum_{k=1}^{M} v_k \right)^2 \right\rangle}, \tag{3}$$

where $\langle \cdot \rangle$ denotes the spatial mean operator. The spread is defined as the ensemble standard deviation:

$$\text{Spread} = \sqrt{\left\langle \frac{1}{M-1} \sum_{j=1}^{K} \left( v_j - \frac{1}{M} \sum_{k=1}^{K} v_k \right)^2 \right\rangle}. \tag{4}$$

Based on Fortin et al. (2014), the spread-to-skill ratio (SSR) for a perfect forecast where the ensemble members are exchangeable is

$$\text{SSR} = \frac{\text{Spread}}{\text{Skill}} \approx \sqrt{\frac{M}{M+1}}. \tag{5}$$

As such, we use the corrected spread-skill ratio $\text{SSR}_{\text{corrected}} = \frac{\text{Spread}}{\text{Skill}} \sqrt{\frac{M+1}{M}}$ as a metric (denoted as SSR in tables). For a well-calibrated ensemble, this should be around one—a ratio smaller than one indicates that the ensemble is biased or under-dispersed, whereas a ratio larger than one indicates over-dispersion. Note, however, that a SSR of 1 is only a necessary but not a sufficient condition for a perfect forecast, as also noted in Rozet et al. (2025).

**Power Spectra Metrics**  Let $p_u(k)$ and $p_v(k)$ represent the isotropic power spectra of the ground-truth and emulated fields respectively, averaged over the evaluated trajectories, evaluated at wave number $k$.

For a given frequency band $B$ containing $|B|$ discrete wave numbers, we define the logarithmic Root Mean Square Error (RMSE$_{\text{log}}$) as:

$$\mathrm{RMSE}_{\mathrm{log}}(B) = \sqrt{\frac{1}{|B|} \sum_{k \in B} \left( \log_{10}(p_v(k) + \epsilon) - \log_{10}(p_u(k) + \epsilon) \right)^2},$$

where $\epsilon = 10^{-6}$ is a small constant added for numerical stability.

Furthermore, to report a single aggregate metric across the primary frequency bands, we compute the total log-RMSE over the union of these targeted bands $\mathcal{B} = B_{\mathrm{low}} \cup B_{\mathrm{mid}} \cup B_{\mathrm{high}}$:

$$\mathrm{RMSE}_{\mathrm{log}}^{\mathrm{total}} = \sqrt{\frac{1}{|\mathcal{B}|} \sum_{k \in \mathcal{B}} \left( \log_{10}(p_v(k) + \epsilon) - \log_{10}(p_u(k) + \epsilon) \right)^2}$$

### D.2. Evaluation Procedure

To quantify performance, we employ a bootstrap resampling procedure (100 iterations) that accounts for the variability across trajectories.

**Absolute Metrics**  For absolute performance evaluation (rollout fCRPS and VRMSE), we first compute the scalar metric $E^{(i)}$ for each test trajectory $i$ by averaging over all time steps and physical fields. To estimate the population statistics, we resample the dataset with replacement and compute the aggregate metric (median) for each bootstrap sample. From the resulting distribution, we report the median as the central estimate. Uncertainty is quantified using the 95% confidence interval (2.5$^{\mathrm{th}}$–97.5$^{\mathrm{th}}$ percentiles) in tables, and the 68% confidence interval (16$^{\mathrm{th}}$–84$^{\mathrm{th}}$ percentiles) in figures.

**Relative Improvement**  When comparing the probabilistic model against the deterministic baseline, we adopt a paired bootstrap approach to ensure stability. Given the high variance in error magnitudes across flow regimes, trajectory-level percentage comparisons are susceptible to numerical instability where baseline errors are small (e.g., laminar flows). To ensure robustness, we evaluate the relative improvement of the aggregate errors (Ratio of Medians). Specifically, for each bootstrap iteration $b$, we sample the test trajectories with replacement and compute the median error for both the deterministic baseline ($\tilde{E}_{\mathrm{det}}^{(b)}$) and the probabilistic model ($\tilde{E}_{\mathrm{prob}}^{(b)}$). We then calculate the relative percentage improvement for that bootstrap sample as:

$$\delta^{(b)} = \frac{\tilde{E}_{\mathrm{det}}^{(b)} - \tilde{E}_{\mathrm{prob}}^{(b)}}{\tilde{E}_{\mathrm{det}}^{(b)}} \times 100. \tag{6}$$

We report the median of these bootstrapped improvements, using the same confidence interval conventions as defined above.

## E. Additional Results

### E.1. Single-system Models

#### E.1.1. ADDITIONAL NUMERICAL RESULTS

We present the full set of results in terms of rollout VRMSE, CRPS, and spread-skill ratio. Additionally, we also provide in Figure 8 the improvement in energy score (alongside CRPS), showing that the improvements in univariate metrics translate into multivariate ones. Deterministic (Base) refers to the deterministic baseline, Ours (CRPS) refers to the CRPS retrofitted model (starting from the deterministic baseline), and Deterministic (FT) refers to a model fine-tuned with the same pre-training loss on a compute-equal budget with the CRPS model.

We also provide a per-field analysis of the results for each dataset in Figure 9. The percentage improvement in metrics is consistent across fields, regardless of whether they are scalar or vector fields.

#### E.1.2. EXAMPLE ROLLOUTS

We provide example rollouts for each model and dataset. These trajectories were randomly sampled from the test set, selected to represent part of the variance of initial conditions. For RB we show the buoyancy field, for Euler we show the energy field, and for Shear Flow we show the tracer field.

We notice a few general trends worth commenting on, and we organise them on a model-by-model basis:

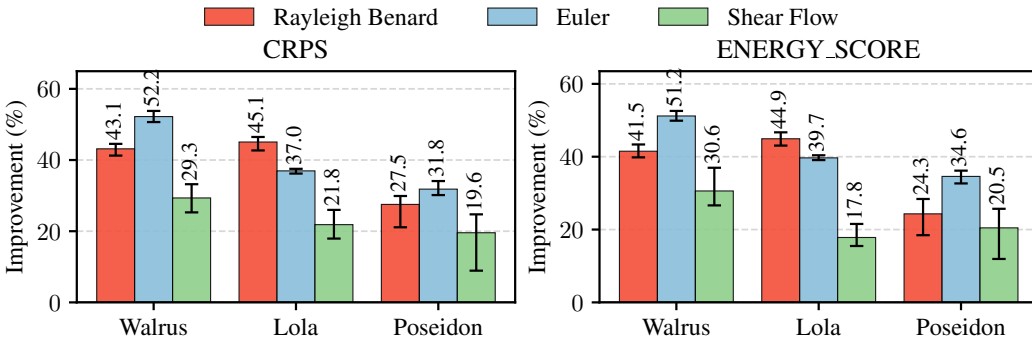

*Figure 8.* Improvement of CRPS retrofitting over deterministic fine-tuning for the single-system models: (Left) CRPS; (Right) Energy Score. Our method obtains similar percentage improvements in energy score as in CRPS, indicating that the improvements in univariate metrics translate into multivariate ones.

- **Walrus**: The CRPS retrofitted model produces highly realistic samples that effectively capture the inherent uncertainty of the system. In Rayleigh-Bénard (RB) convection, the CRPS samples are noticeably sharper than their deterministic counterparts, particularly for challenging initial conditions (Figure 11) where they remain realistic even late in the rollout. Crucially, the ensemble spread appears well-calibrated: for easier conditions where the deterministic model is accurate, the samples remain tightly clustered around the ground truth; conversely, as the system's inherent uncertainty (and the resulting error) increases, the samples disperse accordingly. We observe similar behavior in Euler, though the initial conditions there vary less. Finally, in Shear Flow, the model maintains high accuracy on simpler trajectories (Figures 16 and 18) while correctly increasing spread to account for uncertainty in harder regimes (Figure 17).

- **Lola**: In the case of RB, the CRPS samples accurately reflect forecast ambiguity, often correcting failures of the deterministic model. For instance, in Figure 10, the samples predict initial buoyancy movement (at $t = 35$) that the deterministic model misses. However, likely due to the compression inherent in its latent-space modelling, Lola struggles to capture the same level of fine-grained detail as Walrus on the hardest initial conditions (Figure 11), although it still maintains a reasonable uncertainty spread. In Shear Flow, the benefits of the CRPS approach are evident in the sharpness of the predictions, with the model resolving details—such as the vortex structure in the final time steps of Figure 17—that appear blurred in the deterministic output.

- **Poseidon**: Poseidon's performance appears constrained by the temporal bottleneck in its architecture, resulting in predictions that generally deviate the most from the ground truth across both deterministic and probabilistic modes. This limitation manifests as temporal artifacts, such as an "acceleration" of dynamics compared to the ground truth (Figure 10) and noticeable over-dispersion in the RB and Euler datasets (Figure 15). In Shear Flow, the model shows a trade-off: for simpler initial conditions (Figure 16), the CRPS samples successfully recover high-frequency details that the deterministic model over-smooths; however, in more complex scenarios (Figure 17), the samples tend to become unstable.

*Table 9.* Aggregated quantitative metrics (averaged across all fields). We show the median and the 95% confidence intervals over 100 bootstrapping iterations. Deterministic models are shaded in grey, while CRPS-retrofitted models are highlighted in purple. Best results are **bolded**.

| Model | Dataset | Method | CRPS | VRMSE | SSR |
|---|---|---|---|---|---|
| Walrus | Rayleigh Benard | Deterministic (Base) | 0.087 (0.083, 0.093) | 0.698 (0.672, 0.744) | - |
| | | Deterministic (FT) | 0.090 (0.080, 0.095) | 0.697 (0.677, 0.739) | - |
| | | Ours (CRPS) | **0.050 (0.046, 0.053)** | **0.558 (0.539, 0.579)** | 0.900 (0.860, 0.921) |
| | Euler | Deterministic (Base) | 0.079 (0.070, 0.085) | 0.382 (0.366, 0.396) | - |
| | | Deterministic (FT) | 0.082 (0.075, 0.090) | 0.389 (0.374, 0.408) | - |
| | | Ours (CRPS) | **0.037 (0.035, 0.045)** | **0.273 (0.267, 0.287)** | 0.909 (0.893, 0.926) |
| | Shear Flow | Deterministic (Base) | 0.0045 (0.0038, 0.0052) | 0.0714 (0.0560, 0.0927) | - |
| | | Deterministic (FT) | 0.0045 (0.0036, 0.0056) | 0.0648 (0.0524, 0.0804) | - |
| | | Ours (CRPS) | **0.0031 (0.0024, 0.0041)** | **0.0601 (0.0481, 0.0736)** | 0.8388 (0.8067, 0.9142) |
| Lola | Rayleigh Benard | Deterministic (Base) | 0.105 (0.094, 0.111) | 0.812 (0.790, 0.831) | - |
| | | Deterministic (FT) | 0.101 (0.094, 0.107) | 0.807 (0.789, 0.831) | - |
| | | Ours (CRPS) | **0.055 (0.051, 0.060)** | **0.632 (0.621, 0.652)** | 0.957 (0.941, 0.978) |
| | Euler | Deterministic (Base) | 0.045 (0.041, 0.049) | 0.284 (0.277, 0.292) | - |
| | | Deterministic (FT) | 0.045 (0.042, 0.049) | 0.285 (0.276, 0.295) | - |
| | | Ours (CRPS) | **0.028 (0.027, 0.031)** | **0.250 (0.241, 0.260)** | 1.103 (1.097, 1.109) |
| | Shear Flow | Deterministic (Base) | 0.0079 (0.0066, 0.0091) | 0.1188 (0.1044, 0.1366) | - |
| | | Deterministic (FT) | 0.0076 (0.0069, 0.0092) | 0.1200 (0.1044, 0.1360) | - |
| | | Ours (CRPS) | **0.0059 (0.0052, 0.0075)** | **0.1047 (0.0902, 0.1407)** | 0.6094 (0.5864, 0.6274) |
| Poseidon | Rayleigh Benard | Deterministic (Base) | 0.100 (0.097, 0.108) | 0.770 (0.713, 0.814) | - |
| | | Deterministic (FT) | 0.095 (0.088, 0.103) | 0.740 (0.713, 0.769) | - |
| | | Ours (CRPS) | **0.071 (0.059, 0.083)** | **0.713 (0.664, 0.834)** | 1.615 (1.270, 1.955) |
| | Euler | Deterministic (Base) | 0.131 (0.126, 0.141) | 0.558 (0.541, 0.574) | - |
| | | Deterministic (FT) | 0.131 (0.117, 0.141) | 0.532 (0.519, 0.554) | - |
| | | Ours (CRPS) | **0.088 (0.080, 0.096)** | **0.499 (0.489, 0.515)** | 1.117 (1.105, 1.134) |
| | Shear Flow | Deterministic (Base) | 0.0083 (0.0066, 0.0096) | 0.0892 (0.0792, 0.0976) | - |
| | | Deterministic (FT) | 0.0074 (0.0057, 0.0081) | **0.0849 (0.0806, 0.0904)** | - |
| | | Ours (CRPS) | **0.0059 (0.0050, 0.0075)** | 0.0852 (0.0761, 0.0984) | 0.9773 (0.9257, 1.0295) |

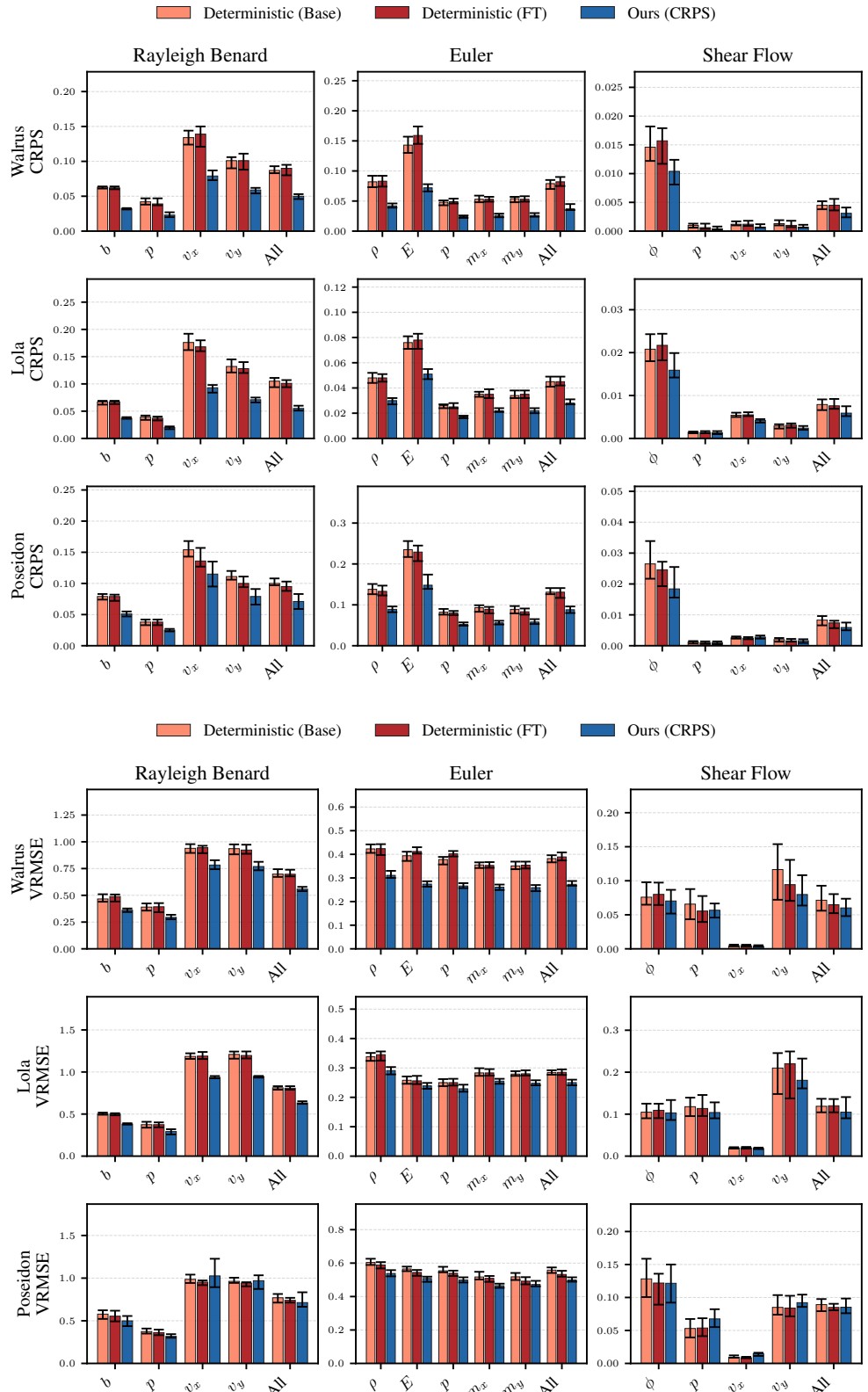

*Figure 9.* Per-field rollout metrics—CRPS (top) and VRMSE (bottom)—for the deterministic baseline, and two fine-tuned models on a compute-equal budget: Deterministic (FT) with the same pre-training loss, and one with a CRPS loss (Ours (CRPS)). The error bars indicate a 95% confidence interval from 100 bootstrapping iterations.

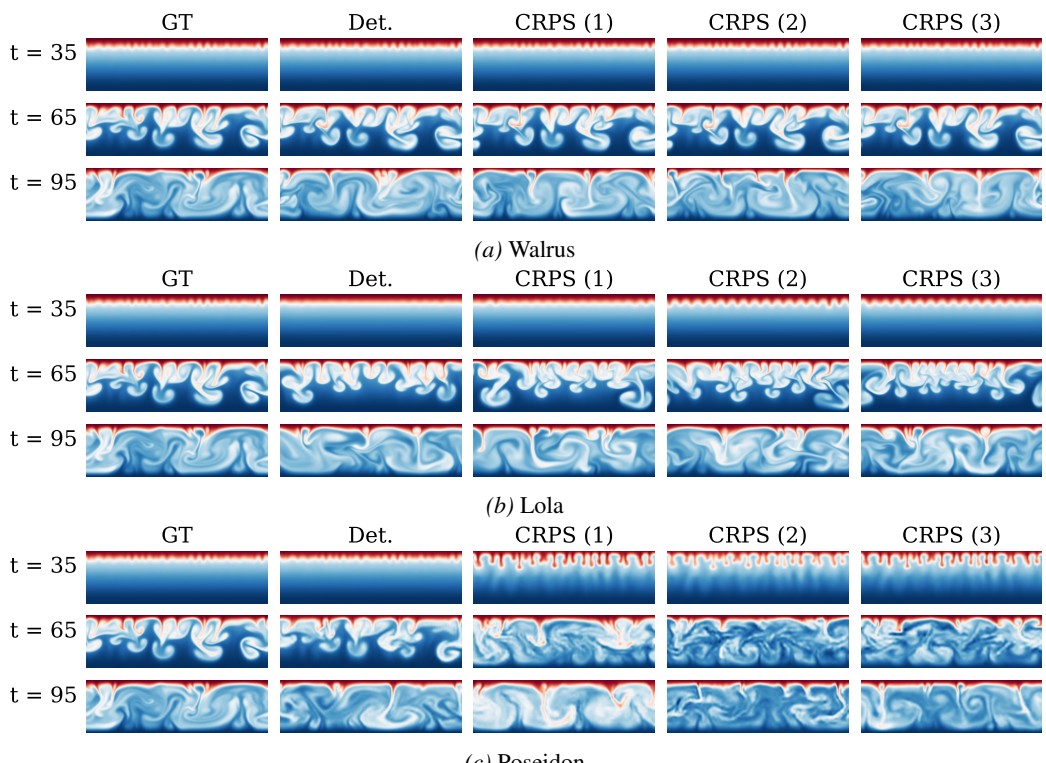

*Figure 10.* First example of rollouts for the Rayleigh-Bénard dataset, for each of the three single-system models: Walrus (top), Lola (middle), and Poseidon (bottom).

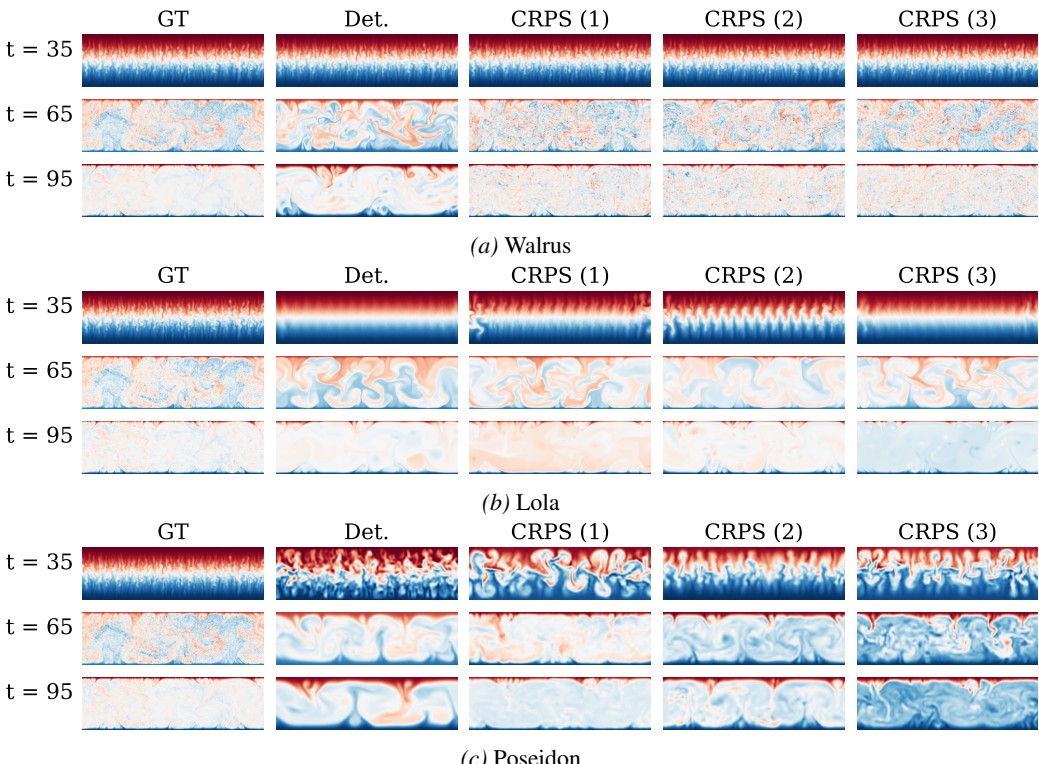

*Figure 11.* Second example of rollouts for the Rayleigh-Bénard dataset, for each of the three single-system models: Walrus (top), Lola (middle), and Poseidon (bottom).

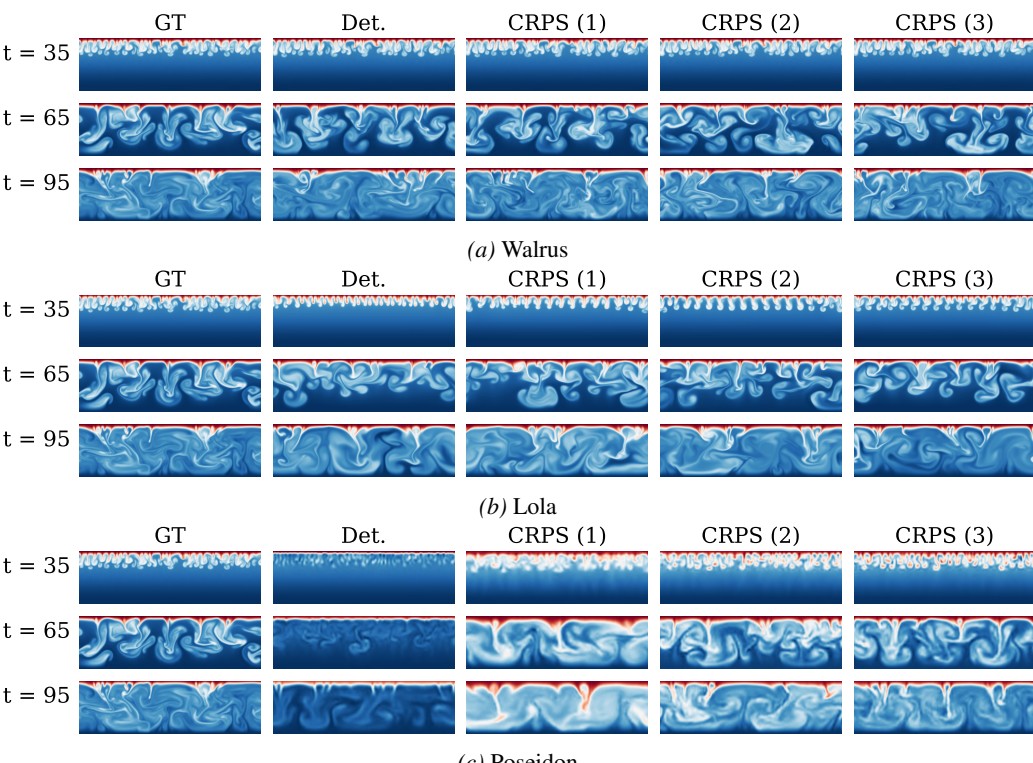

*Figure 12.* Third example of rollouts for the Rayleigh-Bénard dataset, for each of the three single-system models: Walrus (top), Lola (middle), and Poseidon (bottom).

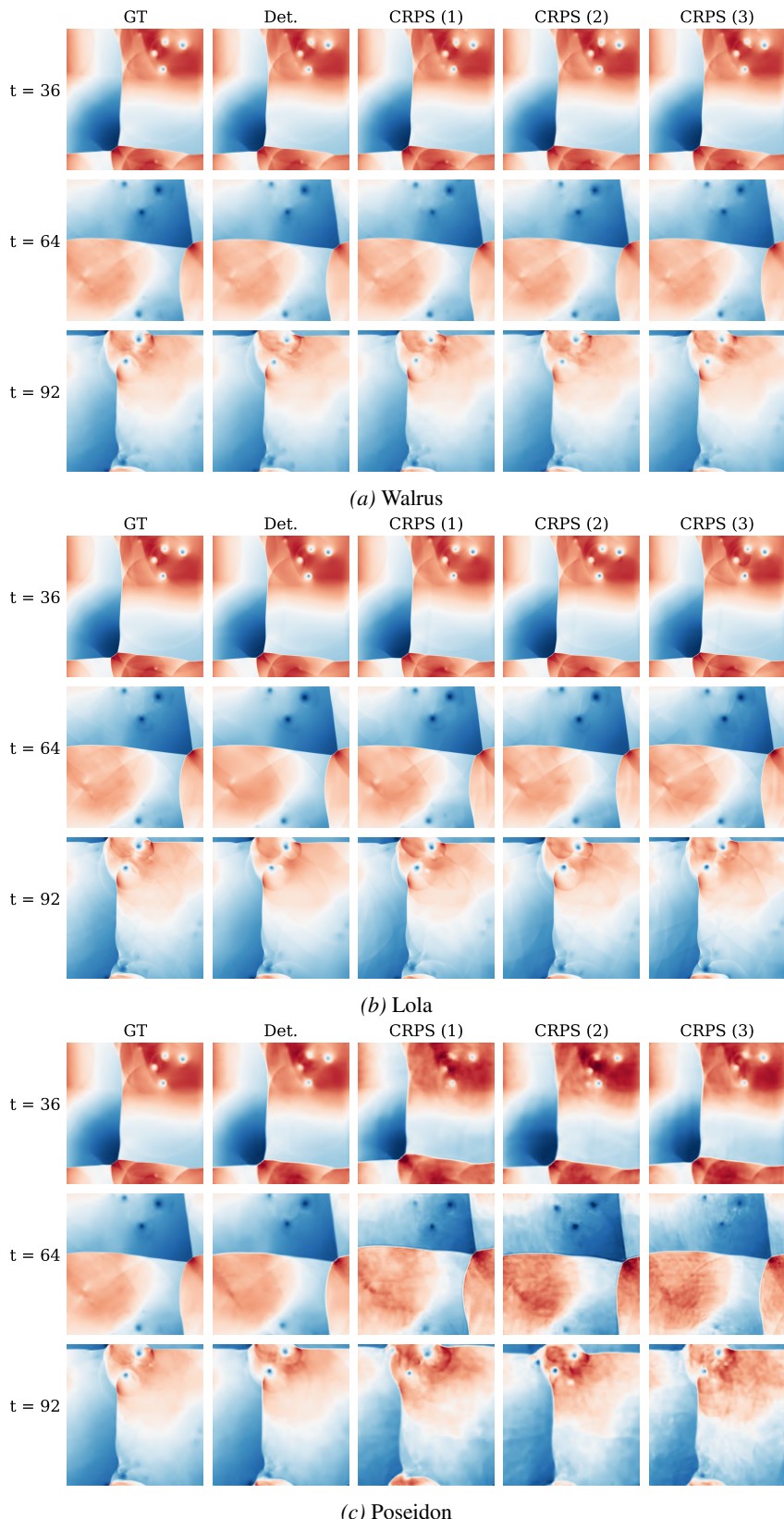

*Figure 13.* First example of rollouts for the Euler dataset, for each of the three single-system models: Walrus (top), Lola (middle), and Poseidon (bottom).

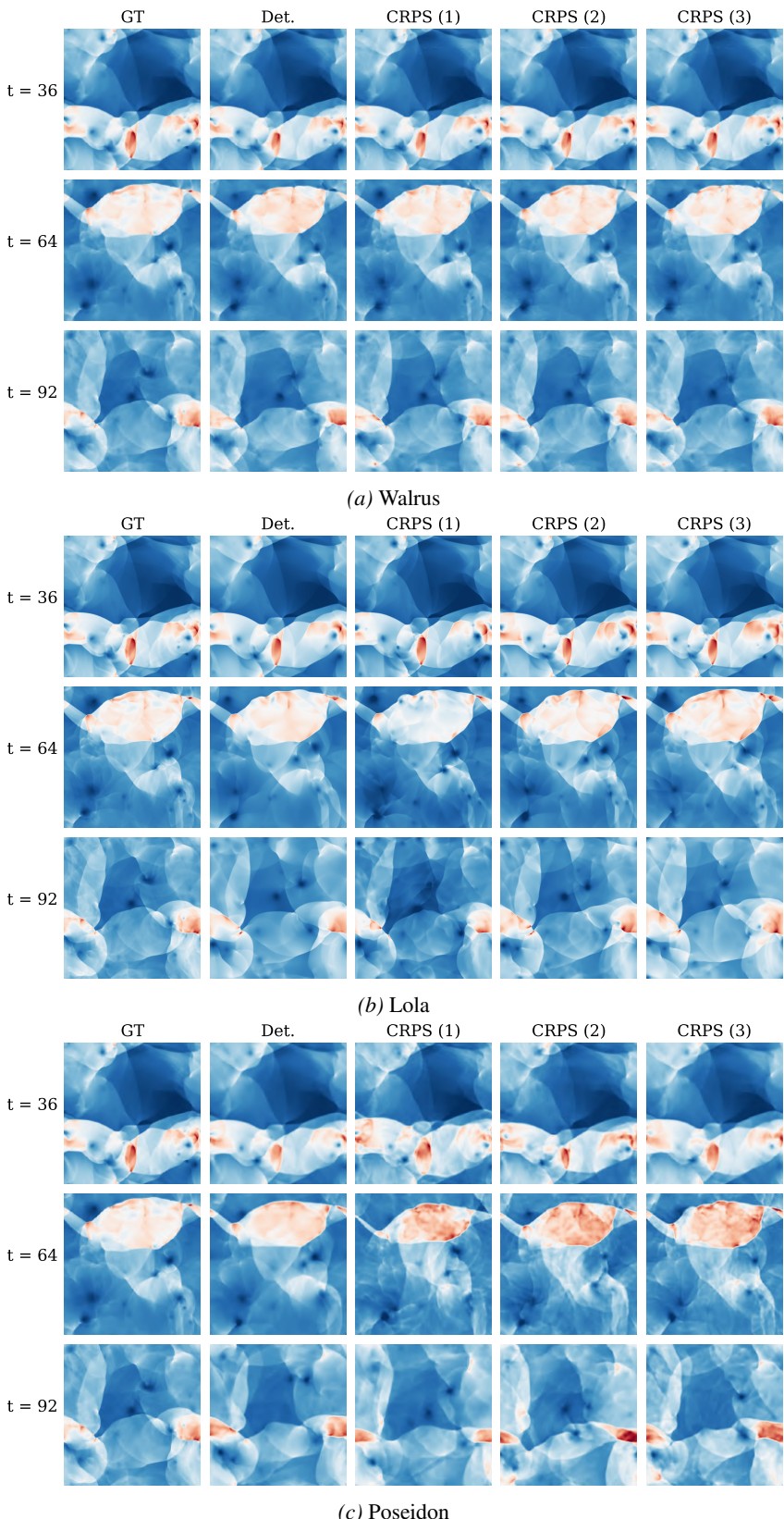

*(a)* Walrus

*(b)* Lola

*(c)* Poseidon

*Figure 14.* Second example of rollouts for the Euler dataset, for each of the three single-system models: Walrus (top), Lola (middle), and Poseidon (bottom).

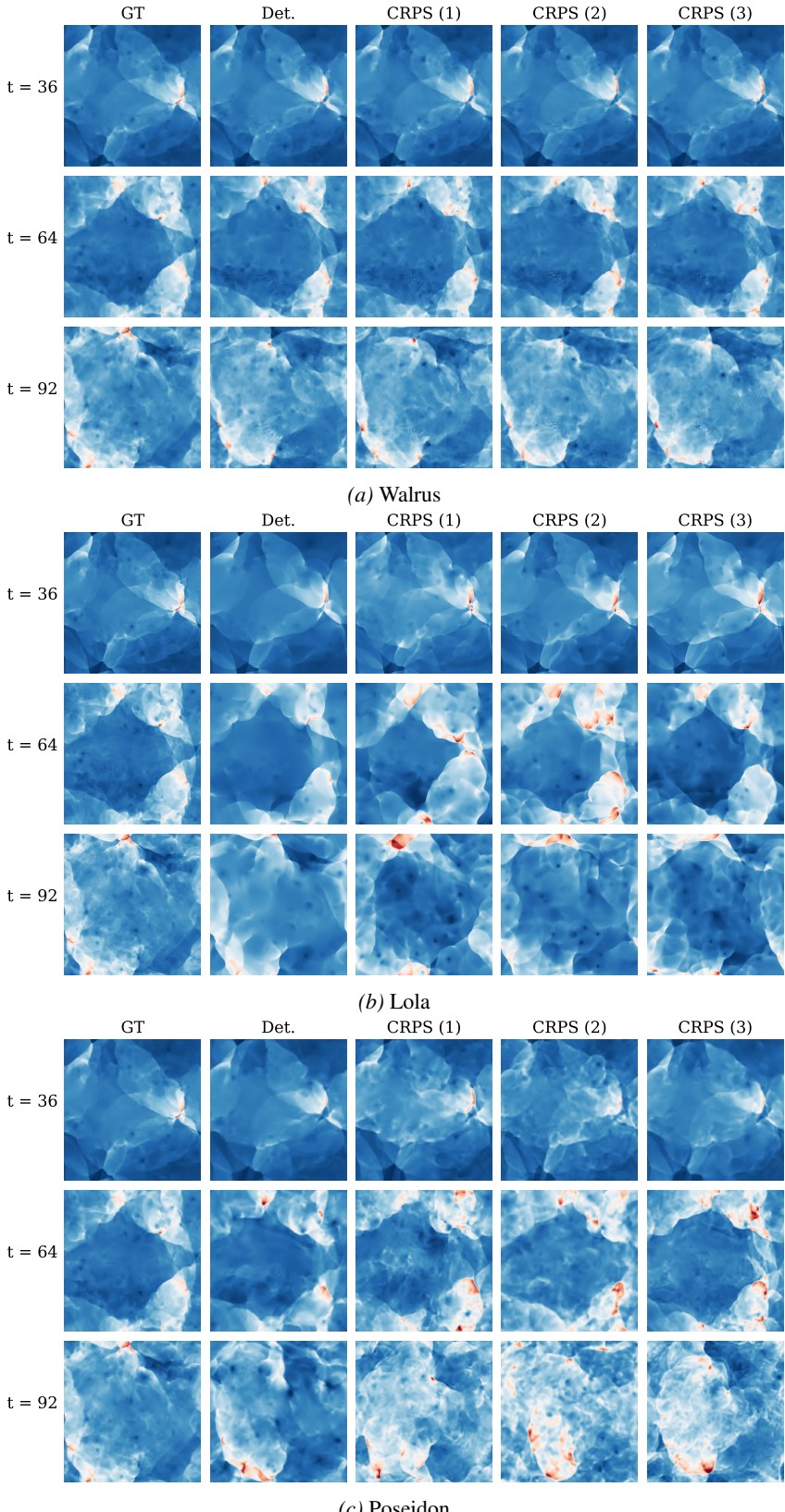

*Figure 15.* Third example of rollouts for the Euler dataset, for each of the three single-system models: Walrus (top), Lola (middle), and Poseidon (bottom).

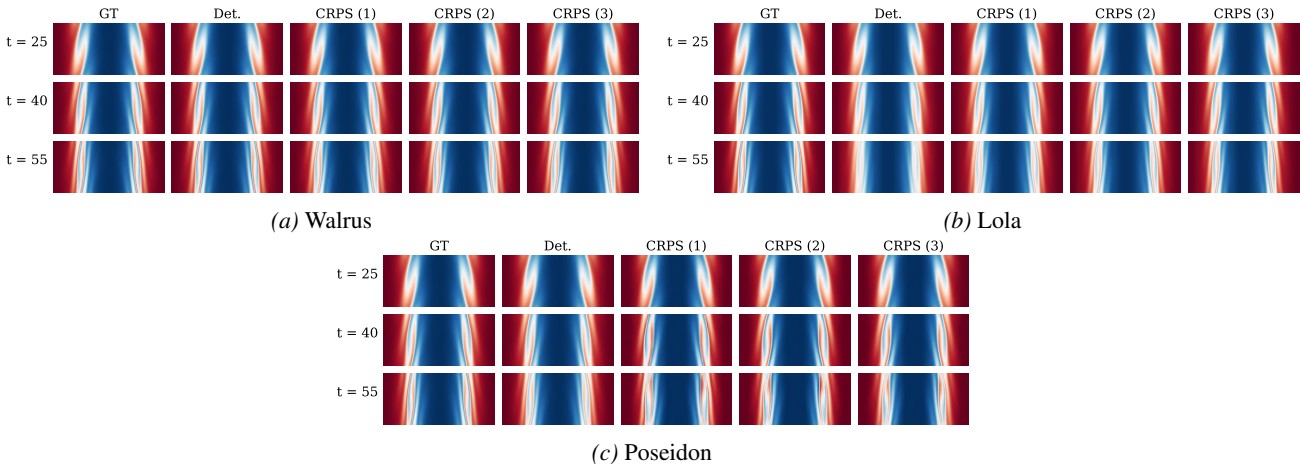

*Figure 16.* First example of rollouts for the Shear Flow dataset, for each of the three single-system models: Walrus (top), Lola (middle), and Poseidon (bottom).

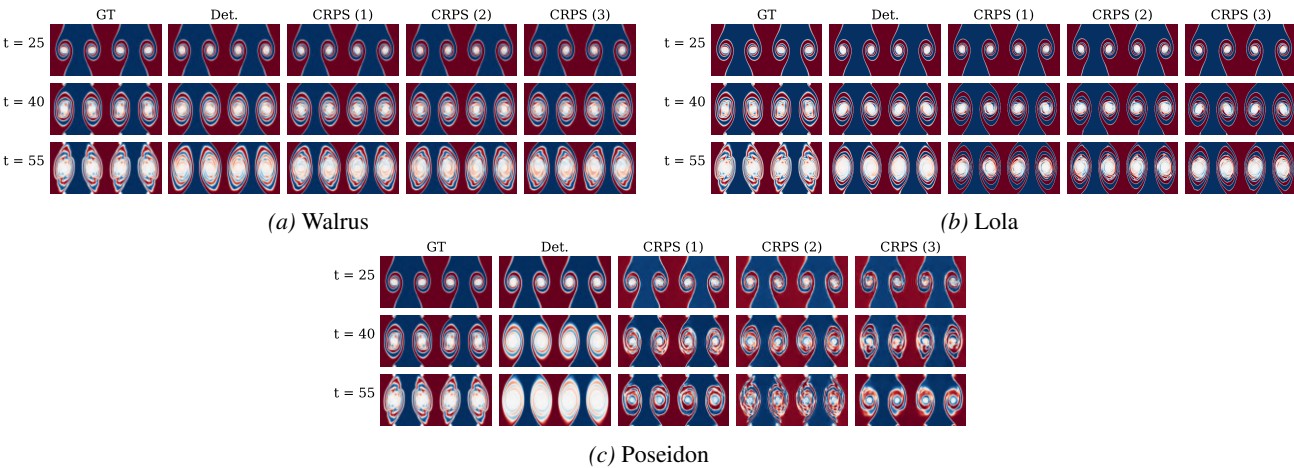

*Figure 17.* Second example of rollouts for the Shear Flow dataset, for each of the three single-system models: Walrus (top), Lola (middle), and Poseidon (bottom).

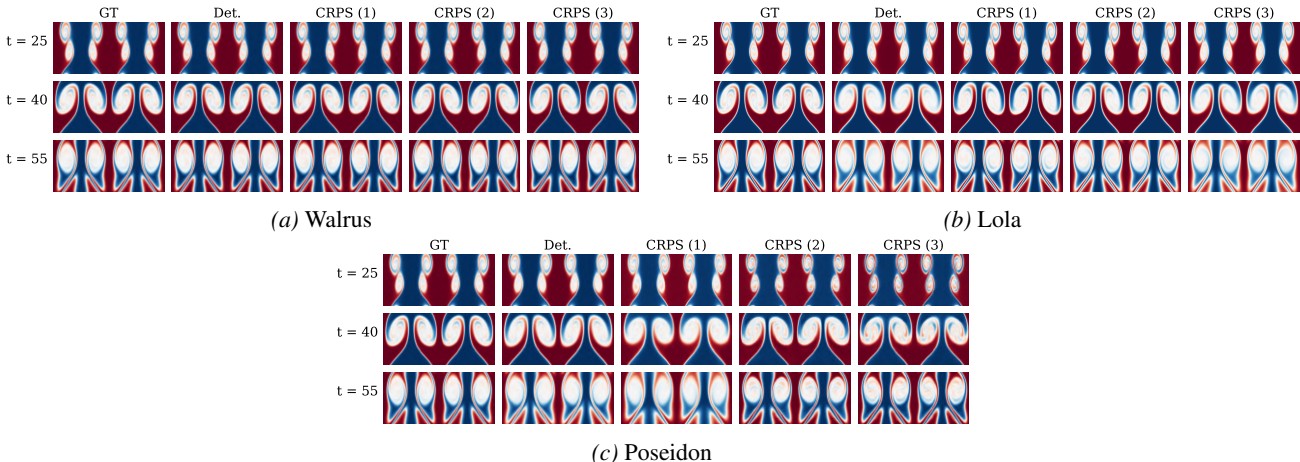

*Figure 18.* Third example of rollouts for the Shear Flow dataset, for each of the three single-system models: Walrus (top), Lola (middle), and Poseidon (bottom).

### E.1.3. IMPROVEMENT WITH ADDITIONAL ENSEMBLE MEMBERS

Our method also offers a test-time knob, where performance can be improved by generating more ensemble members (at the expense of increased compute). Figure 19 shows how the rollout VRMSE normalised by the single-member baseline decreases with more members for all datasets and models.

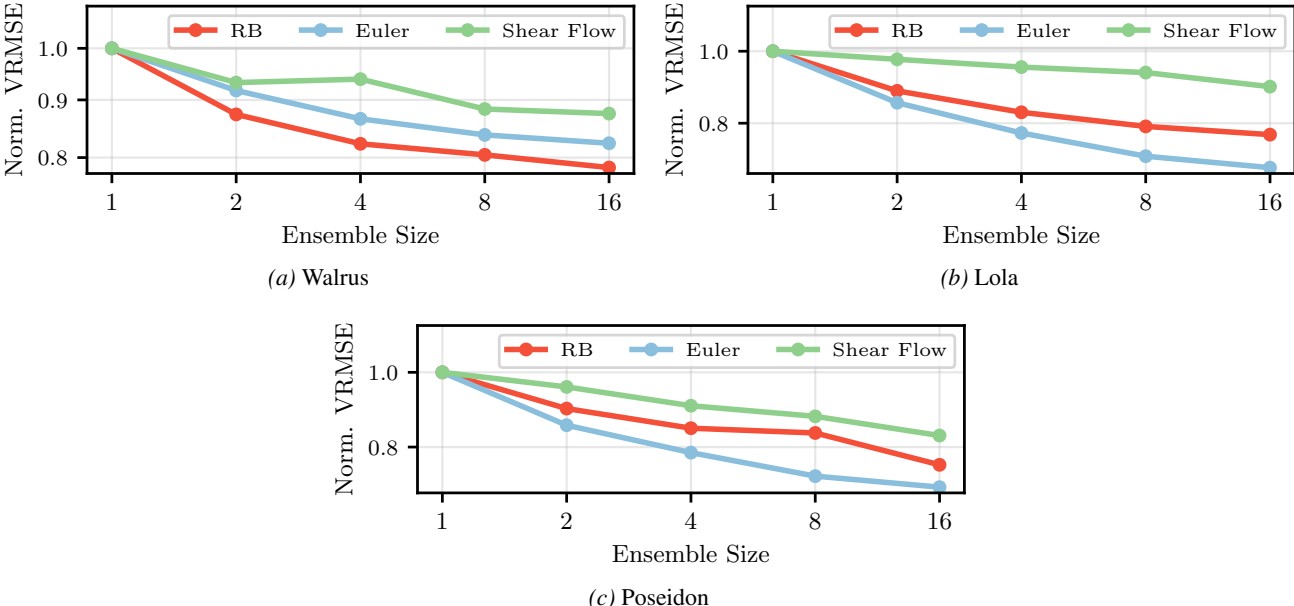

*(a)* Walrus

*(b)* Lola

*(c)* Poseidon

*Figure 19.* Error scaling. Log-log plot of rollout VRMSE (normalised by the single-member baseline) vs. ensemble size ($M$) for the (a) Walrus model, (b) Lola, and (c) Poseidon.

Additionally, Figure 20 visualises the temporal evolution of VRMSE across varying ensemble sizes. These plots demonstrate that the benefits of ensembling are most pronounced towards the end of the rollout, where system uncertainty is highest.

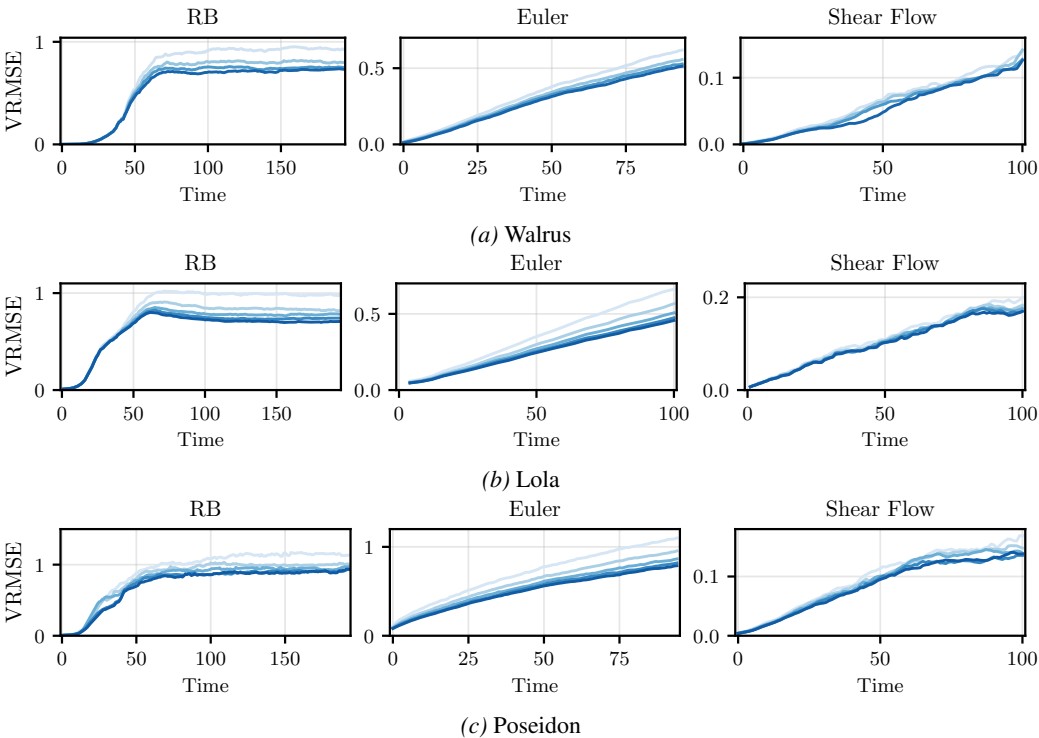

*Figure 20.* Evolution of VRMSE throughout rollout length with increasing ensemble size (1, 2, 4, and 16) indicated by darker shades of blue. We show results for (a) Walrus, (b) Lola, and (c) Poseidon.

### E.1.4. ABLATION STUDIES

Although CRPS retrofitting is generally robust, performance can be sensitive to specific hyperparameters. We present the results of several ablation studies to identify the most critical performance factors and provide some practical recommendations for applying this technique to new tasks.

*Table 10.* Full quantitative metrics for the learning rate sweep. The **bolded** results correspond to the models used in the main paper. Ep. = Number of epochs, M = Number of training ensemble members, LR (B) = Backbone learning rate, LR (N) = Noise Branch learning rate.

| Model | Dataset | Ep. | M | LR (B) | LR (N) | CRPS | VRMSE | SSR |
|---|---|---|---|---|---|---|---|---|
| Walrus | Rayleigh Benard | 50 | 4 | 5e-4 | 1e-3 | 0.054 (0.051, 0.058) | 0.593 (0.570, 0.652) | 1.001 (0.963, 1.051) |
| | | 50 | 4 | 1e-4 | 5e-4 | 0.051 (0.048, 0.055) | 0.571 (0.546, 0.596) | 0.899 (0.878, 0.933) |
| | | 50 | 4 | 5e-5 | 1e-4 | **0.050 (0.046, 0.053)** | **0.558 (0.539, 0.579)** | 0.900 (0.860, 0.921) |
| | Euler | 50 | 4 | 5e-4 | 1e-3 | 0.051 (0.049, 0.055) | 0.330 (0.324, 0.338) | 0.881 (0.869, 0.897) |
| | | 50 | 4 | 1e-4 | 5e-4 | 0.038 (0.035, 0.043) | 0.275 (0.260, 0.286) | 0.900 (0.878, 0.913) |
| | | 50 | 4 | 5e-5 | 1e-4 | **0.037 (0.035, 0.045)** | **0.273 (0.267, 0.287)** | 0.909 (0.893, 0.926) |
| | Shear Flow | 50 | 4 | 1e-4 | 5e-4 | **0.0031 (0.0024, 0.0041)** | **0.0601 (0.0481, 0.0736)** | 0.8388 (0.8067, 0.9142) |
| | | 50 | 4 | 5e-5 | 1e-4 | 0.0034 (0.0027, 0.0039) | 0.0649 (0.0450, 0.0733) | 0.8263 (0.7932, 0.8923) |
| | | 50 | 4 | 1e-5 | 5e-5 | 0.0037 (0.0029, 0.0046) | 0.0630 (0.0471, 0.0793) | 0.8081 (0.7734, 0.8465) |
| Lola | Rayleigh Benard | 100 | 4 | 1e-4 | 3e-4 | **0.055 (0.051, 0.060)** | **0.632 (0.621, 0.652)** | 0.957 (0.941, 0.978) |
| | | 100 | 4 | 3e-5 | 1e-4 | 0.056 (0.051, 0.062) | 0.635 (0.626, 0.642) | 0.968 (0.951, 0.985) |
| | | 100 | 4 | 3e-5 | 5e-5 | 0.056 (0.052, 0.060) | 0.644 (0.635, 0.654) | 0.971 (0.954, 0.994) |
| | Euler | 100 | 4 | 1e-4 | 3e-4 | 0.028 (0.027, 0.031) | 0.253 (0.243, 0.262) | 1.053 (1.043, 1.057) |
| | | 100 | 4 | 3e-5 | 1e-4 | **0.028 (0.027, 0.031)** | **0.250 (0.241, 0.260)** | 1.103 (1.097, 1.109) |
| | | 100 | 4 | 3e-5 | 5e-5 | 0.028 (0.027, 0.030) | 0.251 (0.241, 0.262) | 1.112 (1.105, 1.116) |
| | Shear Flow | 100 | 4 | 1e-4 | 3e-3 | **0.0059 (0.0052, 0.0075)** | **0.1047 (0.0902, 0.1407)** | 0.6094 (0.5864, 0.6274) |
| | | 100 | 4 | 1e-4 | 1e-3 | 0.0068 (0.0061, 0.0083) | 0.1084 (0.0870, 0.1350) | 0.4395 (0.3881, 0.4882) |
| | | 100 | 4 | 1e-4 | 3e-4 | 0.0070 (0.0061, 0.0090) | 0.1093 (0.0874, 0.1359) | 0.3075 (0.2836, 0.3597) |
| | | 100 | 4 | 3e-5 | 3e-3 | 0.0057 (0.0051, 0.0075) | 0.1235 (0.1013, 0.1512) | 0.7529 (0.7099, 0.8095) |
| Poseidon | Rayleigh Benard | 100 | 4 | 1e-4 | 5e-4 | 0.072 (0.065, 0.085) | 0.775 (0.676, 0.980) | 1.552 (1.231, 1.835) |
| | | 100 | 4 | 5e-5 | 5e-4 | 0.071 (0.063, 0.081) | 0.810 (0.743, 0.946) | 1.491 (1.296, 1.834) |
| | | 100 | 4 | 5e-5 | 1e-4 | **0.071 (0.059, 0.083)** | **0.713 (0.664, 0.834)** | 1.615 (1.270, 1.955) |
| | | 100 | 4 | 1e-5 | 5e-5 | 0.075 (0.065, 0.084) | 0.808 (0.706, 2.108) | 1.676 (1.270, 2.191) |
| | | 100 | 4 | 5e-6 | 5e-5 | 0.079 (0.066, 0.091) | 0.988 (0.648, 21.728) | 2.174 (1.321, 2.553) |
| | Euler | 100 | 4 | 1e-4 | 5e-4 | 0.100 (0.095, 0.106) | 0.563 (0.546, 0.576) | 0.974 (0.960, 0.995) |
| | | 100 | 4 | 5e-5 | 1e-3 | 0.090 (0.085, 0.096) | 0.518 (0.504, 0.528) | 1.043 (1.030, 1.061) |
| | | 100 | 4 | 5e-5 | 5e-4 | **0.088 (0.080, 0.096)** | **0.499 (0.489, 0.515)** | 1.117 (1.105, 1.134) |
| | | 100 | 4 | 5e-5 | 1e-4 | 0.103 (0.098, 0.109) | 0.580 (0.567, 0.594) | 0.936 (0.916, 0.950) |
| | | 100 | 4 | 1e-5 | 1e-4 | 0.090 (0.083, 0.095) | 0.520 (0.506, 0.534) | 1.178 (1.153, 1.203) |
| | | 100 | 4 | 1e-5 | 5e-5 | 0.094 (0.087, 0.100) | 0.534 (0.524, 0.543) | 1.046 (1.028, 1.064) |
| | | 100 | 4 | 5e-6 | 1e-5 | 0.092 (0.086, 0.098) | 0.528 (0.509, 0.542) | 1.076 (1.059, 1.099) |
| | Shear Flow | 100 | 4 | 5e-5 | 1e-4 | 0.0071 (0.0062, 0.0094) | 0.0952 (0.0801, 0.1043) | 0.8550 (0.7682, 0.8911) |
| | | 100 | 4 | 1e-5 | 1e-4 | 0.0063 (0.0050, 0.0082) | 0.0892 (0.0799, 0.0990) | 0.9193 (0.8634, 1.0225) |
| | | 100 | 4 | 1e-5 | 5e-5 | 0.0060 (0.0045, 0.0076) | 0.0889 (0.0807, 0.0973) | 1.0257 (0.9425, 1.1238) |
| | | 100 | 4 | 5e-6 | 5e-5 | **0.0059 (0.0050, 0.0075)** | **0.0852 (0.0761, 0.0984)** | 0.9773 (0.9257, 1.0295) |
| | | 100 | 4 | 5e-6 | 1e-5 | 0.0059 (0.0047, 0.0080) | 0.0914 (0.0812, 0.1011) | 1.0146 (0.9790, 1.0698) |

**Learning Rate**   The choice of learning rate (LR)—for both the deterministic backbone and the newly introduced noise branch (noise MLP and conditional normalisation layers)—has the most significant impact on performance. We consistently employ a lower LR for the backbone to preserve the representations learned during pre-training, while still allowing sufficient adaptation to the new loss.

As detailed in Table 10, finding an optimal configuration requires tuning, but the search space is not prohibitively brittle. We observe that provided the learning rates fall within an appropriate regime for each model, multiple configurations often yield statistically comparable results.

For Walrus and Lola, performance remained robust across most settings for the RB and Euler datasets, with overlapping confidence intervals accommodating both modest and higher learning rates. However, we observed distinct behaviors for specific configurations and the Shear Flow dataset:

- Impact of Aggressive Learning Rates: While generally robust, Walrus degrades with an excessively aggressive noise branch learning rate (1e-3) in Euler (and partially in RB), likely requiring extended training to converge.
- Shear Flow specifics: This dataset generally benefits from higher learning rates in Walrus. For Lola, aggressive noise branch learning rates were essential to prevent severe underdispersion on Shear Flow.
- Poseidon Sensitivity: In contrast to the other models, Poseidon showed a pronounced dependency on learning rate, requiring careful tuning. We generally found that lower learning rates were necessary to maintain stability for this model.

Corroborating these results, it appears that models that utilise time history for conditioning (Walrus and Lola) appear more robust to hyperparameter choices than those that do not (Poseidon). Additionally, Shear Flow emerged as the most sensitive dataset across the board. We hypothesise this sensitivity stems from the dataset's high heterogeneity in initial conditions and the state of the underlying backbone. As argued in the main text, CRPS retrofitting appears most efficient on (nearly) converged deterministic models, whereas applying it to models that still benefit from deterministic training might lead to heightened hyperparameter sensitivity.

**Number of Ensemble Members**   We limit this ablation study to the Walrus and Lola architectures, as they proved to be the most stable configurations, making them the reliable candidates for isolating the effects of ensemble size. As shown in Table 11, increasing the number of training ensemble members ($M$) yields diminishing returns. Provided the learning rate is tuned, utilising larger ensemble sizes results in negligible performance gains, with metrics consistently falling within overlapping confidence intervals.

- **CRPS Robustness:** The rollout CRPS remains robustly similar, regardless of the value of $M$.
- **VRMSE Sensitivity:** While rollout VRMSE occasionally improves with increasing ensemble size (e.g., decreasing from 0.584 to 0.558 for Walrus on RB), these differences remain within error margins.

These findings suggest that while we employ $M = 4$ in the main paper to ensure a generous computational budget, the CRPS retrofitting procedure remains highly effective even with smaller, more computationally efficient ensembles.

**Retrofitting Duration**   We restrict this analysis to the Walrus architecture to evaluate the trade-off between training compute and performance. As demonstrated in Table 12, extending the retrofitting duration from 50 to 100 epochs yields negligible gains. In the optimal configurations, the rollout CRPS and VRMSE generally plateau. This confirms that, provided the learning dynamics are carefully tuned, CRPS retrofitting is a training-efficient technique, achieving convergence without requiring extensive computational overhead. We anticipate that rather than simply increasing the duration of the current retrofitting phase, more significant gains in rollout metrics could be achieved by transitioning to rollout fine-tuning after the initial 50 epochs.

**Noise Embedding Dimension**   We next assess the sensitivity of the method to the noise embedding dimension. As presented in Table 13, comparing a dimension of 32 (used in the main experiments) against a reduced dimension of 16 reveals that the rollout CRPS remains largely robust to this choice, particularly on the RB and Euler datasets where differences are minimal. However, the rollout VRMSE generally benefits from the higher embedding dimension. This trend is most pronounced on the Shear Flow dataset, where the larger dimension (32) yields noticeable gains in both metrics, justifying the use of sufficient capacity to capture heterogeneous dynamics.

**Number of Layers with Noise Injection**   Finally, we investigate the sensitivity of the method to the density of noise injection. Since the modulation heads in the processor blocks account for the majority of the added parameters, we assess whether a sparser injection pattern is sufficient to maintain performance, while significantly reducing the parameter count overhead. We compare our default "full" configuration (injection in every block) against a "half" variant (injection in every alternate block). As shown in Table 14, the rollout CRPS remains remarkably consistent across both configurations. While the rollout VRMSE exhibits a slight trend favouring the dense "full" configuration—particularly on the RB and Shear Flow datasets—these improvements largely remain within confidence intervals. This suggests that while dense noise

injection offers marginal gains in minimising variance error, the CRPS retrofitting method is fundamentally robust to sparser conditioning strategies.

*Table 11.* Full quantitative metrics for the number of ensemble members ablation. The **bolded** results correspond to the models used in the main paper. Ep. = Number of epochs, M = Number of training ensemble members, LR (B) = Backbone learning rate, LR (N) = Noise Branch learning rate

| Model | Dataset | Ep. | M | LR (B) | LR (N) | CRPS | VRMSE | SSR |
|-------|---------|-----|---|--------|--------|------|-------|-----|
| Walrus | Rayleigh Benard | 50 | 3 | 5e-4 | 1e-3 | 0.055 (0.051, 0.061) | 0.627 (0.600, 0.649) | 0.968 (0.939, 0.985) |
| | | 50 | 4 | 5e-4 | 1e-3 | 0.054 (0.051, 0.058) | 0.593 (0.570, 0.652) | 1.001 (0.963, 1.051) |
| | | 50 | 3 | 1e-4 | 5e-4 | 0.050 (0.047, 0.054) | 0.583 (0.556, 0.606) | 0.916 (0.885, 0.956) |
| | | 50 | 4 | 1e-4 | 5e-4 | 0.051 (0.048, 0.055) | 0.571 (0.546, 0.596) | 0.899 (0.878, 0.933) |
| | | 50 | 3 | 5e-5 | 1e-4 | 0.050 (0.046, 0.053) | 0.584 (0.534, 0.612) | 0.905 (0.876, 0.925) |
| | | 50 | 4 | 5e-5 | 1e-4 | **0.050 (0.046, 0.053)** | **0.558 (0.539, 0.579)** | 0.900 (0.860, 0.921) |
| | Euler | 50 | 3 | 5e-4 | 1e-3 | 0.060 (0.055, 0.064) | 0.360 (0.352, 0.371) | 0.775 (0.764, 0.792) |
| | | 50 | 4 | 5e-4 | 1e-3 | 0.051 (0.049, 0.055) | 0.330 (0.324, 0.338) | 0.881 (0.869, 0.897) |
| | | 50 | 3 | 1e-4 | 5e-4 | 0.039 (0.037, 0.045) | 0.280 (0.269, 0.291) | 0.916 (0.900, 0.935) |
| | | 50 | 4 | 1e-4 | 5e-4 | 0.038 (0.035, 0.043) | 0.275 (0.260, 0.286) | 0.900 (0.878, 0.913) |
| | | 50 | 3 | 5e-5 | 1e-4 | 0.038 (0.035, 0.042) | 0.271 (0.258, 0.282) | 0.934 (0.920, 0.945) |
| | | 50 | 4 | 5e-5 | 1e-4 | **0.037 (0.035, 0.045)** | **0.273 (0.267, 0.287)** | 0.909 (0.893, 0.926) |
| | Shear Flow | 50 | 3 | 1e-4 | 5e-4 | 0.0032 (0.0026, 0.0043) | 0.0616 (0.0516, 0.0752) | 0.8367 (0.7964, 0.9074) |
| | | 50 | 4 | 1e-4 | 5e-4 | **0.0031 (0.0024, 0.0041)** | **0.0601 (0.0481, 0.0736)** | 0.8388 (0.8067, 0.9142) |
| | | 50 | 3 | 5e-5 | 1e-4 | 0.0036 (0.0030, 0.0046) | 0.0652 (0.0490, 0.0813) | 0.8378 (0.7923, 0.8787) |
| | | 50 | 4 | 5e-5 | 1e-4 | 0.0034 (0.0027, 0.0039) | 0.0649 (0.0450, 0.0733) | 0.8263 (0.7932, 0.8923) |
| | | 50 | 3 | 1e-5 | 5e-5 | 0.0038 (0.0027, 0.0049) | 0.0662 (0.0520, 0.0876) | 0.7997 (0.7674, 0.8278) |
| | | 50 | 4 | 1e-5 | 5e-5 | 0.0037 (0.0029, 0.0046) | 0.0630 (0.0471, 0.0793) | 0.8081 (0.7734, 0.8465) |
| Lola | Rayleigh Benard | 100 | 2 | 1e-4 | 3e-4 | 0.056 (0.051, 0.061) | 0.634 (0.624, 0.645) | 0.979 (0.959, 0.999) |
| | | 100 | 4 | 1e-4 | 3e-4 | **0.055 (0.051, 0.060)** | **0.632 (0.621, 0.652)** | 0.957 (0.941, 0.978) |
| | | 100 | 2 | 3e-5 | 1e-4 | 0.057 (0.052, 0.061) | 0.645 (0.631, 0.657) | 0.978 (0.957, 0.995) |
| | | 100 | 4 | 3e-5 | 1e-4 | 0.056 (0.051, 0.062) | 0.635 (0.626, 0.642) | 0.968 (0.951, 0.985) |
| | | 100 | 2 | 3e-5 | 5e-5 | 0.057 (0.052, 0.061) | 0.647 (0.633, 0.657) | 0.986 (0.967, 1.009) |
| | | 100 | 4 | 3e-5 | 5e-5 | 0.056 (0.052, 0.060) | 0.644 (0.635, 0.654) | 0.971 (0.954, 0.994) |
| | Euler | 100 | 2 | 1e-4 | 3e-4 | 0.030 (0.028, 0.032) | 0.256 (0.248, 0.265) | 1.123 (1.114, 1.133) |
| | | 100 | 4 | 1e-4 | 3e-4 | 0.028 (0.027, 0.031) | 0.253 (0.243, 0.262) | 1.053 (1.043, 1.057) |
| | | 100 | 2 | 3e-5 | 1e-4 | 0.030 (0.028, 0.032) | 0.255 (0.246, 0.268) | 1.199 (1.191, 1.209) |
| | | 100 | 4 | 3e-5 | 1e-4 | **0.028 (0.027, 0.031)** | **0.250 (0.241, 0.260)** | 1.103 (1.097, 1.109) |
| | | 100 | 2 | 3e-5 | 5e-5 | 0.030 (0.028, 0.031) | 0.257 (0.248, 0.268) | 1.202 (1.195, 1.208) |
| | | 100 | 4 | 3e-5 | 5e-5 | 0.028 (0.027, 0.030) | 0.251 (0.241, 0.262) | 1.112 (1.105, 1.116) |
| | Shear Flow | 100 | 2 | 1e-4 | 3e-3 | 0.0061 (0.0055, 0.0078) | 0.1183 (0.0929, 0.1435) | 0.6205 (0.5980, 0.6629) |
| | | 100 | 4 | 1e-4 | 3e-3 | **0.0059 (0.0052, 0.0075)** | **0.1047 (0.0902, 0.1407)** | 0.6094 (0.5864, 0.6274) |
| | | 100 | 2 | 1e-4 | 1e-3 | 0.0058 (0.0053, 0.0077) | 0.1201 (0.1010, 0.1480) | 0.6856 (0.6413, 0.7245) |
| | | 100 | 4 | 1e-4 | 1e-3 | 0.0068 (0.0061, 0.0083) | 0.1084 (0.0870, 0.1350) | 0.4395 (0.3881, 0.4882) |
| | | 100 | 2 | 1e-4 | 3e-4 | 0.0072 (0.0062, 0.0093) | 0.1094 (0.0909, 0.1436) | 0.2639 (0.2267, 0.2875) |
| | | 100 | 4 | 1e-4 | 3e-4 | 0.0070 (0.0061, 0.0090) | 0.1093 (0.0874, 0.1359) | 0.3075 (0.2836, 0.3597) |
| | | 100 | 2 | 3e-5 | 3e-3 | 0.0068 (0.0058, 0.0079) | 0.1252 (0.1074, 0.1485) | 0.6479 (0.5968, 0.6821) |
| | | 100 | 4 | 3e-5 | 3e-3 | 0.0057 (0.0051, 0.0075) | 0.1235 (0.1013, 0.1512) | 0.7529 (0.7099, 0.8095) |

*Table 12.* Full quantitative metrics for retrofitting duration ablation. The **bolded** results correspond to the models used in the main paper. Ep. = Number of epochs, M = Number of training ensemble members, LR (B) = Backbone learning rate, LR (N) = Noise Branch learning rate

| Model | Dataset | Ep. | M | LR (B) | LR (N) | CRPS | VRMSE | SSR |
|---|---|---|---|---|---|---|---|---|
| Walrus | Rayleigh Benard | 50 | 3 | 5e-4 | 1e-3 | 0.055 (0.051, 0.061) | 0.627 (0.600, 0.649) | 0.968 (0.939, 0.985) |
| | | 100 | 3 | 5e-4 | 1e-3 | 0.051 (0.048, 0.058) | 0.587 (0.562, 0.632) | 0.962 (0.930, 0.990) |
| | | 50 | 3 | 1e-4 | 5e-4 | 0.050 (0.047, 0.054) | 0.583 (0.556, 0.606) | 0.916 (0.885, 0.956) |
| | | 100 | 3 | 1e-4 | 5e-4 | 0.051 (0.047, 0.055) | 0.573 (0.551, 0.625) | 0.909 (0.874, 0.932) |
| | | 50 | 4 | 5e-5 | 1e-4 | **0.050 (0.046, 0.053)** | **0.558 (0.539, 0.579)** | 0.900 (0.860, 0.921) |
| | | 100 | 4 | 5e-5 | 1e-4 | 0.051 (0.046, 0.054) | 0.557 (0.539, 0.595) | 0.902 (0.870, 0.924) |
| | Euler | 50 | 4 | 1e-4 | 5e-4 | 0.038 (0.035, 0.043) | 0.275 (0.260, 0.286) | 0.900 (0.878, 0.913) |
| | | 100 | 4 | 1e-4 | 5e-4 | 0.040 (0.036, 0.044) | 0.290 (0.275, 0.298) | 0.908 (0.892, 0.920) |
| | | 50 | 4 | 5e-5 | 1e-4 | **0.037 (0.035, 0.045)** | **0.273 (0.267, 0.287)** | 0.909 (0.893, 0.926) |
| | | 100 | 4 | 5e-5 | 1e-4 | 0.039 (0.035, 0.043) | 0.279 (0.265, 0.286) | 0.909 (0.895, 0.921) |
| | Shear Flow | 50 | 4 | 1e-4 | 5e-4 | **0.0031 (0.0024, 0.0041)** | **0.0601 (0.0481, 0.0736)** | 0.8388 (0.8067, 0.9142) |
| | | 100 | 4 | 1e-4 | 5e-4 | 0.0034 (0.0028, 0.0041) | 0.0618 (0.0462, 0.0803) | 0.8477 (0.8051, 0.9415) |
| | | 50 | 4 | 5e-5 | 1e-4 | 0.0034 (0.0027, 0.0039) | 0.0649 (0.0450, 0.0733) | 0.8263 (0.7932, 0.8923) |
| | | 100 | 4 | 5e-5 | 1e-4 | 0.0034 (0.0028, 0.0039) | 0.0640 (0.0461, 0.0808) | 0.8334 (0.8031, 0.8935) |

*Table 13.* Full quantitative metrics for the noise embedding dimension ablation. The **bolded** results correspond to the models used in the main paper. Ep. = Number of epochs, M = Number of training ensemble members, LR (B) = Backbone learning rate, LR (N) = Noise Branch learning rate.

| Model | Dataset | Ep. | M | LR (B) | LR (N) | Noise dim. | CRPS | VRMSE | Spread-Skill |
|---|---|---|---|---|---|---|---|---|---|
| Walrus | Rayleigh Benard | 50 | 4 | 5e-5 | 1e-4 | 32 | **0.050 (0.046, 0.053)** | **0.558 (0.539, 0.579)** | 0.900 (0.860, 0.921) |
| | | 50 | 4 | 5e-5 | 1e-4 | 16 | 0.051 (0.048, 0.054) | 0.576 (0.549, 0.606) | 0.885 (0.862, 0.921) |
| | Euler | 50 | 4 | 5e-5 | 1e-4 | 32 | **0.037 (0.035, 0.045)** | **0.273 (0.267, 0.287)** | 0.909 (0.893, 0.926) |
| | | 50 | 4 | 5e-5 | 1e-4 | 16 | 0.039 (0.035, 0.043) | 0.283 (0.272, 0.294) | 0.905 (0.890, 0.917) |
| | Shear Flow | 50 | 4 | 1e-4 | 5e-4 | 32 | **0.0031 (0.0024, 0.0041)** | **0.0601 (0.0481, 0.0736)** | 0.8388 (0.8067, 0.9142) |
| | | 50 | 4 | 1e-4 | 5e-4 | 16 | 0.0040 (0.0035, 0.0049) | 0.0674 (0.0532, 0.0905) | 0.8278 (0.7898, 0.8611) |
| Lola | Rayleigh Benard | 100 | 2 | 1e-4 | 3e-4 | 32 | 0.056 (0.051, 0.061) | 0.634 (0.624, 0.645) | 0.979 (0.959, 0.999) |
| | | 100 | 2 | 1e-4 | 3e-4 | 16 | 0.056 (0.052, 0.061) | 0.647 (0.626, 0.658) | 0.971 (0.953, 0.996) |
| | | 100 | 4 | 1e-4 | 3e-4 | 32 | **0.055 (0.051, 0.060)** | **0.632 (0.621, 0.652)** | 0.957 (0.941, 0.978) |
| | | 100 | 4 | 1e-4 | 3e-4 | 16 | 0.057 (0.053, 0.061) | 0.645 (0.627, 0.655) | 0.958 (0.932, 0.976) |
| | Euler | 100 | 2 | 1e-4 | 3e-4 | 32 | 0.030 (0.028, 0.032) | 0.256 (0.248, 0.265) | 1.123 (1.114, 1.133) |
| | | 100 | 2 | 1e-4 | 3e-4 | 16 | 0.030 (0.028, 0.032) | 0.259 (0.250, 0.270) | 1.200 (1.192, 1.208) |
| | | 100 | 4 | 1e-4 | 3e-4 | 32 | **0.028 (0.027, 0.031)** | **0.253 (0.243, 0.262)** | 1.053 (1.043, 1.057) |
| | | 100 | 4 | 1e-4 | 3e-4 | 16 | 0.029 (0.027, 0.031) | 0.251 (0.243, 0.261) | 1.098 (1.092, 1.104) |
| | Shear Flow | 100 | 2 | 1e-4 | 3e-3 | 32 | 0.0061 (0.0055, 0.0078) | 0.1183 (0.0929, 0.1435) | 0.6205 (0.5980, 0.6629) |
| | | 100 | 2 | 1e-4 | 3e-3 | 16 | 0.0061 (0.0051, 0.0073) | 0.1159 (0.0936, 0.1432) | 0.6887 (0.6513, 0.7234) |
| | | 100 | 4 | 1e-4 | 3e-3 | 32 | **0.0059 (0.0052, 0.0075)** | **0.1047 (0.0902, 0.1407)** | 0.6094 (0.5864, 0.6274) |
| | | 100 | 4 | 1e-4 | 3e-3 | 16 | 0.0070 (0.0056, 0.0087) | 0.1435 (0.1088, 0.1898) | 0.7584 (0.6937, 0.8261) |

*Table 14.* Full quantitative metrics for the noise injection strategy ablation. We compare injecting noise into all processor blocks (full) versus every second block (half). The **bolded** results correspond to the models used in the main paper. Ep. = Number of epochs, M = Number of training ensemble members, LR (B) = Backbone learning rate, LR (N) = Noise Branch learning rate.

| Model | Dataset | Ep. | M | LR (B) | LR (N) | Noise layers | CRPS | VRMSE | Spread-Skill |
|---|---|---|---|---|---|---|---|---|---|
| Walrus | Rayleigh Benard | 50 | 4 | 5e-5 | 1e-4 | full | **0.050 (0.046, 0.053)** | **0.558 (0.539, 0.579)** | 0.900 (0.860, 0.921) |
| | | 50 | 4 | 5e-5 | 1e-4 | half | 0.050 (0.046, 0.054) | 0.574 (0.552, 0.589) | 0.898 (0.879, 0.916) |
| | Euler | 50 | 4 | 5e-5 | 1e-4 | full | **0.037 (0.035, 0.045)** | **0.273 (0.267, 0.287)** | 0.909 (0.893, 0.926) |
| | | 50 | 4 | 5e-5 | 1e-4 | half | 0.039 (0.035, 0.042) | 0.277 (0.268, 0.288) | 0.901 (0.881, 0.917) |
| | Shear Flow | 50 | 4 | 1e-4 | 5e-4 | full | **0.0031 (0.0024, 0.0041)** | **0.0601 (0.0481, 0.0736)** | 0.8388 (0.8067, 0.9142) |
| | | 50 | 4 | 1e-4 | 5e-4 | half | 0.0031 (0.0026, 0.0039) | 0.0639 (0.0451, 0.0810) | 0.8253 (0.8042, 0.8592) |
| Lola | Rayleigh Benard | 100 | 4 | 1e-4 | 3e-4 | full | **0.055 (0.051, 0.060)** | **0.632 (0.621, 0.652)** | 0.957 (0.941, 0.978) |
| | | 100 | 4 | 1e-4 | 3e-4 | half | 0.057 (0.053, 0.060) | 0.641 (0.626, 0.649) | 0.956 (0.933, 0.975) |
| | Euler | 100 | 4 | 3e-5 | 1e-4 | full | **0.028 (0.027, 0.031)** | **0.250 (0.241, 0.260)** | 1.103 (1.097, 1.109) |
| | | 100 | 4 | 3e-5 | 1e-4 | half | 0.028 (0.026, 0.031) | 0.251 (0.243, 0.258) | 1.094 (1.088, 1.101) |
| | Shear Flow | 100 | 4 | 1e-4 | 3e-3 | full | **0.0059 (0.0052, 0.0075)** | **0.1047 (0.0902, 0.1407)** | 0.6094 (0.5864, 0.6274) |
| | | 100 | 4 | 1e-4 | 3e-3 | half | 0.0058 (0.0051, 0.0071) | 0.1172 (0.1003, 0.1416) | 0.7008 (0.6540, 0.7536) |

## E.2. Foundation Model - HalfWalrus

### E.2.1. ADDITIONAL NUMERICAL RESULTS

In conjunction with Figure 5, we present the full set of results in terms of rollout VRMSE, CRPS, and SSR in Table 15, as well as the improvement in energy score in Figure 21. Ours (CRPS) refers to the CRPS retrofitted model (starting from the deterministic foundation model), and Deterministic (FT) refers to a model fine-tuned with the same pre-training loss on a compute-equal budget with the CRPS model.

*Table 15.* Aggregated quantitative metrics (averaged across all fields). We show the median and the 95% confidence intervals over 100 bootstrapping iterations. Deterministic models are shaded in grey, while CRPS-retrofitted models are highlighted in purple. Best results are **bolded**.

| Model | Dataset | Method | CRPS | VRMSE | SSR |
|---|---|---|---|---|---|
| HalfWalrus | Rayleigh Benard | Deterministic (FT) | 0.105 (0.097, 0.119) | 0.850 (0.825, 0.885) | - |
| | | Ours (CRPS) | **0.068 (0.065, 0.073)** | **0.734 (0.693, 0.769)** | 0.964 (0.944, 0.992) |
| | TRL | Deterministic (FT) | 1.436 (1.044, 1.951) | 0.932 (0.858, 1.245) | - |
| | | Ours (CRPS) | **0.875 (0.694, 1.273)** | **0.780 (0.634, 0.828)** | 0.905 (0.783, 1.073) |
| | Viscoelastic transition + SAR | Deterministic (FT) | 4.952 (3.038, 7.055) | **0.131 (0.111, 0.157)** | - |
| | | Ours (CRPS) | **4.375 (2.645, 5.681)** | 0.144 (0.108, 0.192) | 1.348 (1.243, 1.434) |
| | Viscoelastic EIT | Deterministic (FT) | 124.011 (115.769, 175.592) | 0.736 (0.715, 1.021) | - |
| | | Ours (CRPS) | **98.624 (74.887, 125.376)** | **0.689 (0.583, 0.893)** | 0.723 (0.631, 0.879) |
| | Viscoelastic CAR | Deterministic (FT) | 148.468 (103.458, 160.618) | 0.822 (0.690, 0.931) | - |
| | | Ours (CRPS) | **100.351 (82.878, 103.076)** | **0.710 (0.681, 0.761)** | 0.750 (0.741, 0.865) |

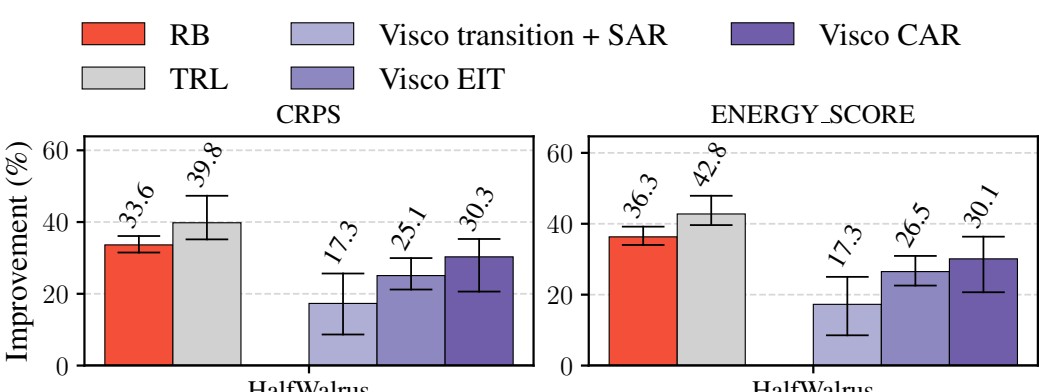

*Figure 21.* Improvement of CRPS retrofitting over deterministic fine-tuning for HalfWalrus: (Left) CRPS; (Right) Energy Score. Our method obtains similar percentage improvements in energy score as in CRPS, indicating that the improvements in univariate metrics translate into multivariate ones.

### E.2.2. EXAMPLE ROLLOUTS

Similarly to the single-system models, we provide example rollouts for each dataset to compare between the deterministically fine-tuned and CRPS retrofitted models.

- For RB, the CRPS-tuned samples demonstrate superior temporal fidelity compared to the deterministic baseline, particularly in capturing the correct rate of flow evolution. As seen in the $t = 30$ snapshots for the first two trajectories in Figure 22, the deterministic model exhibits inaccurate acceleration, evolving noticeably faster than the ground truth. Conversely, in the third trajectory at $t = 60$, the deterministic prediction appears "sluggish" and under-developed, while the CRPS samples maintain a pace consistent with the ground truth. At longer lead times, where the intrinsic uncertainty within the system makes it hard to predict the exact ground truth state, the CRPS ensemble provides an estimate of the system's uncertainty, with a spread that plausibly captures the range of potential outcomes.

- For TRL, model predictions begin to diverge from the ground truth even at short lead times ($t = 10$). The CRPS retrofitted model successfully captures this uncertainty through its ensemble. As the rollout progresses to longer

lead times, the deterministic baseline suffers from over-smoothing. In contrast, the CRPS samples maintain physical plausibility and exhibit a spread that effectively represents the range of potential valid outcomes.

- In the Viscoelastic (VI) dataset, we examine performance across four distinct flow regimes: two chaotic phases—Elasto-Inertial Turbulence (EIT) and the Chaotic Arrowhead Regime (CAR)—as well as the Steady Arrowhead Regime (SAR) and a transitional phase. In the chaotic regimes, the deterministic baseline exhibits significant over-smoothing at longer lead times. This is most striking in the CAR trajectory, where the deterministic prediction blurs substantially by $t = 45$. While the CRPS-tuned predictions do not perfectly recover the ground truth and show signs of under-dispersion in these highly non-linear cases, they remain physically consistent, retaining flow structures that the deterministic model smoothens out. Finally, in the stable SAR and transitional regimes, where the dynamics are less stochastic, both deterministic and probabilistic models adhere closely to the ground truth.

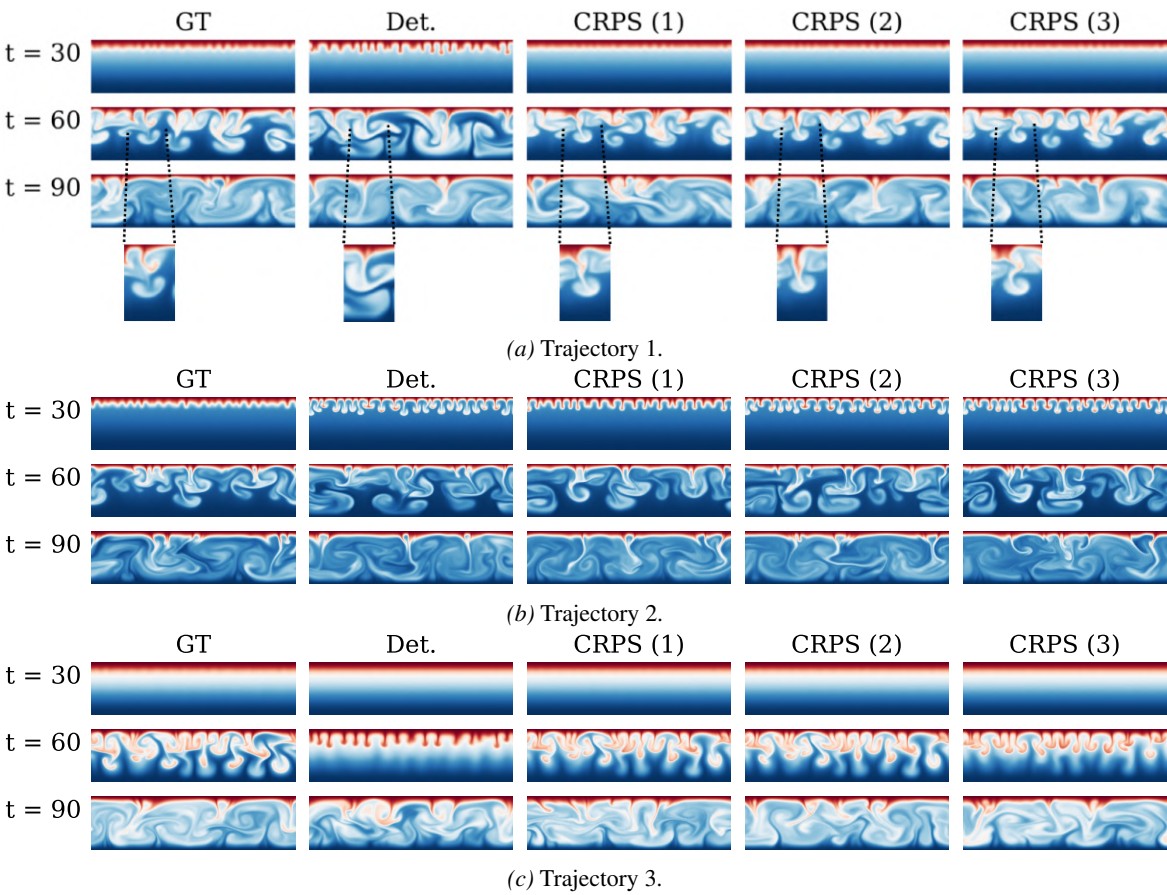

*(a)* Trajectory 1.

*(b)* Trajectory 2.

*(c)* Trajectory 3.

*Figure 22.* Three example of rollouts for the Rayleigh-Bénard dataset for HalfWalrus. We show the buoyancy channel. For the first trajectory we zoom in on a region to show how the CRPS samples better resemble the ground truth, while still showing some spread. For the second and third samples, the CRPS samples better manage to capture the speed of the flow, as opposed to the deterministic one which either accelerates too fast (in trajectory 2) or too slowly (in trajectory 3).

### E.2.3. Improvement with additional ensemble members

We observe the same test-time scaling behaviour as with the single-system models whereby increasing the ensemble members at inference improves the rollout VRMSE (normalised with respect to the single-member prediction). We also show the VRMSE evolution throughout the rollout for the three datasets in Figure 25.

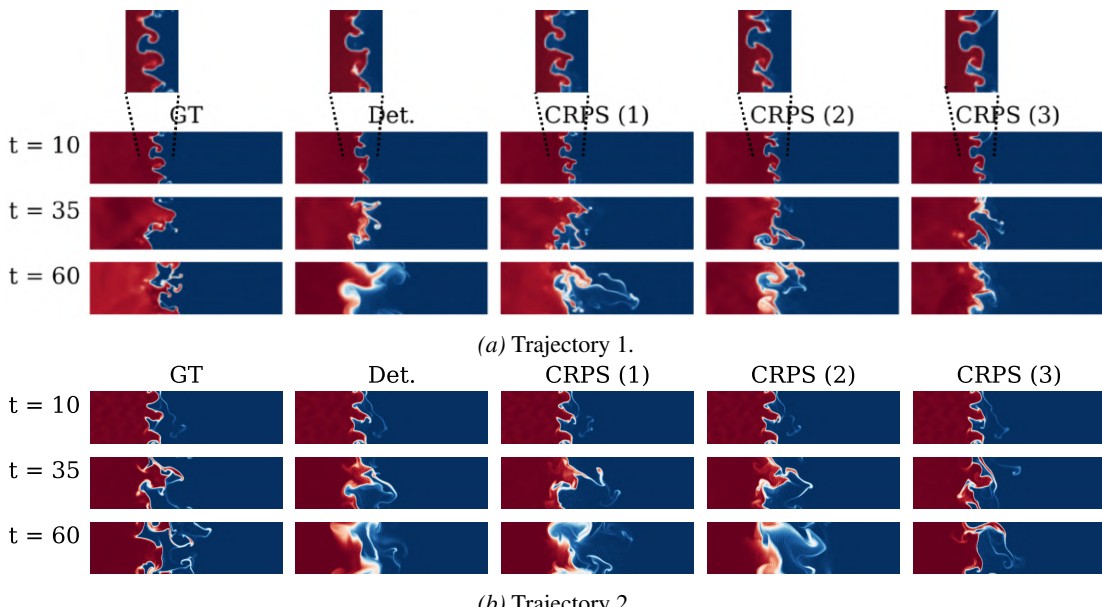

*(a)* Trajectory 1.

*(b)* Trajectory 2.

*Figure 23.* Two examples of rollouts for the TRL dataset for HalfWalrus. We show the density channel. We zoom into the the mixing area for the first trajectory at $t = 10$ to show how the CRPS samples show a good spread, and more detail than the deterministic prediction.

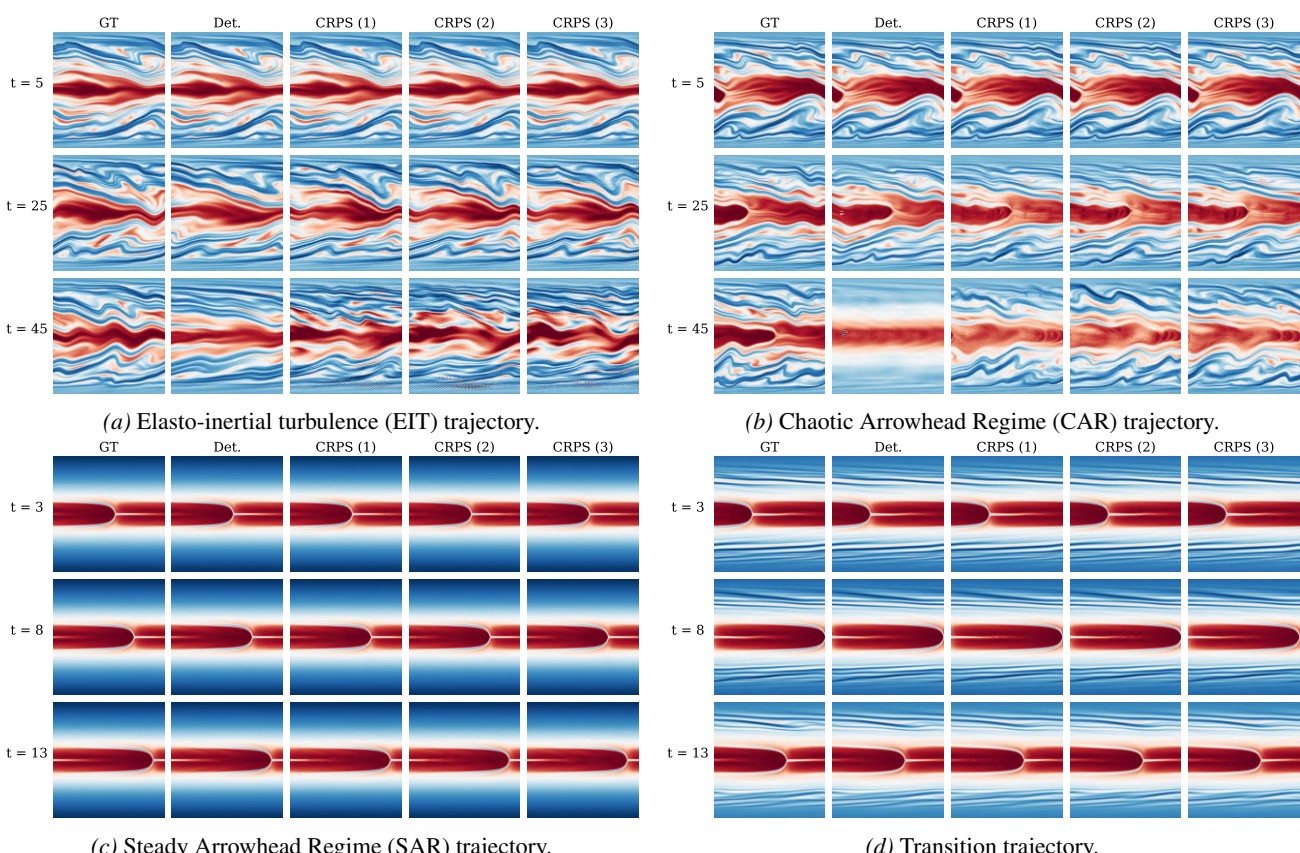

*(a)* Elasto-inertial turbulence (EIT) trajectory.

*(b)* Chaotic Arrowhead Regime (CAR) trajectory.

*(c)* Steady Arrowhead Regime (SAR) trajectory.

*(d)* Transition trajectory.

*Figure 24.* Four examples of rollouts for the viscoelastic dataset with four different types of regimes for HalfWalrus. We show the $c_{zz}$ channel and average . Note how the deterministic predictions over-smooth at longer rollout times for the chaotic regimes (EIT and CAR). For the other regimes, all predictions closely resemble the ground truth (SAR and transition).

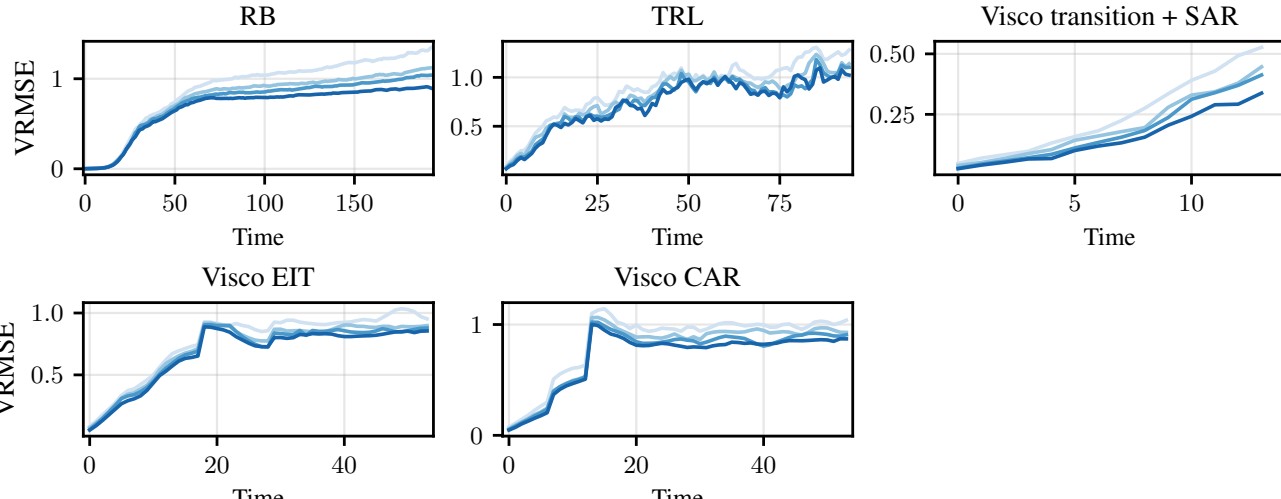

*Figure 25.* Evolution of VRMSE throughout rollout length for HalfWalrus with increasing ensemble size (1, 2, 4, and 16) indicated by darker shades of blue.

### E.2.4. ABLATION STUDIES

We additionally provide several ablations to study the effect of learning rate, retrofitting duration, and the number of layers in which noise is injected for HalfWalrus.

**Learning Rate**  We extend the learning rate analysis to the foundation model setting (HalfWalrus) to assess stability across diverse physical regimes. As shown in Table 16, the optimal configuration exhibits some dataset dependence. Specifically, the TRL dataset benefits from more aggressive learning rates (particularly in the noise branch), whereas the RB and VI datasets generally favour more conservative, lower learning rates to maintain stability. This sensitivity is further nuanced within the VI dataset itself: by splitting it into its three constituent flow regimes, we observe that the laminar regimes (transition + SAR) generally favour lower learning rates, whereas the chaotic regimes (EIT and CAR) benefit from increased noise branch adaptation. This heterogeneity makes selecting a single global hyperparameter configuration non-trivial. However, crucially, these performance differences are largely contained within overlapping confidence intervals across configurations. This suggests that, similar to the single-system setting, the method is relatively robust provided the hyperparameters are selected within a reasonable regime.

*Table 16.* Full quantitative metrics for the learning rate ablation for HalfWalrus. The **bolded** results correspond to the models used in the main paper. Ep. = Number of epochs, M = Number of training ensemble members, LR (B) = Backbone learning rate, LR (N) = Noise Branch learning rate.

| Model | Dataset | Ep. | M | LR (B) | LR (N) | CRPS | VRMSE | Spread-Skill |
|---|---|---|---|---|---|---|---|---|
| HalfWalrus | Rayleigh Benard | 50 | 2 | 1e-4 | 5e-4 | 0.070 (0.067, 0.076) | 0.784 (0.748, 0.803) | 0.963 (0.923, 1.010) |
| | | 50 | 2 | 1e-5 | 1e-4 | **0.068 (0.065, 0.073)** | **0.734 (0.693, 0.769)** | 0.964 (0.944, 0.992) |
| | TRL | 50 | 2 | 1e-4 | 1e-3 | **0.875 (0.694, 1.273)** | **0.780 (0.634, 0.828)** | 0.905 (0.783, 1.073) |
| | | 50 | 2 | 1e-4 | 5e-4 | 0.956 (0.602, 1.286) | 0.879 (0.732, 0.923) | 0.843 (0.720, 1.058) |
| | | 50 | 2 | 1e-5 | 1e-4 | 0.856 (0.655, 1.197) | 0.806 (0.792, 0.857) | 0.861 (0.758, 0.958) |
| | Viscoelastic transition + SAR | 50 | 2 | 5e-4 | 1e-3 | 5.060 (2.542, 6.651) | 0.167 (0.097, 0.176) | 1.376 (1.272, 1.503) |
| | | 50 | 2 | 1e-4 | 1e-3 | 4.269 (2.301, 6.044) | 0.138 (0.084, 0.196) | 1.365 (1.315, 1.561) |
| | | 50 | 2 | 1e-4 | 5e-4 | **4.375 (2.645, 5.681)** | **0.144 (0.108, 0.192)** | 1.348 (1.243, 1.434) |
| | | 50 | 2 | 1e-5 | 1e-4 | 4.847 (3.203, 6.558) | 0.176 (0.137, 0.225) | 1.383 (1.236, 1.444) |
| | Viscoelastic EIT | 50 | 2 | 5e-4 | 1e-3 | 91.645 (74.982, 120.763) | 0.692 (0.608, 0.886) | 0.875 (0.813, 0.987) |
| | | 50 | 2 | 1e-4 | 1e-3 | 98.987 (80.144, 129.475) | 0.688 (0.609, 0.920) | 0.760 (0.700, 0.931) |
| | | 50 | 2 | 1e-4 | 5e-4 | **98.624 (74.887, 125.376)** | **0.689 (0.583, 0.893)** | 0.723 (0.631, 0.879) |
| | | 50 | 2 | 1e-5 | 1e-4 | 96.328 (78.865, 126.558) | 0.680 (0.611, 0.913) | 0.799 (0.726, 0.924) |
| | Viscoelastic CAR | 50 | 2 | 5e-4 | 1e-3 | 94.020 (73.424, 96.004) | 0.715 (0.666, 0.743) | 0.899 (0.884, 1.087) |
| | | 50 | 2 | 1e-4 | 1e-3 | 100.178 (85.337, 105.315) | 0.726 (0.711, 0.770) | 0.824 (0.805, 0.925) |
| | | 50 | 2 | 1e-4 | 5e-4 | **100.351 (82.878, 103.076)** | **0.710 (0.681, 0.761)** | 0.750 (0.741, 0.865) |
| | | 50 | 2 | 1e-5 | 1e-4 | 99.668 (98.529, 100.769) | 0.727 (0.707, 0.757) | 0.807 (0.782, 0.853) |

**Retrofitting Duration**  We also examine the impact of extending the retrofitting duration from 50 to 100 epochs in the foundation model setting (HalfWalrus). As detailed in Table 17, the benefits of longer training are regime-dependent.

- **Consistent Gains in Stable Regimes:** For the RB dataset and the laminar regime of the VI dataset (transition + SAR), extending training consistently improves both rollout CRPS and VRMSE, suggesting that these dynamics benefit from further retrofitting.
- **Mixed Results in Complex Dynamics:** In the TRL dataset, results are mixed: while CRPS generally improves with longer training (provided the optimal learning rate regime is used), VRMSE stagnates or slightly degrades. We attribute this discrepancy to the inherent difficulty of the dataset and the potential limitations of VRMSE in capturing improvements in highly turbulent probabilistic predictions.
- **Saturation in Chaotic Regimes:** Conversely, for the chaotic VI regimes (EIT and CAR), performance tends to plateau, with metrics being within error bars.

Overall, while extended training offers benefits in specific stable regimes, 50 epochs represents a robust and resource-efficient compromise across the diverse physics covered by the foundation model. We anticipate that rather than simply increasing the duration of the current retrofitting phase, more significant gains in rollout metrics could be achieved by transitioning to rollout fine-tuning after the initial 50 epochs.

*Table 17.* Full quantitative metrics for the retrofitting duration ablation for HalfWalrus. The **bolded** results correspond to the models used in the main paper. Ep. = Number of epochs, M = Number of training ensemble members, LR (B) = Backbone learning rate, LR (N) = Noise Branch learning rate.

| Model | Dataset | Ep. | M | LR (B) | LR (N) | CRPS | VRMSE | Spread-Skill |
|---|---|---|---|---|---|---|---|---|
| HalfWalrus | Rayleigh Benard | 50 | 2 | 1e-4 | 5e-4 | 0.070 (0.067, 0.076) | 0.784 (0.748, 0.803) | 0.963 (0.923, 1.010) |
| | | 100 | 2 | 1e-4 | 5e-4 | 0.065 (0.060, 0.069) | 0.686 (0.674, 0.717) | 0.941 (0.916, 0.981) |
| | | 50 | 2 | 1e-5 | 1e-4 | **0.068 (0.065, 0.073)** | **0.734 (0.693, 0.769)** | 0.964 (0.944, 0.992) |
| | | 100 | 2 | 1e-5 | 1e-4 | 0.062 (0.057, 0.066) | 0.700 (0.661, 0.730) | 0.952 (0.927, 1.023) |
| | TRL | 50 | 2 | 1e-4 | 1e-3 | **0.875 (0.694, 1.273)** | **0.780 (0.634, 0.828)** | 0.905 (0.783, 1.073) |
| | | 100 | 2 | 1e-4 | 1e-3 | 0.838 (0.642, 1.418) | 0.847 (0.713, 0.967) | 0.962 (0.703, 1.070) |
| | | 50 | 2 | 1e-4 | 5e-4 | 0.956 (0.602, 1.286) | 0.879 (0.732, 0.923) | 0.843 (0.720, 1.058) |
| | | 100 | 2 | 1e-4 | 5e-4 | 0.855 (0.694, 1.359) | 0.882 (0.746, 0.902) | 0.919 (0.718, 1.008) |
| | | 50 | 2 | 1e-5 | 1e-4 | 0.856 (0.655, 1.197) | 0.806 (0.792, 0.857) | 0.861 (0.758, 0.958) |
| | | 100 | 2 | 1e-5 | 1e-4 | 0.864 (0.657, 1.149) | 0.773 (0.752, 0.805) | 0.900 (0.842, 1.031) |
| | Viscoelastic transition + SAR | 50 | 2 | 1e-4 | 5e-4 | **4.375 (2.645, 5.681)** | **0.144 (0.108, 0.192)** | 1.348 (1.243, 1.434) |
| | | 100 | 2 | 1e-4 | 5e-4 | 3.533 (1.931, 4.646) | 0.120 (0.077, 0.154) | 1.382 (1.281, 1.450) |
| | | 50 | 2 | 1e-5 | 1e-4 | 4.847 (3.203, 6.558) | 0.176 (0.137, 0.225) | 1.383 (1.236, 1.444) |
| | | 100 | 2 | 1e-5 | 1e-4 | 4.665 (2.722, 6.225) | 0.180 (0.120, 0.224) | 1.422 (1.280, 1.489) |
| | Viscoelastic EIT | 50 | 2 | 1e-4 | 5e-4 | **98.624 (74.887, 125.376)** | **0.689 (0.583, 0.893)** | 0.723 (0.631, 0.879) |
| | | 100 | 2 | 1e-4 | 5e-4 | 99.492 (77.002, 134.864) | 0.677 (0.597, 0.976) | 0.765 (0.751, 1.036) |
| | | 50 | 2 | 1e-5 | 1e-4 | 96.328 (78.865, 126.558) | 0.680 (0.611, 0.913) | 0.799 (0.726, 0.924) |
| | | 100 | 2 | 1e-5 | 1e-4 | 98.619 (80.226, 127.975) | 0.697 (0.631, 0.936) | 0.867 (0.806, 1.027) |
| | Viscoelastic CAR | 50 | 2 | 1e-4 | 5e-4 | **100.351 (82.878, 103.076)** | **0.710 (0.681, 0.761)** | 0.750 (0.741, 0.865) |
| | | 100 | 2 | 1e-4 | 5e-4 | 103.684 (88.666, 107.375) | 0.722 (0.697, 0.827) | 0.907 (0.797, 0.997) |
| | | 50 | 2 | 1e-5 | 1e-4 | 99.668 (98.529, 100.769) | 0.727 (0.707, 0.757) | 0.807 (0.782, 0.853) |
| | | 100 | 2 | 1e-5 | 1e-4 | 99.395 (96.509, 101.177) | 0.724 (0.713, 0.771) | 0.893 (0.823, 0.947) |

**Number of layers with noise injection** Finally, we assess the sensitivity of the foundation model (HalfWalrus) to the density of noise injection layers. Comparing the default "full" injection strategy against a "half" variant (injecting noise only in alternate blocks) reveals a nuanced trade-off, as shown in Table 18.

- **Performance Stability:** In the RB dataset, the difference between configurations is negligible across both CRPS and VRMSE.
- **Metric-Specific Degradation:** For the TRL dataset, while CRPS remains stable, the sparse injection leads to a noticeable degradation in VRMSE (increasing from 0.780 to 0.828) and a slight reduction in the Spread-Skill Ratio (SSR).
- **Consistency in VI:** Across the VI regimes, the "full" configuration consistently yields slightly lower CRPS values, although VRMSE remains largely unaffected.

Overall, the "half" configuration appears to be a good candidate for a more parameter-efficient alternative. However, its exact utility is likely dataset-dependent and contingent on the optimisation of other hyperparameters

*Table 18.* Full quantitative metrics for the noise injection layers ablation for HalfWalrus. The **bolded** results correspond to the models used in the main paper. Ep. = Number of epochs, M = Number of training ensemble members, LR (B) = Backbone learning rate, LR (N) = Noise Branch learning rate.

| Model | Dataset | Ep. | M | LR (B) | LR (N) | Noise layers | CRPS | VRMSE | Spread-Skill |
|---|---|---|---|---|---|---|---|---|---|
| HalfWalrus | Rayleigh Benard | 50 | 2 | 1e-5 | 1e-4 | full | **0.068 (0.065, 0.073)** | **0.734 (0.693, 0.769)** | 0.964 (0.944, 0.992) |
| | | 50 | 2 | 1e-5 | 1e-4 | half | 0.067 (0.064, 0.072) | 0.726 (0.693, 0.766) | 0.936 (0.885, 0.974) |
| | TRL | 50 | 2 | 1e-4 | 1e-3 | full | **0.875 (0.694, 1.273)** | **0.780 (0.634, 0.828)** | 0.905 (0.783, 1.073) |
| | | 50 | 2 | 1e-4 | 1e-3 | half | 0.878 (0.671, 1.265) | 0.828 (0.788, 0.854) | 0.895 (0.761, 0.982) |
| | Viscoelastic transition + SAR | 50 | 2 | 1e-4 | 5e-4 | full | **4.375 (2.645, 5.681)** | **0.144 (0.108, 0.192)** | 1.348 (1.243, 1.434) |
| | | 50 | 2 | 1e-4 | 5e-4 | half | 4.437 (2.302, 6.211) | 0.145 (0.101, 0.212) | 1.368 (1.256, 1.565) |
| | Viscoelastic EIT | 50 | 2 | 1e-4 | 5e-4 | full | **98.624 (74.887, 125.376)** | **0.689 (0.583, 0.893)** | 0.723 (0.631, 0.879) |
| | | 50 | 2 | 1e-4 | 5e-4 | half | 99.630 (75.185, 129.481) | 0.691 (0.581, 0.928) | 0.726 (0.644, 0.948) |
| | Viscoelastic CAR | 50 | 2 | 1e-4 | 5e-4 | full | **100.351 (82.878, 103.076)** | **0.710 (0.681, 0.761)** | 0.750 (0.741, 0.865) |
| | | 50 | 2 | 1e-4 | 5e-4 | half | 100.664 (86.817, 104.955) | 0.729 (0.703, 0.769) | 0.806 (0.793, 0.878) |

## E.3. Comparison to Probabilistic Baselines

### E.3.1. COMPARISON TO DEEP ENSEMBLES

To isolate the value of the CRPS objective, we compared our method against a Deterministic Deep Ensemble (fine-tuning 4 separate times with different initialisations), with results shown in Table 19. Despite Deep Ensembles requiring 4x the compute of a single CRPS run, we found:

- Superior Accuracy: CRPS-retrofitted models significantly outperform Deep Ensembles in both VRMSE and CRPS across datasets.

- Better Calibration: Deep Ensembles exhibit underdispersion. While CRPS models are not perfectly calibrated, their spread-to-skill ratio is much closer to 1 throughout the trajectory length.

This confirms the performance gains are largely driven by the CRPS objective itself, not just the presence of an ensemble.

*Table 19.* Results (VRMSE, CRPS, Spread to Skill ratio, and Energy Score) for the comparison between deterministic fine-tuning, deterministic deep ensembles, and CRPS retrofitting.

| Model | Dataset | Method | VRMSE | CRPS | Spread-Skill | Energy Score |
|---|---|---|---|---|---|---|
| Walrus | Rayleigh Benard | Deterministic (FT) | 0.711 (0.677, 0.736) | 0.086 (0.083, 0.091) | 0.164 (0.154, 0.167) | 65.512 (64.748, 69.038) |
| | | Deterministic Deep Ensemble | 0.589 (0.576, 0.602) | 0.051 (0.049, 0.053) | 0.829 (0.818, 0.856) | 38.771 (37.322, 40.504) |
| | | Ours (CRPS) | 0.553 (0.543, 0.572) | 0.050 (0.048, 0.052) | 0.905 (0.887, 0.921) | 39.024 (37.210, 40.174) |
| | Euler | Deterministic (FT) | 0.387 (0.377, 0.395) | 0.081 (0.077, 0.087) | 0.018 (0.017, 0.018) | 186.121 (179.344, 194.326) |
| | | Deterministic Deep Ensemble | 0.369 (0.362, 0.376) | 0.062 (0.059, 0.064) | 0.653 (0.645, 0.661) | 133.687 (129.344, 137.870) |
| | | Ours (CRPS) | 0.273 (0.270, 0.280) | 0.038 (0.037, 0.041) | 0.908 (0.898, 0.917) | 91.050 (86.143, 95.162) |
| Lola | Rayleigh Benard | Deterministic (FT) | 0.807 (0.800, 0.819) | 0.101 (0.099, 0.103) | 0.119 (0.115, 0.122) | 36.067 (35.020, 36.690) |
| | | Deterministic Deep Ensemble | 0.700 (0.690, 0.712) | 0.059 (0.058, 0.061) | 0.794 (0.787, 0.803) | 20.891 (20.452, 21.243) |
| | | Ours (CRPS) | 0.635 (0.631, 0.642) | 0.055 (0.054, 0.058) | 0.948 (0.937, 0.956) | 19.574 (18.936, 20.428) |
| | Euler | Deterministic (FT) | 0.285 (0.280, 0.291) | 0.045 (0.043, 0.047) | 0.019 (0.018, 0.019) | 42.886 (41.645, 43.932) |
| | | Deterministic Deep Ensemble | 0.274 (0.269, 0.279) | 0.036 (0.035, 0.038) | 0.401 (0.398, 0.405) | 32.912 (31.936, 33.816) |
| | | Ours (CRPS) | 0.250 (0.245, 0.256) | 0.029 (0.028, 0.030) | 1.107 (1.102, 1.112) | 25.842 (25.014, 26.582) |

### E.3.2. COMPARISON TO DIFFUSION

In the main manuscript, we hypothesise that, unlike alternative probabilistic retrofitting methods such as diffusion or flow matching, our proposed CRPS-based procedure is better positioned to efficiently leverage the representations within a deterministic checkpoint. This is primarily due to the similarity in training objectives: the CRPS objective is significantly closer to the MAE/MSE objectives used for deterministic training, whereas diffusion models are trained on a distinct denoising task. To explore the differences between these two retrofitting paradigms, we compare our method against a diffusion-based approach within the Lola modelling framework.

Crucially, out of the three model families considered, diffusion is likely to be the most effective in Lola due to its compressed latent space, which mitigates the high computational cost typically associated with pixel-space diffusion. To ensure a strong baseline, we train the diffusion model on a compute-matched budget. This effectively allows the diffusion model to see $4\times$ more samples than the CRPS model (given our ensemble size of $M = 4$), providing it with a significant advantage. We generally use the hyperparameters outlined in Table 5, but increase the batch size to 256 (following Rozet et al. 2025) and employ a learning rate of $5 \times 10^{-4}$.

**Retrofitting Performance Comparison**

**Competitive Accuracy with Faster Inference** As illustrated in Table 20, the retrofitted diffusion model offers no significant improvement over our CRPS retrofitted method on two out of the three datasets (Rayleigh Benard and Euler), while performing significantly worse on the third. Crucially, even in the cases where the predictive performance is statistically matched, the computational burden is not. Diffusion models require iterative denoising steps during inference, whereas our CRPS-based method acts as a single-step predictor, offering faster inference. In this case, we follow Rozet et al. 2025 and employ 16 steps of the 3rd order Adams-Bashforth multi-step integration method (Zhang & Chen, 2023), implying $48\times$ more forward passes through the latent-space emulator.

**Robustness to Dataset Heterogeneity** On the third dataset (Shear Flow), diffusion achieves significantly worse performance, with a rollout CRPS of 0.0084 compared to 0.0059 for our method. This highlights a key limitation of the diffusion-based approach: when applied to datasets with high heterogeneity and slower convergence rates, it becomes less data-efficient than CRPS retrofitting. We attribute this to the diffusion model's inability to efficiently repurpose the pre-learnt representations from the deterministic checkpoint. This limitation is particularly concerning for the retrofitting of foundation models, where downstream tasks are inherently diverse and robustness to varying dataset characteristics is crucial.

Overall, while diffusion retrofitting is competitive on two out of three datasets, it presents two major downsides: 1) increased inference cost, and 2) decreased robustness to dataset characterised conditioned on a fixed compute budget. We derive these conclusions despite applying the diffusion baseline in its most advantageous setting—a compressed latent space where training costs are manageable. We expect the case for diffusion to weaken further when considering pixel-space foundation models, where training and inference costs would become prohibitive. This observation aligns with recent trends in medium-range weather forecasting, where state-of-the-art systems are shifting away from pure generative approaches (e.g., GenCast; Price et al. 2025) to CRPS-minimising objectives to maintain operational feasibility (Alet et al., 2025).

*Table 20.* Quantitative metrics for the comparison between CRPS-based and diffusion-based retrofitting.

| Model | Dataset | Method | CRPS | VRMSE | Spread-Skill |
|---|---|---|---|---|---|
| Lola | Rayleigh Benard | Ours - CRPS (Retrofitted) | 0.055 (0.051, 0.060) | 0.632 (0.621, 0.652) | 0.957 (0.941, 0.978) |
| | | Diffusion (Retrofitted) | 0.055 (0.050, 0.057) | 0.631 (0.617, 0.645) | 0.941 (0.930, 0.946) |
| | Euler | Ours - CRPS (Retrofitted) | 0.028 (0.027, 0.031) | 0.250 (0.241, 0.260) | 1.103 (1.097, 1.109) |
| | | Diffusion (Retrofitted) | 0.029 (0.027, 0.032) | 0.254 (0.244, 0.264) | 0.886 (0.882, 0.893) |
| | Shear Flow | Ours - CRPS (Retrofitted) | 0.0059 (0.0052, 0.0075) | 0.1047 (0.0902, 0.1407) | 0.6094 (0.5864, 0.6274) |
| | | Diffusion (Retrofitted) | 0.0084 (0.0071, 0.0107) | 0.1740 (0.1566, 0.2315) | 1.0314 (0.9460, 1.0842) |

**Effectiveness in leveraging the deterministic representations** Finally, we perform an experiment to probe the intuition that CRPS-retrofitted models are more efficient at leveraging pre-trained deterministic representations. We train two variations of each model for the same duration: one initialised from the deterministic checkpoint, and one randomly initialised (trained from scratch).

**CRPS retrofitting benefits from initialisation** As illustrated in Table 21, CRPS retrofitting initialised from the deterministic checkpoint yields significantly better performance than random initialisation. This confirms that the CRPS objective effectively leverages the features learnt by the deterministic model, accelerating convergence and improving performance.

*Table 21.* Quantitative metrics for the comparison between retrofitting and training from scratch the CRPS model.

| Model | Dataset | Method | CRPS | VRMSE | Spread-Skill |
|---|---|---|---|---|---|
| Lola | Rayleigh Benard | Ours - CRPS (Retrofitted) | 0.055 (0.051, 0.060) | 0.632 (0.621, 0.652) | 0.957 (0.941, 0.978) |
| | | CRPS (Scratch) | 0.062 (0.058, 0.065) | 0.692 (0.681, 0.703) | 1.246 (1.214, 1.297) |
| | Euler | Ours - CRPS (Retrofitted) | 0.028 (0.027, 0.031) | 0.250 (0.241, 0.260) | 1.103 (1.097, 1.109) |
| | | CRPS (Scratch) | 0.048 (0.045, 0.052) | 0.373 (0.361, 0.386) | 1.017 (1.011, 1.025) |
| | Shear Flow | Ours - CRPS (Retrofitted) | 0.0059 (0.0052, 0.0075) | 0.1047 (0.0902, 0.1407) | 0.6094 (0.5864, 0.6274) |
| | | CRPS (Scratch) | 0.0894 (0.0817, 0.0967) | 0.7223 (0.6474, 1.1292) | 0.1207 (0.1034, 0.1558) |

**Diffusion struggles to leverage initialisation** In contrast, Table 22 shows that the retrofitted diffusion model performs similarly to the version trained from scratch in two out of three cases (RB and Euler). This suggests that the denoising objective is distinct enough from the deterministic forecasting task that the model cannot easily leverage the pre-trained weights. While there is a benefit in Shear Flow—likely because the deterministic checkpoint is already highly accurate in laminar regimes—the final performance remains significantly inferior to the CRPS-retrofitted counterpart, and even to the initial deterministic checkpoint (rollout CRPS of 0.0084 for the diffusion retrofitted model in comparison to 0.0079 for the deterministic base model).

*Table 22.* Quantitative metrics for the comparison between retrofitting and training from scratch the diffusion model.

| Model | Dataset | Method | CRPS | VRMSE | Spread-Skill |
|---|---|---|---|---|---|
| | Rayleigh Benard | Diffusion (Retrofitted) | 0.055 (0.050, 0.057) | 0.631 (0.617, 0.645) | 0.941 (0.930, 0.946) |
| | | Diffusion (Scratch) | 0.053 (0.050, 0.056) | 0.619 (0.605, 0.630) | 0.954 (0.941, 0.969) |
| | Euler | Diffusion (Retrofitted) | 0.029 (0.027, 0.032) | 0.254 (0.244, 0.264) | 0.886 (0.882, 0.893) |
| | | Diffusion (Scratch) | 0.030 (0.028, 0.032) | 0.265 (0.254, 0.279) | 0.969 (0.962, 0.974) |
| | Shear Flow | Diffusion (Retrofitted) | 0.0084 (0.0071, 0.0107) | 0.1740 (0.1566, 0.2315) | 1.0314 (0.9460, 1.0842) |
| | | Diffusion (Scratch) | 0.0127 (0.0101, 0.0188) | 0.3484 (0.3201, 0.3848) | 0.8066 (0.7410, 0.8887) |

