# OpenReview forum: "Probabilistic Retrofitting of Learned Simulators"
_ICML.cc/2026/Conference — ICML 2026 regular_

### Official Review · Reviewer_ZhQh · 2026-03-03

**Soundness:** 2
**Presentation:** 2
**Significance:** 2
**Originality:** 2
**Overall Recommendation:** 4
**Confidence:** 3

**Summary:**

This paper studies how to retrofit pretrained deterministic PDE surrogate models into probabilistic forecasters without training a generative model from scratch. The method injects learned stochastic modulation into the pretrained backbone via a noise-conditioned AdaLN mechanism and fine-tunes the model with the CRPS objective to produce ensembles of plausible rollouts. Experiments on various backbones and dynamical systems show the efficacy of their approach.

**Compliance With Llm Reviewing Policy:**

Affirmed.

**Final Justification:**

Taking into account the authors' rebuttal and other reviewers' view points, I am happy to recommend a weak accept. I still maintain my point about novelty, but I would prefer a paper with useful results without overly penalizing for novelty.

**Key Questions For Authors:**

See weakness

**Strengths And Weaknesses:**

Strengths:
- The approach is relatively modular -- introduce stochastic modulation via a noise conditioned normalization (AdaLN-style), then optimize a distributional scoring rule (CRPS).

- The experiments cover multiple backbones and multiple dynamical systems and demonstrate CRPS improvements across various settings

Weakness:
- This work has limited novelty -- the core recipe (stochastic conditioning + CRPS/fair CRPS training on top of a pretrained deterministic model) is very close to prior work, especially Alet et al. As a result, the main contribution reads more as an empirical transfer to PDE surrogate backbones than a new conceptual method.

- Improvements are much clearer on CRPS than on VRMSE, so gains in point-forecast quality are less convincing. I would like to see a standard RMSE/MAE reported as well for easier interpretation and comparability against determinstic models. The authors are free to choose their method for point prediction.

- The method relies on a particular AdaLN modulation scheme and specific choices of where to inject noise. The results seem to show that their chosen strategy works, but it does not convincingly establish that it is the best or that gains are not achievable with simpler alternatives (input concatenation, per-layer noise without AdaLN, etc.).

- Most comparisons are against deterministic fine-tuning. That is necessary but not sufficient to justify the method as the best retrofit strategy. At minimum, the paper should include one or more low-overhead probabilistic baselines that are natural competitors; for example: deep ensembles of independently fine-tuned deterministic models or dropout-based stochasticity.

---

> ### Author Rebuttal · Authors · 2026-03-29
>
> We thank the reviewer for recognising our method's modularity and its demonstrated CRPS improvements across various settings.
>
> W1. **Limited novelty** - While the underlying strategy exists in literature [1], its application to retrofitting pre-trained PDE foundation models is novel. As noted by Reviewer pKBT, this constitutes "a significant contribution" as the PDE community shifts toward deterministically trained foundation models. Our contribution is twofold:
>
> - Introducing CRPS training to the broader PDE modelling community.
>
> - Adapting it specifically to leverage massive, computationally expensive pre-trained representations, rather than training from scratch.
>
> W2. **Additional metrics: RMSE**
> For chaotic systems where long-term point-wise prediction is fundamentally limited, we believe point-wise metrics alone do not offer a comprehensive evaluation. This is why we additionally report the CRPS. Furthermore, as detailed in our response to Reviewer 2TFH (W1 and Q3), we have now included the Energy Score (multivariate generalisation of CRPS) and frequency-space metrics, which are evaluate the spectral properties of PDEs.
>
> For point-wise metrics, we selected Variance-Scaled RMSE (VRMSE) following The Well [2]. VRMSE makes errors dimensionless and scale-invariant, which is mathematically critical for PDEs where interacting fields have very different magnitudes. To show that similar conclusions hold for the RMSE, we show a comparison between the % improvement in VRMSE and RMSE for [single-systems](https://anonymous.4open.science/r/ICML_Probabilistic_Retrofitting_CRPS-2EE5/VRMSE_RMSE_Single_Systems.png) and [HalfWalrus](https://anonymous.4open.science/r/ICML_Probabilistic_Retrofitting_CRPS-2EE5/VRMSE_RMSE_HalfWalrus.png). Numerical results for [single-systems](https://anonymous.4open.science/r/ICML_Probabilistic_Retrofitting_CRPS-2EE5/Table3_RMSE_VrMSE.png) and [HalfWalrus](https://anonymous.4open.science/r/ICML_Probabilistic_Retrofitting_CRPS-2EE5/Table4_RMSE_VRMSE_HalfWalrus.png) are attached in the links.
>
> W3. **Noise injection mechanism** - We appreciate the reviewer's point. While we do not claim AdaLN is the only viable noise injection mechanism, we selected it because it is the established state-of-the-art for conditioning transformer-based generative models (e.g., DiT). Simpler alternatives might struggle in deep architectures because the global stochastic signal tends to "wash out" in deeper layers. AdaLN solves this by ensuring the noise vector actively modulates the feature space throughout the entire network.
>
> Because of these reasons, we chose to perform a rigorous ablation of the AdaLN strategy rather than testing a wide array of potentially sub-optimal mechanisms. As noted by other reviewers, we thoroughly investigated the most critical hyperparameters of this approach, including where the noise should be injected (Table 14) and its dimensionality (Table 13).
>
> We also note that concurrent literature [3] is exploring alternative conditioning mechanisms (e.g., SwiGLU layer biases). This shares our philosophy of producing ensemble forecasts via learned functional perturbations. While benchmarking these emerging alternatives against AdaLN is an excellent direction for future work, the proven success of AdaLN in diffusion models made it the most robust choice for our method.
>
> **W4. Deep Ensemble Baseline**
> Thank you for the suggestion, we agree that comparing to a deep ensemble is a great addition to the paper. We fine-tuned a deep ensemble of 4 deterministic models and, despite this requiring 4$\times$ the computational resources of a single CRPS retrofitting run, we found:
>
> - Superior Accuracy: CRPS-retrofitted models significantly outperform Deep Ensembles in both VRMSE and CRPS across datasets (e.g., Walrus on Euler achieves a 37% CRPS improvement over the Deep Ensemble) - see [Figure 3](https://anonymous.4open.science/r/ICML_Probabilistic_Retrofitting_CRPS-2EE5/Deep_Ens_VRMSE_CRPS.png).
>
> - Better Calibration: Deep Ensembles exhibit underdispersion. While CRPS-retrofitted models are not perfectly calibrated, their spread-to-skill ratio is much closer to 1 throughout the trajectory length (see [Table 1](https://anonymous.4open.science/r/ICML_Probabilistic_Retrofitting_CRPS-2EE5/Table1_Deep_Ens_metrics.png)).
>
> This confirms the performance gains are largely driven by the CRPS objective itself, not just the presence of an ensemble.
>
> We hope the additional results addressed your concerns, and we would be grateful if you reconsidered your score in the light of the new evidence.
>
> **References**
> 1. Alet, F. et al. (2025). Skillful joint probabilistic weather forecasting from marginals.
> 2. Ohana, R. et al. (2025). The well: a large-scale collection of diverse physics simulations for machine learning. Advances in Neural Information Processing Systems.
> 3. Zhdanov, M. et al. (2026). (Sparse) Attention to the Details: Preserving Spectral Fidelity in ML-based Weather Forecasting Models.

---

> > ### Author Rebuttal · Reviewer_ZhQh · 2026-04-01
> >
> > Taking into account the authors' rebuttal and other reviewers' view points, I am happy to recommend a weak accept. I still maintain my point about novelty, but I would prefer a paper with useful results without overly penalizing for novelty.

---

> > > ### Author Response · Authors · 2026-04-05
> > >
> > > We are very grateful for your reassessment of our work and would like to thank the reviewer for raising their score and recommending our paper for acceptance. We are glad you agree that the empirical results and insights will be useful to the community.
> > >
> > > Thank you again for your time and efforts in evaluating our manuscript!

---

### Official Review · Reviewer_pKBT · 2026-03-11

**Soundness:** 4
**Presentation:** 4
**Significance:** 3
**Originality:** 3
**Overall Recommendation:** 5
**Confidence:** 3

**Summary:**

The paper focuses on the problem of PDE modeling. It takes a probabilistic approach, i.e. the future prediction of system is a distribution itself. Different to previous probabilistic approaches leveraging generative models e.g. flow matching or diffusion models, the focus here is on fine-tuning existing checkpoints of deterministic models.  Therefore, the generative model is parameterized as $F(x,\epsilon)$ where $F$ is a modified pre-trained model and $\epsilon$ is noise that is added as an additional input to the network. Here, noise is injected into the architecture via an architectural modification akin to Adaptive Layer Normalization (AdaLN) in the context of Diffusion Transformers. The network is then trained using scoring rules, in particular Continuous Ranked Probability Score (CRPS). Experiments fine-tuning various pre-trained models on various tasks are presented.

**Compliance With Llm Reviewing Policy:**

Affirmed.

**Final Justification:**

The authors addressed my concerns. The (minor) weaknesses - that would make me increase my rating - were not addressed but would also be out-of-scope for a rebuttal. I confirm my recommendation for acceptance.

**Key Questions For Authors:**

- How is the probabilistic nature taken into account for evaluation? 4.5 explains it to an extent but it would be great if you could make it clearer. Shouldn’t a distributional metric be used (i.e. one that measures the actual distance between distributions as opposed to just a normalized distance of the means)? I have limited expertise in the specific domains considered in the experiments. So I would appreciate clarifications for a wider ML audience.
- Did the authors observe any computational problems while training with scoring rules? How as the large batch size addressed?

**Limitations:**

Yes.

**Strengths And Weaknesses:**

**Strengths:**
- Well-written work and good presentation. It is easy to follow.
- To my knowledge, the method is novel and can be considered a significant contribution.
- The method is sound and the main claim of the work - that fine-tuning a checkpoint by minimizing the CRPS is a minimal, yet effective change of making existing large deterministic models probabilistic - makes sense.
- Extensive experiments: Experiments on various models and datasets are presented highlighting the advantage of the approach.

**Weaknesses:**
- Scoring rules require large sample sizes usually and are computationally expensive (high batch sizes M are used). Some of the limitations described in the experiments section might be explained by this. Further, learning to sample via scoring rules is known to lead to suboptimal results in the context of generative modeling. It is an open question whether this can be resolved or whether the method might always lead to a degradation in performance compared to other probabilistic approaches.
- The methods presented are a combination of well-known components and not in themselves novel. I do not consider this a strong weakness, however.

---

> ### Author Rebuttal · Authors · 2026-03-29
>
> We sincerely thank the reviewer for their positive assessment, specifically highlighting our method's novelty, soundness, and the extensiveness of our empirical evaluation, as well as noting that our work "can be considered a significant contribution".
>
> **W1. Scoring rules vs. Generative models & Batch size limitations** - Whether scoring rules or generative models (like diffusion) yield optimal probabilistic forecasts remains an active debate in the community. Recent empirical evidence from weather modelling (e.g., [1]) explores the trade-offs between stochastic interpolants, diffusion, and CRPS training, where they show the differences are relatively small. However, fully resolving this debate is heavily problem- and implementation-dependent.
>
> In the specific context of our work—adapting an existing deterministic checkpoint into a probabilistic one—we hypothesise that CRPS retrofitting is highly effective because its objective is structurally much closer to the model's original pre-training objective than a diffusion framework would be. We validate this empirically in Appendix E.3, where we demonstrate that diffusion models are less efficient at leveraging the representations learned during deterministic pre-training compared to our CRPS-retrofitted models.
>
> With respect to batch sizes, we did not do any analysis of the effect of the batch size, but this is not an inherent limitation of the framework. Larger effective batch sizes can easily be achieved via gradient accumulation, and memory bottlenecks can be mitigated via gradient checkpointing.
>
> **Q1. Clarification of probabilistic evaluation metrics** - We are happy to clarify this for a wider ML audience. CRPS is indeed a distributional metric (a proper scoring rule). When assessed empirically across an ensemble, CRPS effectively decomposes into the difference between the Mean Absolute Error (MAE) and the spread of the ensemble (see Equation 1).
>
> In highly chaotic systems, predicting the exact future trajectory is often impossible, meaning MAE will inevitably grow. However, if the system's inherent uncertainty is high, a probabilistic model should produce varied, diverse samples. This diversity increases the ensemble "spread". Because CRPS subtracts this spread from the error, the metric explicitly rewards models that accurately capture uncertainty when point-wise accuracy fails. Thus, we are not merely measuring the normalised distance of means, but rather evaluating the model's ability to produce a well-calibrated distribution.
>
> One limitation of CRPS is that it is strictly univariate and does not evaluate spatial correlations. To address this, we additionally evaluated the Energy Score (the multivariate generalisation of CRPS). As shown in the [Figures 1](https://anonymous.4open.science/r/ICML_Probabilistic_Retrofitting_CRPS-2EE5/CRPS_Energy_Score_Single_Models.png) and [2](https://anonymous.4open.science/r/ICML_Probabilistic_Retrofitting_CRPS-2EE5/CRPS_Energy_Score_HalfWalrus.png), our CRPS-retrofitted models demonstrate significant improvements in Energy Score over the deterministic baseline.
>
> In addition to that, we also computed frequency-space metrics, which is common in PDE modelling to assess the spectral properties of the samples. The results can be found [here](https://anonymous.4open.science/r/ICML_Probabilistic_Retrofitting_CRPS-2EE5/Table2_Power_Spectra_metrics.png) (also see Q3 from reviewer 2TFH).
>
> **Q2. Computational problems and training stability** - We did not observe any significant computational instabilities while training with scoring rules. As demonstrated in our ablation studies, the retrofitting procedure is remarkably stable and robust to hyperparameter choices. Most notably, the method proved highly robust to the number of ensemble members ($M$) used during training. We found that the model can be retrofitted successfully using as few as $M=2$ members (Table 11), which drastically reduces the computational overhead and memory requirements of the scoring rule without sacrificing significant performance.
>
> **References**
> 1. Kossaifi, J. et al. (2026). Demystifying Data-Driven Probabilistic Medium-Range Weather Forecasting.

---

> > ### Author Rebuttal · Reviewer_pKBT · 2026-03-31
> >
> > I thank the authors for the clarifications and for answering my questions. This answers my questions. I maintain my score and recommendation for acceptance.

---

> > > ### Author Response · Authors · 2026-04-05
> > >
> > > We sincerely thank the reviewer for their time, constructive feedback, and engagement throughout the review process. We are very glad that our clarifications fully answered your questions, and we deeply appreciate your support and recommendation for acceptance.

---

### Official Review · Reviewer_KYLh · 2026-03-12

**Soundness:** 3
**Presentation:** 3
**Significance:** 3
**Originality:** 1
**Overall Recommendation:** 4
**Confidence:** 3

**Summary:**

The paper presents a method to finetune existing deterministic models for physical systems using fairCRPS. CRPS is a probabilitic loss, that measures both accuracy and diversity. To add stochasticity to the models, the authors utilize the same noise injection technique as in [1].
This technique is applied on multiple models, with slight modifications for each model; And evaluated on multiple datasets from The Well benchmark. The models are single-dataset, and one foundational model, which is evaluated on datasets available in its training data.
The main claim of the model is that using CRPS for finetuning improves the model both on rollout VRMSE and, trivially, on CRPS.




Alet, Ferran, et al. "Skillful Joint Probabilistic Weather Forecasting from Marginals." 106th AMS Annual Meeting. AMS, 2026.

**Compliance With Llm Reviewing Policy:**

Affirmed.

**Final Justification:**

The rebuttal adressed most of my concerns, but I mantain my evaluation, since the lack of novelty of the proposed approach limits the ceiling of this paper.

**Key Questions For Authors:**

Q1: the authors argue that the baseline poseidon has not reached convergence, and that is why the results are comparable between finetuning and CRPS. But in the appendix the authors mention that the L-Poseidon checkpoint is finetuned for the datasets, to create the baseline. Why is the baseline not trained until convergence?

Q2: In 290 left, the authors mention a 16-member ensenble, but on the appendix 961, M defaults to 4, for all results. Why the choice of M=4? Is it 16 for inference in table 9, or 4?

Q3: for a comparable evaluation, the authors evaluate the CRPS finetuning against a deterministic finetuning. did the authors consider an ablation over the deterministic loss of the models?

Q4: Similarly, did you consider training M separate deterministic models with different seeds, to compare an ensemble with a deterministic loss against a probabilistic model with M samples?

Q5:The diffusion comparison is limited in details. please expand. for example, what is the target of the diffusion model: epsilon or x0?

Q6: The results for CRPS (scratch) are better than the baseline, for the rayleight-bénard dataset. What do you extract from this? is CRPS a better loss than MSE for certain datasets?

**Limitations:**

yes

**Strengths And Weaknesses:**

The paper presents thorough and compute intensive empirical evaluation of CRPS during the finetuning stage of deep learning models for physical systems. The results are significant and relevant to the overall comunity, but the techniques used are not novel, and have already been published in other works for a very similar task of weather forecasting.

Strenghts:
- Strong and thorought evaluation with multiple ablation studies.
- Clear introduction of similar work

weakness:
-  lack of detailed examination in the computational cost of the CRPS finetuning vs deterministic vs scratch.
-  the noise injection method requires adaptation for each model, which hinders the generability of the proposed method
- missing comparison against other common models for this task, such as FNO for example

---

> ### Author Rebuttal · Authors · 2026-03-30
>
> We thank the reviewer for finding our evaluation "thorough" and results "significant and relevant to the overall community".
>
> **W1. Originality and Novelty** - While the method is established, we are the first to show its success in retrofitting pre-trained PDE models. Our contribution is a novel, compute-efficient pathway to turn large deterministic checkpoints into probabilistic models, providing practical value without requiring training from scratch.
>
> **W2. Computational Cost** - We controlled for compute across all experiments: deterministic fine-tuning and CRPS retrofitting used the same wall-clock time. Detailed costs are provided throughout the appendix and summarised in Table 7. We will highlight this more prominently in the main text.
>
> **W3. Noise Injection Adaptation** - While injecting a shared global noise vector requires minor architectural adaptation depending on the backbone, these structural adjustments involve only a few lines of code. The procedure consistently follows Alg. 1, demonstrating the method is generalisable with little engineering overhead.
>
> **W4. Comparison to FNO** - Comparing retrofitted models against other deterministic baselines (like FNO) conflates architectural differences with the benefits of our stochastic treatment. To isolate the efficacy of probabilistic retrofitting, we controlled for the architecture by comparing against: 1) continued deterministic fine-tuning of the same baseline architectures, and 2) other probabilistic baselines like Diffusion (App. E.3) and our new Deep Ensemble (see Q4).
>
> **Q1. Baseline not trained to convergence** - While further training could yield marginal gains—indicated by a 3-5% improvement in rollout VRMSE during fine-tuning—we dedicated a budget of 3.2M additional samples per dataset to Poseidon, which is 4x the 800k budget used for Walrus.
>
> We hypothesise that a stronger factor contributing to Poseidon's performance ceiling is its reduced Markov history. As argued in the main text, this is probably what causes its overdispersion, especially for RB. Prior work [1] shows longer histories are important for datasets with high-frequency Fourier modes. Thus, we decided that investing additional compute into Poseidon would only yield diminishing returns, making our setup a realistic baseline for evaluating the efficiency of probabilistic retrofitting.
>
> **Q2. M definition** - M=4 is the training ensemble size for the CRPS loss. During inference, we always generate a 16-member ensemble for robust metrics (L290).
>
> **Q3. Ablation over deterministic loss** - In chaotic systems, MSE learns the conditional mean and MAE the median. At long horizons, both inevitably collapse to non-physical, smoothed states. Thus, ablating deterministic losses cannot address the fundamental limits of deterministic forecasting. CRPS retrofitting is not just a loss swap; it enables entirely new probabilistic behaviour, which is the core focus of our work. To empirically verify this, we fine-tuned Lola using MAE on RB and Shear Flow. As shown [here](https://anonymous.4open.science/r/ICML_Probabilistic_Retrofitting_CRPS-2EE5/Table5_MAE_vs_MSE_Lola.png), deterministic losses alone remain insufficient.
>
> **Q4. Comparison to Deep Ensembles (M separate models)** - We trained a Deterministic Deep Ensemble (fine-tuning 4 separate checkpoints with different seeds). Despite requiring 4x the compute, we found:
> - Superior Accuracy: CRPS-retrofitted models significantly outperform Deep Ensembles in VRMSE and CRPS (e.g., Walrus on Euler achieves a 37% CRPS improvement) - see [here](https://anonymous.4open.science/r/ICML_Probabilistic_Retrofitting_CRPS-2EE5/Deep_Ens_VRMSE_CRPS.png).
> - Better Calibration: Deep Ensembles exhibit underdispersion. CRPS models maintain a spread-to-skill ratio much closer to 1 (see [here](https://anonymous.4open.science/r/ICML_Probabilistic_Retrofitting_CRPS-2EE5/Table1_Deep_Ens_metrics.png)).
>
> This confirms performance gains stem directly from the CRPS objective, not just the presence of an ensemble.
>
> **Q5. Diffusion model details** - For the diffusion comparison (App. E.3), we followed Rozet et al. (Lola) using the EDM formulation. We will expand on configuration details in App. E.3 to ensure reproducibility.
>
> **Q6. CRPS (scratch) vs Baseline** - Given the same short training time, CRPS (Scratch) beats Deterministic FT only on the highly chaotic RB dataset; it is worse on Euler and unconverged on Shear Flow. We believe this implies that in chaotic systems like RB, point predictions fail quickly. Even a briefly CRPS-trained probabilistic model (or similarly, a deep ensemble) improves metrics via ensembling. In less chaotic systems, this no longer holds, and more training or a more thorough probabilistic treatment is needed for obtaining gains over the baseline.
>
> If our reply addresses your concerns, we would appreciate you reconsidering your score.
>
> **References**
> 1. Ruiz, R. B. et al. (2024). On the benefits of memory for modeling time-dependent pdes.

---

> > ### Author Rebuttal · Reviewer_KYLh · 2026-04-01
> >
> > The main limitation of the paper lies in its level of originality. While the work is clearly well executed and relies on large-scale computational resources, the overall contribution appears to be incremental rather than fundamentally novel.
> >
> > That said, the paper is well written, and the authors have provided satisfactory answers to my questions during the rebuttal phase. However, these clarifications do not substantially change my assessment of the paper’s novelty, and therefore I will maintain my original score.

---

> > > ### Author Response · Authors · 2026-04-05
> > >
> > > We sincerely thank the reviewer for their time, their engagement with our rebuttal, and their recognition of the scale, execution, and clarity of our work.
> > >
> > > We agree that the underlying techniques—such as noise injection and CRPS training—are established in fields like weather modelling. However, we respectfully argue that our core contribution lies in _how_ and _where_ we apply them, as well as the insights we provide with our empirical analysis.
> > >
> > > Specifically, our contribution is threefold:
> > >
> > > - **Retrofitting vs. Training from Scratch**: Unlike prior work (e.g., FGN [1]) that trains probabilistic models from scratch, we retrofit existing deterministic checkpoints. This introduces unique architectural and training challenges, such as balancing the noise branch's capacity to learn a useful spread without destroying the valuable representations acquired during deterministic pre-training.
> > >
> > > - **PDEs vs. Weather Modelling**: While the majority of weather modelling research focuses on a single dataset (ERA5), PDEs exhibit far greater heterogeneity. We provide the first comprehensive study applying CRPS training across various PDEs, demonstrating that performance gains are dependent on specific PDE characteristics, yet consistent.
> > >
> > > - **Thorough Empirical Investigation**: By rigorously investigating CRPS retrofitting across diverse systems, we provide valuable insights into the dynamics of transforming deterministic checkpoints into probabilistic models.
> > >
> > > Originality can also stem from combining existing techniques in novel ways to derive new insights, rather than solely from inventing new architectures. Demonstrating how to effectively leverage the community's existing pre-trained deterministic models for probabilistic PDE forecasting is a highly valuable step forward for the field. Indeed, this aligns with ICML’s own guidelines for what constitutes an original contribution.
> > >
> > > We deeply appreciate your constructive feedback, which has helped us clarify our contributions and enhace our quantitative analysis, and will ultimately lead to an improved manuscript.
> > >
> > > **References**
> > > 1. Alet, F. et al. (2025). Skillful joint probabilistic weather forecasting from marginals.

---

### Official Review · Reviewer_2TFH · 2026-03-13

**Soundness:** 3
**Presentation:** 3
**Significance:** 2
**Originality:** 2
**Overall Recommendation:** 5
**Confidence:** 3

**Summary:**

This paper leverage retrofitting method to fine-tune deterministic neural PDE solvers into stochastic ones, without incurring heavy costs from training from scratch. Noise embedding injects stochasticity through LayerNorm layer.
An unbiased Continuous Ranked Probabilistic Score is proposed to evaluate a stochastic PDE solver, describing both prediction loss and uncertainty support, initialized from intra-layer features from deterministic models.
The application of the proposed method to deterministic PDE foundation models, especially, avoids heavy-cost generative backbones and potentially reduce the energy consumption in future PDE foundation model training and inference.

**Compliance With Llm Reviewing Policy:**

Affirmed.

**Final Justification:**

The rebuttal comprehensively addressed all my concerns with quantitative evidence: Energy Score analysis (W1), Deep Ensemble comparison isolating the CRPS contribution (Q1), demonstrated scalability to 600M+ parameters (Q2), and power spectra metrics (Q3). Long-horizon stability (W2) remains a shared field-level challenge. Other reviewers' concerns about novelty and baselines were also addressed satisfactorily. This is a solid, well-executed contribution. I maintain my score of 5 (Accept).

**Key Questions For Authors:**

1. We are interested in seeing the independent performance gains given noise injection or CRPS objective.

2. How does the approach scale to very large models?

3. Can the retrofitting process be proven to improve physical consistencies, as we are discussing PDE foundation models here?

**Limitations:**

Yes.

**Strengths And Weaknesses:**

Strengths:
1. The proposed retrofitting method with CRPS objective is conceptually simple and grounded in established probabilistic forecasting practices.
2. The evaluation across single dynamics solver and PDE foundation models is comprehensive and inspiring to the community.
3. Ablation studies answered the common concerns for the probabilistic method.
4. The low-cost fine-tuning method for a probabilistic foundation model is of significance.

Weaknesses:
1. CRPS is still univariate marginal distribution score, may not guarantee spatial or multivariate consistency.
2. Long-horizon stability remains unsolved.

---

> ### Author Rebuttal · Authors · 2026-03-29
>
> We thank the reviewer for highlighting our evaluation protocol as "comprehensive and inspiring" and recognising the significance of our low-cost retrofitting procedure for foundation models.
>
> **W1. CRPS Univariate limitation** - This is a valid concern that we acknowledge in the paper (L185). While CRPS evaluates univariate marginals, we—like [1]—did not observe catastrophic failures in spatial consistency. As outlined in [1], injecting a low-dimensional global noise source shared across all spatial dimensions naturally encourages spatially coherent fields.
>
> To quantitatively confirm this spatial and multivariate consistency, we computed the Energy Score (the multivariate generalisation of CRPS, which captures spatial correlations). As shown in [Figures 1](https://anonymous.4open.science/r/ICML_Probabilistic_Retrofitting_CRPS-2EE5/CRPS_Energy_Score_Single_Models.png) and [2](https://anonymous.4open.science/r/ICML_Probabilistic_Retrofitting_CRPS-2EE5/CRPS_Energy_Score_HalfWalrus.png), the Energy Score for our CRPS-retrofitted models significantly outperforms the deterministically fine-tuned baselines. The relative Energy Score improvements closely mirror CRPS improvements across single-system models and HalfWalrus, proving our univariate gains translate to multivariate consistency.
>
> **W2. Long-horizon stability** - We agree this is an open challenge. Future work could address this by integrating multi-step rollout fine-tuning [1] or applying frequency-space regularisation [2, 3]. Combining these techniques with CRPS retrofitting should further stabilise long-term rollouts.
>
> **Q1. Independent performance gains from noise injection vs. CRPS objective** - Because the CRPS objective inherently requires an ensemble of predictions (and thus a stochastic mechanism to generate them), the noise injection and the objective are coupled during training. To isolate the value of the CRPS objective, we compared our method against a Deterministic Deep Ensemble (fine-tuning 4 separate times with different initialisations). Despite Deep Ensembles requiring 4x the compute of a single CRPS run, we found:
>
> - Superior Accuracy: CRPS-retrofitted models significantly outperform Deep Ensembles in both VRMSE and CRPS across datasets (e.g., Walrus on Euler achieves a 37% improvement) (see [Figure 3](https://anonymous.4open.science/r/ICML_Probabilistic_Retrofitting_CRPS-2EE5/Deep_Ens_VRMSE_CRPS.png)).
>
> - Better Calibration: Deep Ensembles exhibit underdispersion. While CRPS models are not perfectly calibrated, their spread-to-skill ratio is much closer to 1 throughout the trajectory length ([Table 1](https://anonymous.4open.science/r/ICML_Probabilistic_Retrofitting_CRPS-2EE5/Table1_Deep_Ens_metrics.png)).
>
> This confirms the performance gains are largely driven by the CRPS objective itself, not just the presence of an ensemble.
>
> **Q2. Scaling to very large models** - Our evaluated models are already big by PDE standards (Poseidon and HalfWalrus exceed 600M parameters, in comparison to 180M in [1]). CRPS retrofitting does not introduce significant scaling bottlenecks compared to deterministic training. It only requires a batch size large enough for the ensemble. Because our method works effectively with as few as two members during training, it scales to massive foundation models in a similar way to standard deterministic fine-tuning. Additionally, our noise injection mechanism follows standard DiT conditioning, a scalable architecture that should not bottleneck scaling.
>
> **Q3. Proof of improved physical consistency** - Following [4], we assessed statistical plausibility by computing the RMSE between the logarithms of the isotropic power spectra of the ground-truth and emulated fields. We evaluated this across low, mid, and high-frequency bands for Lola and Walrus on the RB and Euler datasets ([Table 2](https://anonymous.4open.science/r/ICML_Probabilistic_Retrofitting_CRPS-2EE5/Table2_Power_Spectra_metrics.png)).
>
> - Lola: Power spectra metrics strictly improve for the CRPS-retrofitted models across all bands and all datasets.
>
> - Walrus: CRPS models show improved consistency in low- and mid-frequency bands. We observe some deterioration in the high-frequency band, but this is a known architectural limitation of Walrus, which produces grid-like artifacts [5]. As noted in our text, retrofitting can sometimes exacerbate these existing artifacts, but the overall spectral distribution remains highly competitive.
>
> **References**
> 1. Alet, F. et al. (2025). Skillful joint probabilistic weather forecasting from marginals.
> 2. Kossaifi, J. et al. (2026). Demystifying Data-Driven Probabilistic Medium-Range Weather Forecasting.
> 3. Nordhagen, E. M. et al. (2025). High-resolution probabilistic data-driven weather modeling with a stretched-grid.
> 4. Ruben Ohana et al. (2024). “The Well: a Large-Scale Collection of Diverse Physics Simulations for Machine Learning”.
> 5. McCabe, M. et al. (2025). Walrus: A cross-domain foundation model for continuum dynamics.

---

> > ### Author Rebuttal · Reviewer_2TFH · 2026-03-31
> >
> > The rebuttal comprehensively addresses my concerns with quantitative evidence:
> >
> > - **W1 (univariate CRPS):** The Energy Score analysis directly resolves my concern about multivariate consistency. The improvements in Energy Score mirroring CRPS gains are convincing.
> > - **Q1 (noise injection vs CRPS):** The Deep Ensemble comparison (4× compute, yet inferior in both VRMSE and calibration) cleanly isolates the CRPS objective as the driver of gains.
> > - **Q2 (scaling):** The existing evaluation on Poseidon and HalfWalrus (600M+ parameters) sufficiently demonstrates scalability.
> > - **Q3 (physical consistency):** The power spectra metrics across frequency bands provide the evidence I requested.
> >
> > W2 (long-horizon stability) remains an open field-level challenge. The authors appropriately acknowledge this and suggest promising future directions (multi-step rollout fine-tuning, frequency-space regularisation). I do not consider this a weakness of the current paper, as it is a shared limitation across the field.
> >
> > I maintain my current score.

---

> > > ### Author Response · Authors · 2026-04-05
> > >
> > > We sincerely thank the reviewer for their time, constructive feedback, and for engaging so thoughtfully with our rebuttal. We are delighted that our added quantitative analyses comprehensively addressed your concerns.
> > >
> > > Thank you again for your support for our work, and for helping us improve our manuscript.

---

### Decision · Program_Chairs · 2026-04-30

**Decision:**

Accept (regular)

**Comment:**

This paper proposes a CRPS-based method that converts deterministic neural PDE solvers into a stochastic neural PDE solvers without requiring retraining from scratch, and evaluates it on multiple datasets.

The reviewers generally agree that the problem addressed by this paper is meaningful, the evaluation is comprehensive, and the presentation is clear. However, they have concerns about the novelty of the proposed algorithm compared with previous works and the fairness of the comparisons with other methods. Meanwhile, the authors' rebuttal resolves most of these concerns and the reviewers' overall attitude is positive, so I recommend acceptance.